# Loss of CDK4/6 activity in S/G2 phase leads to cell cycle reversal

James A. Cornwell[1], Adrijana Crncec[1], Marwa M. Afifi[1], Kristina Tang[1], Ruhul Amin[1] & Steven D. Cappell[1✉]

In mammalian cells, the decision to proliferate is thought to be irreversibly made at the restriction point of the cell cycle[1,2], when mitogen signalling engages a positive feedback loop between cyclin A2/cyclin-dependent kinase 2 (CDK2) and the retinoblastoma protein[3–5]. Contrary to this textbook model, here we show that the decision to proliferate is actually fully reversible. Instead, we find that all cycling cells will exit the cell cycle in the absence of mitogens unless they make it to mitosis and divide first. This temporal competition between two fates, mitosis and cell cycle exit, arises because cyclin A2/CDK2 activity depends upon CDK4/6 activity throughout the cell cycle, not just in G1 phase. Without mitogens, mitosis is only observed when the half-life of cyclin A2 protein is long enough to sustain CDK2 activity throughout G2/M. Thus, cells are dependent on mitogens and CDK4/6 activity to maintain CDK2 activity and retinoblastoma protein phosphorylation throughout interphase. Consequently, even a 2-h delay in a cell's progression towards mitosis can induce cell cycle exit if mitogen signalling is lost. Our results uncover the molecular mechanism underlying the restriction point phenomenon, reveal an unexpected role for CDK4/6 activity in S and G2 phases and explain the behaviour of all cells following loss of mitogen signalling.

The restriction point (R point) marks the point in the cell cycle when mammalian cells become independent of mitogen signalling and are irreversibly committed to proliferation[1,2]. Molecularly, this irreversible cell fate decision is thought to arise because cells convert extracellular mitogen signals into a self-sustaining positive feedback loop (Fig. 1a). Mitogens activate cyclin-dependent kinases 4 and 6 (CDK4/6), which phosphorylate the tumour suppressor retinoblastoma protein (Rb), leading to activation of the transcription factors E2F1–3. These E2Fs promote transcription of cyclins E and A, which form complexes with CDK2 that also phosphorylate Rb and further drive E2F-mediated transcription of cyclins E and A. Thus, once activated, CDK2 is thought to form a positive feedback loop with Rb that can maintain continuous CDK2 activity even in the absence of upstream mitogen signalling[6], exhibiting properties of bistability and irreversible hysteresis with respect to mitogen concentration (Fig. 1b)[7,8]. Therefore, the textbook model of the R point is that CDK2 activation and Rb phosphorylation determine the transition from a pre-R to post-R state[3–5]. However, recent studies have also found that CDK2 activity and Rb phosphorylation status cannot determine whether all cells have crossed the R point, observing outlier cells that appear to contradict the textbook model[3,4,9–11]. This discrepancy calls into question both the long-standing R-point model and our basic understanding of mammalian cell cycle control. Thus, new single-cell studies are needed to uncover a universal model of cell cycle control.

## Mitogen signalling maintains CDK2 activity in S/G2

The principal tenet of the R-point model is that pre-R cells are sensitive to loss of mitogen signalling and will exit the cell cycle to G0 if mitogens are removed, whereas post-R cells are insensitive and will complete mitosis, after which their daughter cells will arrest in G0 (refs. 9,12) (Fig. 1c). To test this fundamental prediction of the R-point model, we serum starved MCF-10A cells or treated them with a mitogen-activated protein kinase kinase inhibitor (MEKi) or a cyclin-dependent kinases 4 and 6 inhibitor (CDK4/6i) for 48 h and then, measured DNA content and Rb phosphorylation[13]. Contrary to the R-point model, as many as 15% of MCF-10A cells had 4N DNA content (4 copies of each chromosome, for example diploid), indicating that these cells did not complete mitosis and arrest in G0 but instead, exited the cell cycle from G2 phase (Fig. 1d, histograms). However, unlike a typical G2 arrested cell with hyperphosphorylated Rb, these cells had hypophosphorylated Rb (Fig. 1d, scatterplots). Thus, rather than all cells arresting in G0 with 2N DNA content following loss of mitogen signalling, we found that some cells entered a 'G0-like' state with 4N DNA content and hypophosphorylated Rb, suggesting that the proposed feedback loop between CDK2 and Rb was not indefinitely maintained in these cells. Indicative of a general phenomenon, we recapitulated this cell cycle state in primary cells, non-transformed cells, transformed cells and cells lacking the stress-induced CDKis p21, p27 and p16 (Extended Data Fig. 1).

To assess how loss of mitogen signalling affected CDK2 activity and Rb phosphorylation, we transduced MCF-10A cells with a CDK2 activity sensor (Extended Data Fig. 2a), the activation of which corresponds with Rb phosphorylation[4]. Since it has been shown that the R point precedes S phase[14] and inactivation of the anaphase-promoting complex/cyclosome (APC/C) marks entry into S phase[15], we utilized an APC/C activity sensor[16] to identify pre-R (APC/C on) or post-R (APC/C off) cells (Extended Data Fig. 2a,b). We combined these activity sensors

[1]Laboratory of Cancer Biology and Genetics, Center for Cancer Research, National Cancer Institute, Bethesda, MD, USA. ✉e-mail: steven.cappell@nih.gov

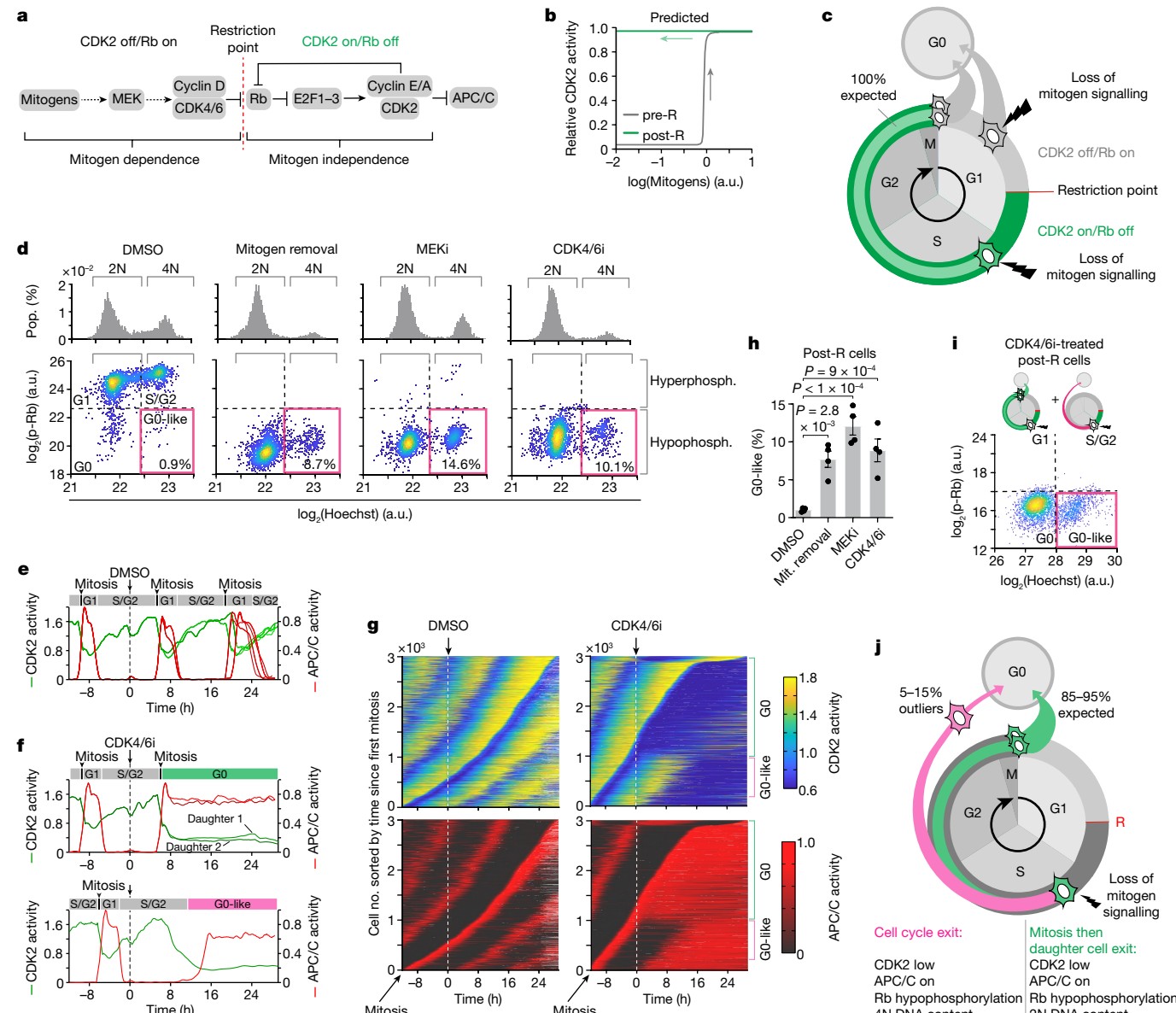

**Fig. 1 | Mitogen signalling maintains CDK2 activity in S/G2. a**, Textbook signalling pathway indicating that the R point marks the switch from mitogen dependence to independence. **b**, Mathematical model adapted from Yao et al.[8] showing bistability and hysteresis in CDK2 activity with respect to mitogen signalling. **c**, Predicted fates for pre- and post-R cells made by the R-point model. **d**, Histograms show DNA content (upper panels). Scatterplots of Rb phosphorylation versus DNA content (lower panels). Pink boxes mark the G0-like state (hypophosphorylated Rb and 4N DNA content). The percentage of G0-like cells is indicated. *N* = 2,000 cells per condition. **e**, CDK2 and APC/C activity from an example MCF-10A cell treated with DMSO at the indicated time. The cell divides multiple times, giving rise to four granddaughter cells (Supplementary Video 1). **f**, CDK2 and APC/C activity from two example MCF-10A cells treated with CDK4/6i at the indicated time. In the upper panel, the cell divides, and its daughters arrest in G0. In the lower panel, the cell exits

the cell cycle to a G0-like state without dividing (Supplementary Videos 2 and 3). **g**, Heat maps show CDK2 and APC/C activity sorted by time of mitosis for cells treated with DMSO (left panels) or a CDK4/6i (right panels). Extended Data Fig. 2b demonstrates how CDK2 and APC/C activities are converted to the heat map. **h**, Percentages of post-R cells that exit to the G0-like state after mitogen (Mit.) removal, MEKi or CDK4/6i. Error bars represent s.e.m. from *n* = 4 independent experiments. *P* values were calculated using a one-way analysis of variance. *P* values from top to bottom are $9 \times 10^{-4}$, less than $1 \times 10^{-4}$ and $2.8 \times 10^{-3}$. **i**, Scatterplot of Rb phosphorylation versus DNA content for CDK4/6i-treated post-R cells from **g** showing two distinct cell cycle trajectories for post-R cells after loss of mitogen signalling. The pink box indicates the G0-like state, and cartoons (upper panel) show cell cycle trajectories. *N* = 3,621 cells. **j**, Schematic showing observed fate outcomes for post-R cells after loss of mitogen signalling. a.u., arbitrary unit. phosph., phosphorylation.

with live-cell imaging and automated single-cell tracking to measure CDK2 activity after mitogen removal, MEK inhibition or CDK4/6 inhibition in post-R cells.

Post-R cells treated with dimethylsulfoxide (DMSO) divided repeatedly (Fig. 1e,g and Supplementary Video 1), while treatments perturbing the mitogen signalling pathway resulted in two distinct cell cycle trajectories (Fig. 1f,g and Extended Data Fig. 2b,c). Most post-R cells built up

CDK2 activity until they divided into two daughter cells, which arrested in G0 with low CDK2 activity and high APC/C activity (Fig. 1f, upper panel and Supplementary Video 2). However, up to 15% of post-R cells did not enter mitosis but instead, gradually lost CDK2 activity and prematurely reactivated the APC/C (Fig. 1f, lower panel, Fig. 1h and Supplementary Videos 3, 4 and 5), indicating that these post-R cells had exited the cell cycle to a G0-like state (CDK2 low, APC/C on, hypophosphorylated

Rb and 4N DNA content) (Fig. 1i). Notably, entry into this G0-like state explained the appearance of a persistent G2 population when looking at DNA content alone (Fig. 1d, pink boxes). Therefore, in contrast to the textbook model of the R point (Fig. 1c), we find that not all post-R cells are irreversibly committed to proliferation since some cells exit the cell cycle to a G0-like state after loss of mitogen signalling (Fig. 1j). Furthermore, these G0-like cells eventually exhibited senescence-associated β-galactosidase activity (Extended Data Fig. 2d–f), consistent with previous reports that premature APC/C reactivation in S/G2 phase is a precursor to cellular senescence[17,18].

## Competition between mitosis and exit determines cell fate

To understand why these outlier post-R cells failed to enter mitosis after loss of mitogen signalling, we sorted all cells by time of treatment with respect to S-phase entry (that is, APC/C inactivation) and found that cells were more likely to exit the cell cycle when closer to the start of S phase than mitosis when treated (Fig. 2a and Extended Data Fig. 3). Since mitosis and cell cycle exit are mutually exclusive fates that preclude the observation of each other[19], we hypothesized that a cell closer to mitosis when mitogen signalling was lost may not have had enough time to respond and would enter mitosis instead of exiting the cell cycle, while a cell closer to S phase would exit the cell cycle instead of entering mitosis. This temporal competition between opposing fate outcomes can be conceptualized as a competition between two molecular clocks, representing the time taken to progress from S phase to mitosis (the mitosis clock) and the time required to lose CDK2 activity after loss of mitogen signalling (the cell cycle exit clock) (Fig. 2b). If, for a given cell, its cell cycle exit clock is longer than its mitosis clock, then mitosis wins the competition and cell cycle exit would not be observed and vice versa.

The competing clock model makes two main predictions. The first prediction is that since all cells contain both a mitosis clock and a cell cycle exit clock that operate independently, blocking either clock would allow the opposing fate to win the competition (Fig. 2c). To test this first prediction, we tipped the balance in favour of mitosis by growing cells in full-growth media and observed that nearly all cells reached mitosis, revealing an underlying 'pre-competition' distribution of mitosis times (Fig. 2d,e and Extended Data Fig. 4d–f). Conversely, we tipped the balance in favour of cell cycle exit by disabling the mitosis clock using a CDK1i (Extended Data Fig. 4a,b) at the same time as blocking mitogen signalling (Fig. 2d and Extended Data Fig. 4a–c). With the mitosis clock disabled, nearly all post-R cells lost CDK2 activity and exited the cell cycle after loss of mitogen signalling, revealing an underlying 'pre-competition' distribution of cell cycle exit times that could only be revealed by blocking mitosis (Fig. 2e and Extended Data Fig. 4c). Disabling the mitosis clock by arresting cells in G2 phase with a small molecule inhibitor of Polo-like kinase 1, a key regulator of the G2/M transition, or triggering the G2/M DNA damage checkpoint using the radiomimetic drug neocarzinostatin yielded similar results (Extended Data Fig. 4g,h). Thus, by disabling the mitosis clock, we found that all post-R cells and not just the 10–15% of outlier cells contain a cell cycle exit clock. This means that all post-R cells are unable to sustain CDK2 activity in the absence of CDK4/6 signalling, in stark contrast to the textbook model in which CDK2 activity is self-sustaining.

The second prediction made by the competing clock model is that that knowledge of the 'pre-competition' mitosis and cell cycle clock distributions alone should be sufficient to explain whether a cell will decide to enter mitosis or exit the cell cycle after loss of mitogen signalling. To test this second prediction, we first measured the 'pre- and post-competition' distributions of mitosis and cell cycle exit times (Fig. 2e,f and Extended Data Fig. 5a,b). We then used a Monte Carlo algorithm to randomly sample from the 'pre-competition' distributions for cell cycle exit and mitosis and simulated a competition

(Extended Data Fig. 5c). For the competing clock model to be correct, the simulated competition should match the experimentally measured 'post-competition' distributions of cell cycle exit and mitosis times. Cell cycle exit and mitosis were modelled as independent competing processes because once the mitosis clock is disabled, the probability of cell cycle exit was independent of a cell's proximity to mitosis upon treatment (Extended Data Fig. 5d). Results from our simulation agreed with our experimental observations for CDK4/6i-treated cells, including the frequency of cell cycle exit (Fig. 2g,h and Extended Data Fig. 5e), 'post-competition' cell cycle exit clock times (Fig. 2h and Extended Data Fig. 5f) and the relationship between proximity to the start of S phase and the probability of cell cycle exit (Extended Data Fig. 5g). Notably, despite absolute differences in the cell cycle exit and mitosis clocks between cell lines (Extended Data Fig. 5h–j), we found that the simulations agreed with experimental observations from multiple cell lines (Extended Data Fig. 5k–m). These simulations indicate that the reason some cells are observed to exit the cell cycle in G2 phase is because these cells had a shorter cell cycle exit clock than their mitosis clock and that the relative difference in the timing of each clock combined with cell-to-cell variability at the population level can account for the frequency of cells that exit the cell cycle upon loss of mitogen signalling that we observed experimentally (Fig. 2i). Thus, our data support an alternative model for cell cycle commitment that can account for the behaviour of all cells after loss of mitogen signalling.

An important implication of this revised model of cell cycle commitment is that cells have limited time to complete the cell cycle given that the median 'pre-competition' cell cycle exit and mitosis clock times differed on average by only 4 h (for example, 15 versus 11 h, respectively, in MCF-10A cells) (Fig. 2e and Extended Data Fig. 5h–j). To test this, we sought to extend the mitosis clock rather than blocking it completely by treating cells transiently with either hydroxyurea or thymidine, which stalls DNA replication in S phase without causing DNA damage (Extended Data Fig. 6a–d), to extend the time from S phase to mitosis of post-R cells by 2-h increments (Extended Data Fig. 6e,f). Imposing this delay on the mitosis clock in combination with CDK4/6i treatment increased the frequency of cell cycle exit as a function of the increase in the mitosis clock (Fig. 2j and Extended Data Fig. 6g–i), revealing that in the absence of mitogen signalling, cells have limited time to complete the cell cycle (Fig. 2k). While we observed a drop in CDK2 activity following hydroxyurea or thymidine treatment, we could rescue this drop by inhibiting Wee1, a kinase that negatively regulates CDKs (Extended Data Fig. 6j–l). Again, we observed cells exiting the cell cycle into the G0-like state when treated with the CDK4/6i and as little as a 4-h pulse of thymidine with or without the Wee1 inhibitor. Thus, we extended S/G2 length without profoundly interfering with CDK2 activity, and we still observe cells exiting into the G0-like state after CDK4/6 inhibition.

## CDK4/6 promotes cyclin A2 synthesis in S/G2

Our data demonstrating that CDK2 activation and Rb phosphorylation are reversible in all post-R cells after loss of mitogen signalling raise the question of whether CDK2 and Rb comprise a bona fide self-sustaining feedback loop. To address this, we measured each component of the proposed CDK2–Rb feedback loop in post-R cells after CDK4/6i treatment (Fig. 3a). While cyclin E activates CDK2 in G1 phase, it is degraded[20–22] and does not contribute to CDK2 activity in S/G2 phase; therefore, it was not included in this analysis (Extended Data Fig. 7a,b). Cyclin A2 protein levels, CDK2 activity, Rb phosphorylation and E2F1 mRNA remained high for 6–8 h before declining concomitantly, while cyclin A2 mRNA levels fell within 2 h (Fig. 3b and Extended Data Fig. 7c–e). Real-time quantitative reverse-transcriptase PCR (qRT–PCR) analysis confirmed that a 2-h CDK4/6i treatment reduced cyclin A2 mRNA levels in post-R cells by 50%, while mRNA levels of canonical E2F target genes (CCNE1 and E2F1) remained unchanged (Extended Data Fig. 7f).

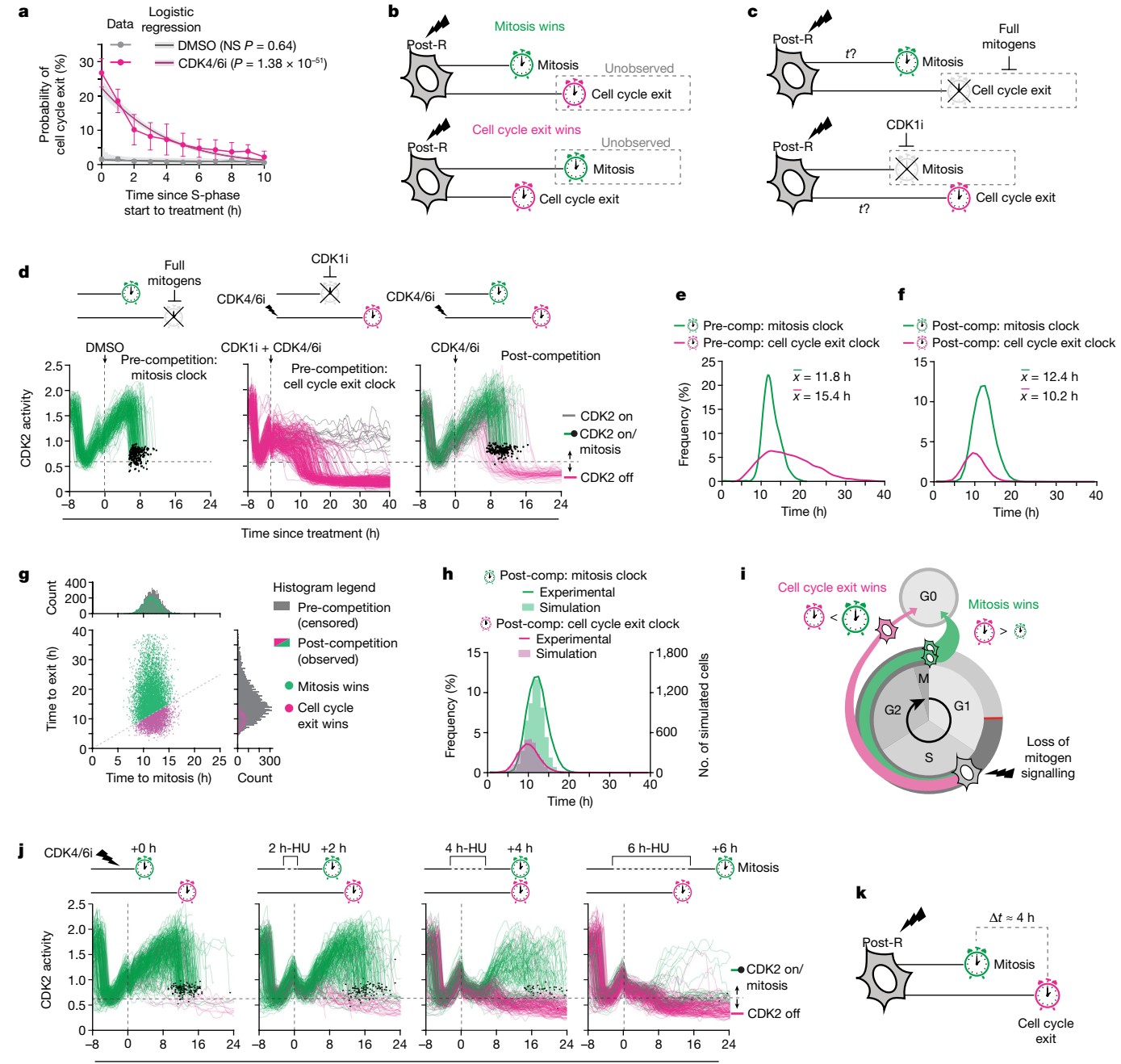

**Fig. 2 | Competition between mitosis and exit determines cell fate. a**, The observed percentages of post-R cells exiting in each bin for DMSO (grey circles) and CDK4/6i treatment (pink circles) are shown. Error bars represent s.e.m. from $n = 3$ experiments. A logistic regression model was fitted to the data. Shaded regions represent 95% confidence intervals. $P$ values were calculated from a logistic regression model using a two-tailed Wald test: DMSO ($P = 0.64$) and CDK4/6i ($P = 1.38 \times 10^{-51}$). **b**, Schematic illustrating temporal competition between mitosis and cell cycle exit for a post-R cell that lost mitogen signalling in S/G2 phase. **c**, Schematic illustrating that inhibition of either the mitosis clock or the cell cycle exit clock enables measurement of the pre-competition cell cycle exit clock or mitosis clock, respectively. **d**, Single-cell traces of CDK2 activity aligned to time of treatment. Green lines depict cells that entered mitosis (indicated by black dots). Grey lines depict cells that remain committed to the cell cycle with high CDK2 activity (greater than 0.6). Pink lines depict cells that lost CDK2 activity (less than 0.6) and exited the cell cycle. $N = 232$, 299 and 242 cells, respectively. **e**, Histograms showing pre-competition distributions of the mitosis and cell cycle exit clocks measured from the left and middle panels of **d**, respectively. **f**, Histograms showing post-competition distributions

of the mitosis and cell cycle exit clocks measured from the right panel of **d**. **g**, Monte Carlo simulation showing post-competition distributions for mitosis (green histogram) and cell cycle exit (pink histogram) overlaid over their respective pre-competition distributions (grey histograms). The scatterplot shows simulated times for cell cycle exit and mitosis colour coded by whether mitosis (green dots) or cell cycle exit (pink dots) won the competition. **h**, Histograms of post-competition times for cell cycle exit and mitosis. Lines represent experimentally measured distributions, and solid bars represent simulated distributions from **g**. **i**, Schematic showing that after loss of mitogen signalling, the difference in timing of the cell cycle exit and mitosis clocks determines whether a cell will exit the cell cycle or reach mitosis. **j**, CDK2 activity traces from MCF-10A p21$^{-/-}$ cells aligned to the time of treatment with hydroxyurea (HU) and CDK4/6i coloured as in **d**. $N = 300$, 286, 300 and 297 cells, respectively. **k**, Schematic illustrating that after loss of mitogen signalling in post-R cells, CDK2 activity can be maintained for approximately 15 h, which is approximately 4 h longer than the median time to enter mitosis. NS, not significant.

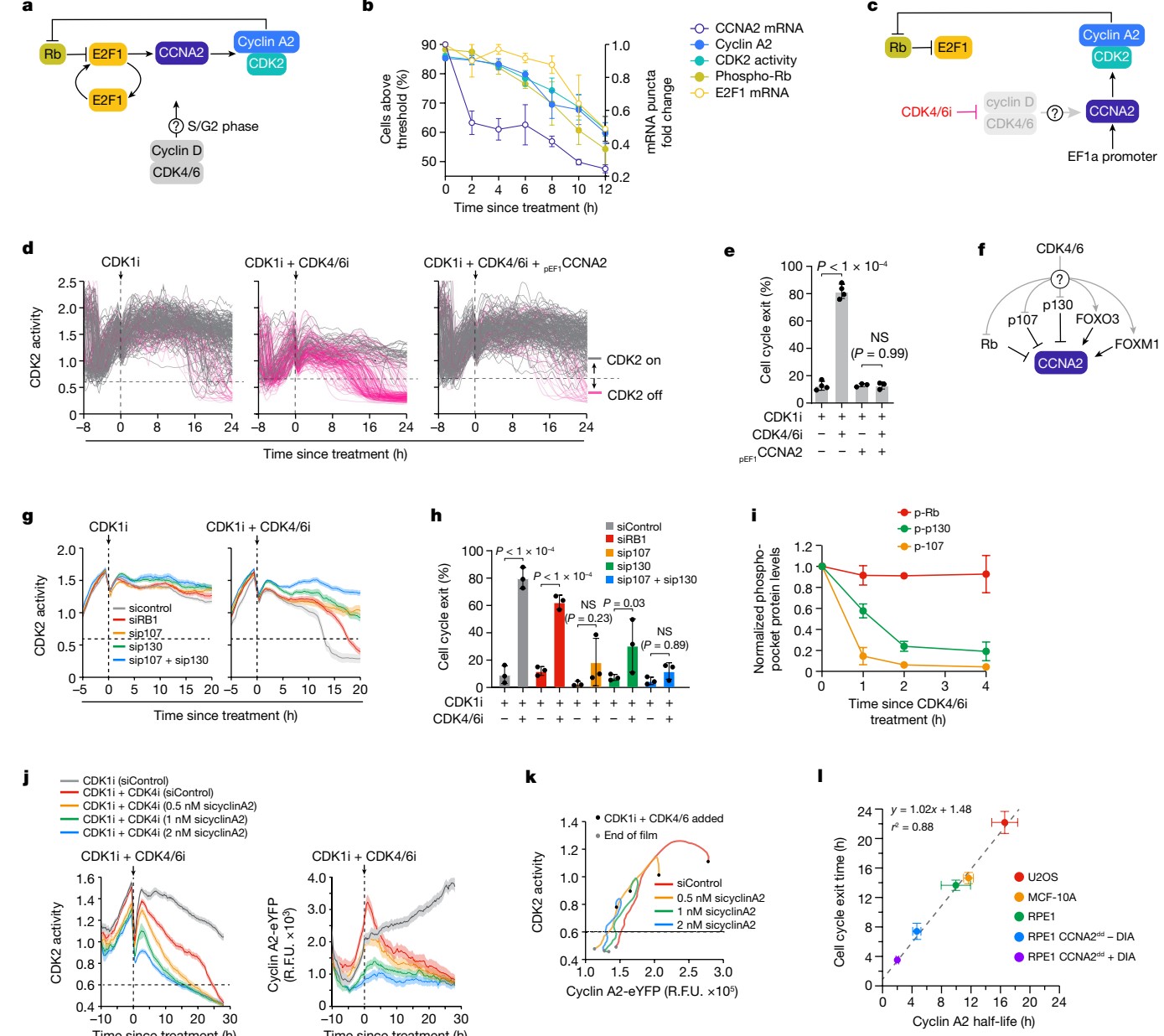

**Fig. 3 | CDK4/6 promotes cyclin A2 synthesis in S/G2. a**, Proposed CDK2–Rb feedback loop model. **b**, Measurements of each component in **a** taken after treatment with CDK1i + CDK4/6i. $n = 2$ for CCNA2 mRNA and E2F1 mRNA, $n = 3$ for cyclin A2 protein and CDK2 activity and $n = 4$ for phospho-Rb levels. Error bars represent s.e.m. **c**, Experimental design to test if cyclin A2 expression from an unregulated promoter can rescue loss of CDK2 activity after CDK4/6i treatment. **d**, CDK2 activity traces for cells treated as indicated. Grey and pink traces represent cells with CDK2 greater than 0.6 and CDK less than 0.6 activity at 24 h, respectively. $N = 200$, 209 and 200 cells, respectively. **e**, Percentages of cells that exited the cell cycle from **d**. Error bars represent s.e.m. from $n = 4$ independent experiments. $P$ values were calculated using a one-way analysis of variance. $P$ values from top to bottom are less than $1 \times 10^{-4}$ and 0.99. **f**, CDK4/6 substrates that are proposed regulators of CCNA2 transcription. **g**, Median CDK2 activity traces for cells aligned to time since treatment for the indicated conditions. Shaded regions represent 95% confidence intervals. $N > 1,000$ cells per condition. Dotted horizontal lines represent CDK2 activity below the threshold required for Rb phosphorylation (CDK2 less than 0.6). **h**, Percentages of cells that exited the cell cycle from **g**. Error bars represent s.e.m. from $n = 3$ independent experiments. $P$ values were calculated using a one-way analysis of variance. $P$ values from left to right are less than $1 \times 10^{-4}$, less than $1 \times 10^{-4}$, 0.23, 0.03 and 0.89. **i**, Quantification of phosphorylated pocket proteins after CDK4/6i treatment. Levels of each phosphoprotein were normalized to the total levels of the respective protein. Error bars represent s.e.m. (Rb, $n = 5$; p107, $n = 2$; p130, $n = 3$ independent experiments). **j**, Median CDK2 activity traces (left panel) and endogenous cyclin A2 levels (right panel) aligned to time since treatment for U2OS-eYFP cells. Shaded regions indicate 95% confidence intervals. **k**, Phase plot of median CDK2 activity and median endogenous cyclin A2 levels. **l**, Cell cycle exit times as a function of cyclin A2 half-life for indicated cell lines. Equations for the best-fit line and $r^2$ value are shown. Error bars represent s.e.m. (U2OS, $n = 5$; MCF-10A, $n = 6$; RPE-1, $n = 4$; RPE-1 CCNA2$^{dd}$-DIA, $n = 2$; RPE CCNA2$^{dd}$ + DIA, $n = 2$ independent experiments). R.F.U., relative fluorescence units.

Expression of cyclin A2 from an unregulated promoter prevented post-R cells from losing CDK2 activity after CDK4/6i treatment, establishing that repression of CDK4/6-mediated cyclin A2 transcription is the cause of cell cycle exit (Fig. 3c–e).

To identify proteins regulated by CDK4/6 that may mediate the transcription of cyclin A2, we tested whether small-interfering RNA (siRNA) knockdown of various known CDK4/6 substrates that control cell cycle gene transcription could rescue the loss of CDK2 activity

(Fig. 3f). Knockdown of the CDK4/6-regulated transcription factors FOXO3 and FOXM1 did not rescue the loss of CDK2 activity (Extended Data Fig. 7g–i), nor did knockdown of Rb (Fig. 3g,h and Extended Data Fig. 8a,b). However, knockdown of p107 and p130, two members of the same protein family as Rb, did rescue loss of CDK2 activity (Fig. 3g,h and Extended Data Fig. 8a,b). Likewise, knockdown of p107 and p130, but not Rb, prevented loss of cyclin A2 mRNA after CDK4/6i treatment (Extended Data Fig. 8c), indicating that CDK4/6 represses cyclin A2 transcription primarily through p107/p130 and not Rb. Consistent with this mode of regulation, we found that p107 and p130 were dephosphorylated within 1–2 h of CDK4/6i treatment, while Rb remained hyperphosphorylated, indicating that p107 and p130, but not Rb, were activated by CDK4/6i treatment (Fig. 3i and Extended Data Fig. 8d–f). Activated forms of p107 and p130 repress transcription through complex formation with repressor E2Fs E2F4 and E2F5[23–26]. Consistent with this, CDK4/6 inhibition induced complex formation between E2F4 and p107 (Extended Data Fig. 8g), and combined knockdown of E2F4 and E2F5 also prevented the loss of CDK2 activity (Extended Data Fig. 8h–j). Therefore, we find that CDK4/6–p107/p130 regulate cyclin A2 transcription and CDK2 activity in post-R cells, implying a critical role for CDK4/6 activity in maintaining CDK2 activity in S/G2 phase.

Given that CDK2 activity is driven by cyclin A2 in post-R cells[4] and that the cyclin A2 protein has an approximately 12-h half-life in post-R cells (Extended Data Fig. 9a,b), this suggests that cyclin A2 protein levels are the primary contributor to the timing of the cell cycle exit clock, which had a median time of approximately 15 h. To test this, we expressed the CDK2 biosensor in U2OS cells with cyclin A2 endogenously tagged with yellow fluorescent protein (YFP), incubated them with cyclin A2 siRNA (Extended Data Fig. 9c,d) and then, measured endogenous cyclin A2 levels and CDK2 activity over time after CDK1i + CDK4/6i treatment. In keeping with our prediction that cyclin A2 protein levels set the timing of the cell cycle exit clock, we found that cells with lower starting levels of cyclin A2 took less time to lose CDK2 activity and exit the cell cycle after loss of mitogen signalling (Fig. 3j,k), that the time to lose CDK2 activity and exit the cell cycle was strongly correlated with the time to lose cyclin A2 protein (Extended Data Fig. 9e) and that the cyclin A2 levels at the time mitogen signalling was lost were predictive of whether cells reached mitosis or exited the cell cycle (Extended Data Fig. 9f,g). Furthermore, in support of the mitosis and cell cycle exit clocks being driven by independent molecular processes, we noted that only a small reduction in cyclin A2 levels was able to shorten the timing of the cell cycle exit clock, while the duration of the mitosis clock remained unchanged (Extended Data Fig. 9h).

To further test whether cyclin A2 protein stability is the primary contributor to the timing of the cell cycle exit clock, we directly manipulated cyclin A2 protein stability using a CRISPR-engineered RPE-1 cell line where the endogenous cyclin A2 protein was fused to two inducible degrons: an auxin-inducible degron and a small molecule-assisted shutoff (SMASh) tag[27,28] (RPE-1 CCNA2[dd] cells) (Extended Data Fig. 10a). The addition of the inducible degron already reduced the stability of cyclin A2 from 10 h in normal RPE-1 cells to 5 h in the engineered line (Fig. 3l). The cyclin A2 stability could be further reduced to 2 h by the addition of the DIA cocktail (doxycycline, indole-3-acetic acid and asunaprevir) (Supplementary Methods). Shortening the half-life of cyclin A2 and treating cells with a CDK4/6i led to more rapid cell cycle exit (Extended Data Fig. 10b,c), allowed a greater proportion of cells to exit the cell cycle in S/G2 phase (Extended Data Fig. 10d) and extended the period when cells were sensitive to CDK4/6 inhibition up to 3 h before they entered mitosis (Extended Data Fig. 10d–f).

Finally, we took advantage of natural variation between cell lines and plotted the cyclin A2 protein half-life as well as the cell cycle exit time for five different cell lines. We found a strong correlation between the cyclin A2 half-life and the cell cycle exit times (Fig. 3l). Notably, the best-fit line has a y intercept of less than 2 h, which is strikingly similar to the time it takes for cyclin A2 mRNA to be lost after CDK4/6 inhibition,

and a slope of approximately one, indicating that cyclin A2 stability is the primary contributor to the cell cycle exit clock.

## A feed-forward loop underlies CDK2 reversibility

Our data demonstrating that after loss of mitogen signalling, cyclin A2 transcription is repressed by p107/p130 several hours before any detectable change in CDK2 activity or Rb phosphorylation are at odds with the textbook model, which predicts that cyclin A2–CDK2 activity is self-sustaining due to a positive feedback loop (Fig. 4a), although is consistent with the presence of a dominant feed-forward loop (Fig. 4b). However, how this signalling architecture, as opposed to one with only a feedback loop, could generate the apparent irreversible cell cycle commitment commonly associated with the R-point phenomenon remains unclear. To address this, we used a mathematical model developed by Yao et al.[8], which demonstrates irreversibility in the proposed CDK2–Rb loop after mitogen removal (Fig. 4c) and adapted it to model the feed-forward pathway after mitogen removal (Fig. 4d). The steady-state levels of CDK2 generated by the feedback loop model remained consistently high over time after loss of mitogen signalling, inconsistent with our experimental observations (Fig. 4c,e). However, CDK2 activity generated by the feed-forward pathway remained high at the time when a cell would normally enter mitosis (Fig. 4d,e), but at steady state, corresponding to the median time to exit the cell cycle, CDK2 activity was low (Fig. 4d,e). Thus, the feed-forward signalling pathway matches our experimental observations that CDK2 activity is reversible in post-R cells. This lack of true irreversibility in CDK2 activation challenges the textbook model, which states that hysteresis in CDK2 activity with respect to mitogen signalling underlies irreversible cell cycle commitment at the R point. To test for hysteresis, we treated pre- and post-R cells with varying doses of MEKi and measured the fraction of cells with high CDK2 activity at different times after treatment. Demonstrating a lack of hysteresis in CDK2 activity with respect to mitogen signalling at the R point, we observed that both pre-R and post-R cells lost CDK2 activity at the same dose of MEKi if given enough time to reach steady state (Fig. 4e). These results agreed with the competing clock model, arguing that irreversible commitment to the cell cycle after loss of mitogen signalling has only been observed previously because a cell enters mitosis before the molecular signalling pathway driving CDK2 activity reaches a steady state that forces cell cycle exit. In other words, the R-point phenomenon is observed because in most cells, the half-life of cyclin A2 protein allows CDK2 activity to persist after loss of mitogen signalling for a longer time than is required for a cell to reach mitosis. This revised model of cell cycle commitment implies that there is no single point when cells irreversibly commit to proliferation that can be defined by a single molecular event, but it is rather determined by the cell's proximity to mitosis as well as the levels of cyclin A2 protein when mitogen signalling is lost (Fig. 4f).

## Discussion

Here, we used high-throughput single-cell imaging and tracking to study how the loss of mitogen signalling affected cell cycle commitment in post-R cells. We detected outlier cells whose behaviour contradicted the main prediction of the R-point model that all post-R cells will make it to mitosis and divide even in the absence of mitogen signalling (Fig. 1d). Notably, these outlier cells are found in numerous previous studies[3,4,10], but their contradictory behaviour has been left largely unexplained. In our attempt to understand why these post-R cells exited the cell cycle instead of dividing, we found that the temporal distance of any post-R cell from mitosis at the time of mitogen removal was predictive of whether it would exit the cell cycle or make it to mitosis and divide. Subsequently, by blocking mitosis in cells deprived of mitogen signalling, we found that CDK2 activity and Rb phosphorylation are reversible in all cells. We could observe this reversibility in CDK2 activity and cell

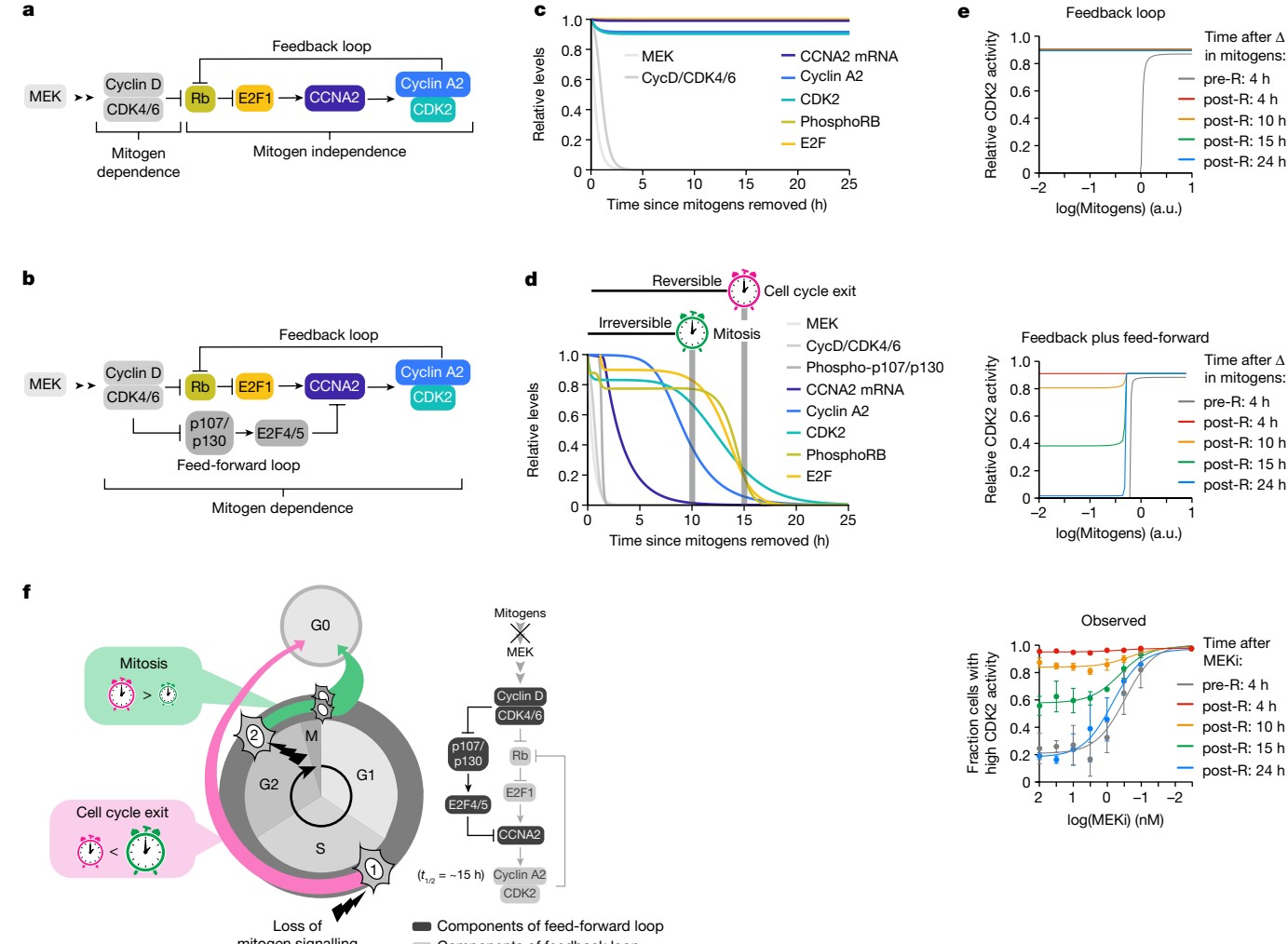

**Fig. 4 | A feed-forward loop underlies CDK2 reversibility. a**, Textbook model of the signalling pathway showing that cyclin A2 levels and CDK2 activity can be continuously maintained independent of mitogen signalling due to the proposed CDK2–Rb feedback loop. **b**, Revised model showing a feed-forward signalling pathway where mitogen signalling continuously maintains cyclin A2 and CDK2 activity in post-R cells. **c**, Output from mathematical modelling of the feedback loop model in **a** after mitogen removal. Cyclin A2 levels and CDK2 activity remain high. **d**, Output from mathematical modelling of the feed-forward pathway model in **b** after mitogen removal. Cyclin A2 levels and CDK2 activity appear irreversible at the time cells normally enter mitosis, although they eventually reach a steady state of zero, explaining why cells exit the cell cycle. **e**, CDK2 activity levels for pre- and post-R cells with respect to mitogen concentration for the feedback loop model (top panel) and the feed-forward pathway model (middle panel) at the indicated times after simulated change in mitogens. The observed (bottom panel) dose response relationship between CDK2 activity and MEKi concentration for pre- and post-R cells is shown. CDK2 activity was evaluated after MEKi treatment at the times indicated. Mitosis was blocked using a CDK1i. Error bars are s.e.m. from *n* = 2 replicates. **f**, For MCF-10A cells, in the absence of mitogens, competition between mitosis and cell cycle exit determines the fate of the cell due to a feed-forward loop regulating cyclin A2. A cell that is greater than approximately 15 h away from mitosis at the time mitogen signalling is blocked (cell 1) will lose cyclin A2 and exit, while a cell further along the cell cycle (cell 2) will reach mitosis before it will lose cyclin A2 and divide into two daughter cells.

cycle commitment because by preventing cells from undergoing mitosis, the molecular signalling pathway driving CDK2 activity was allowed sufficient time to reach steady state. Previous studies failed to observe this reversibility because cells were allowed to enter mitosis before CDK2 activity reached steady state, precluding the observation of cell cycle exit in post-R cells. Our study also uncovered an unexpected role for CDK4/6 in S and G2 phases by maintaining cyclin A2 transcription throughout the cell cycle, and it implies that small molecule CDK4/6 is may be effective in phases of the cell cycle beyond G1 phase, particularly if combined with traditional chemotherapies that extend the mitosis clock by inducing DNA damage and triggering cell cycle checkpoints.

Using genetic, biochemical and pharmacological approaches as well as mathematical modelling, we find that the signalling pathway connecting mitogen signalling with CDK2 activity is predominantly regulated by a feed-forward loop rather than containing a dominant

positive feedback loop. This signalling architecture pathway contains ultrasensitive signalling nodes, which make the system bistable and switch like, but the lack of a dominant positive feedback loop means the system is still reversible. We also cannot exclude the existence of additional positive feedback loops being involved in this signalling pathway since similar signalling systems, such as the one controlling the G2/M transition, have also been shown to contain positive feedback loops yet are still reversible[29,30]. Nevertheless, our data demonstrate that the dominant signalling architecture contains a feed-forward loop, resulting in a lack of hysteresis. Given that the two competing fates (mitosis versus cell cycle exit) are mutually exclusive, there is likely a double-negative feedback loop at the level of these cell fate decisions that ensures that once a cell exits, it makes it impossible to undergo mitosis and vice versa. Our findings provide a simple explanation for why the R-point phenomenon has been previously observed and led us

to a new model of cell cycle commitment that does not contain a single decision point. This model of temporal competition between mitosis and cell cycle exit can explain the behaviour of all cells, including the apparent contradictory behaviour of outlier cells observed here and in previous studies[3,4,10].

Our results show that mutually exclusive competing cell fates (mitosis and cell cycle exit) coupled with changes in cell state (either dividing or exiting the cell cycle) can function as irreversible fate transitions. More generally, temporal competition between any two mutually exclusive fates could be utilized by cells as a means of cellular decision-making in other contexts[31–33]: for example, deciding between proliferation or differentiation or deciding between mitosis and apoptosis. Thus, our findings may have implications for cellular decision-making outside of cell cycle regulation and may be a more general theme that underpins other cell decision-making.

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

## Reporting summary

Further information on research design is available in the Nature Portfolio Reporting Summary linked to this article.

## Data availability

Source data are provided with this paper. The datasets generated during and/or analysed during the current study are also available from the corresponding author on reasonable request. All data supporting the findings of this study are available from the corresponding author on reasonable request.

## Code availability

All original code has been deposited at https://github.com/scappell/Cell_tracking and is publicly available.

**Acknowledgements** We thank A. Lindqvist for the U2OS CycA2-eYFP cell line; H. Hochegger for the RPE-1 CCNA2[dd] cell line; C. Cataisson, B. Carofino and H. Al Khamici for technical support; and G. Altan-Bonnet and S. R. Collins for helpful discussions and critical reading of the manuscript. We thank the Flow Cytometry Core Facility of the Center for Cancer Research at the National Cancer Institute for technical support and all the members of the laboratory of S.D.C. for helpful comments and support. This research was supported by the European Molecular Biology Organization (ALTF 247-2022 to A.C.) and the National Institutes of Health Intramural Research Program (ZIA BC 011830 to S.D.C.).

**Author contributions** J.A.C. and S.D.C. conceived the project and designed all experiments. J.A.C., A.C., M.M.A., K.T., R.A. and S.D.C. performed all experiments and analysed data. J.A.C. performed all the live-cell imaging, RT-PCR and IF experiments and the Monte Carlo simulations. J.A.C. and S.D.C. analysed all of the live-cell imaging data and developed the mathematical model. K.T. and R.A. performed all of the siRNA validation western blots. A.C. performed all the immunoprecipitation experiments, and A.C. and K.T. performed phosphopocket protein time course western blots. M.M.A. performed the senescence-associated β-galactosidase staining and image analysis. J.A.C. and S.D.C. wrote the manuscript with help from A.C. and M.M.A. S.D.C. supervised and funded the project.

**Competing interests** The authors declare no competing interests.

## Additional information

**Correspondence and requests for materials** should be addressed to Steven D. Cappell.

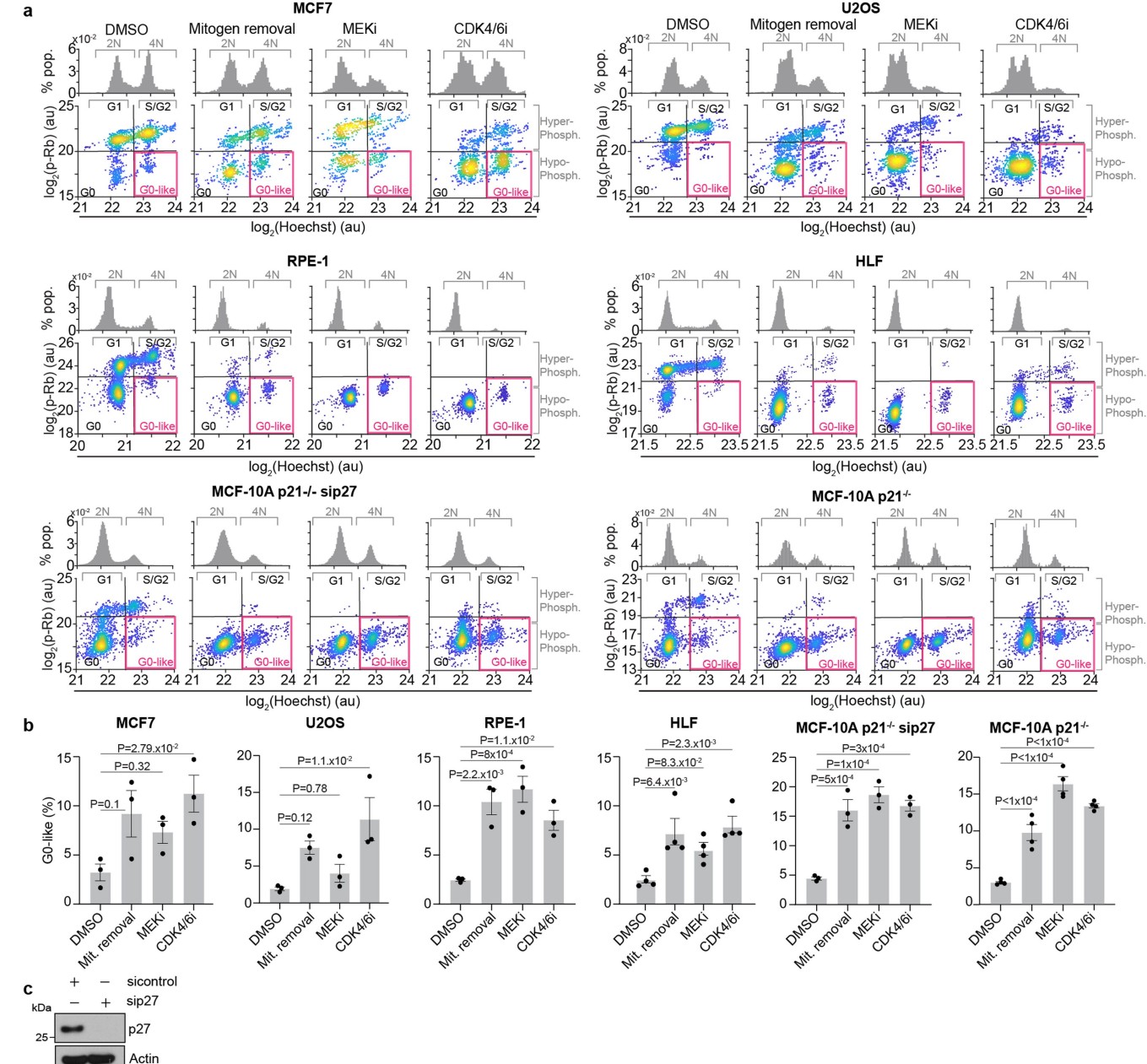

**Extended Data Fig. 1 | Diverse cell types rely on mitogens in S/G2. a,** Effect of loss of mitogen signalling on Rb phosphorylation and DNA content in diverse cell types: MCF7, transformed breast epithelial cells; U2OS, transformed osteosarcoma cells; RPE-1, non-transformed hTERT-immortalized retina pigmented epithelial cells; HLF, primary human lung fibroblasts; MCF-10A, non-transformed breast epithelial cells. Histograms show DNA content for each treatment (top). Scatter plots of Rb phosphorylation vs DNA content for each treatment (bottom). Pink boxes mark the G0-like state (hypo-phosphorylated Rb and 4N DNA content). All treatments were for 48 hrs. N = 2,000 cells are plotted per condition. **b,** The percentage of G0-like cells for each condition from (a) is indicated. Error bars are SEM from at least n = 3 biological replicates. P-values were calculated using a one-way ANOVA. P-values from top to bottom; MCF7: $2.79 \times 10^{-2}$, 0.32, 0.1, U2OS: $1.1 \times 10^{-2}$, 0.78, 0.12, RPE-1: $1.1 \times 10^{-2}$, $8 \times 10^{-4}$, $2.2 \times 10^{-3}$, HLF: $2.3 \times 10^{-3}$, $8.3 \times 10^{-2}$, $6.4 \times 10^{-3}$, MCF-10A p21$^{-/-}$ sip27: $3 \times 10^{-4}$, $1 \times 10^{-4}$, $5 \times 10^{-4}$, MCF-10A p21$^{-/-}$: $<1 \times 10^{-4}$, $<1 \times 10^{-4}$, $<1 \times 10^{-4}$. **c,** Western blot validation of p27 knockdown using siRNA. Representative image of n = 2 independent experiments.

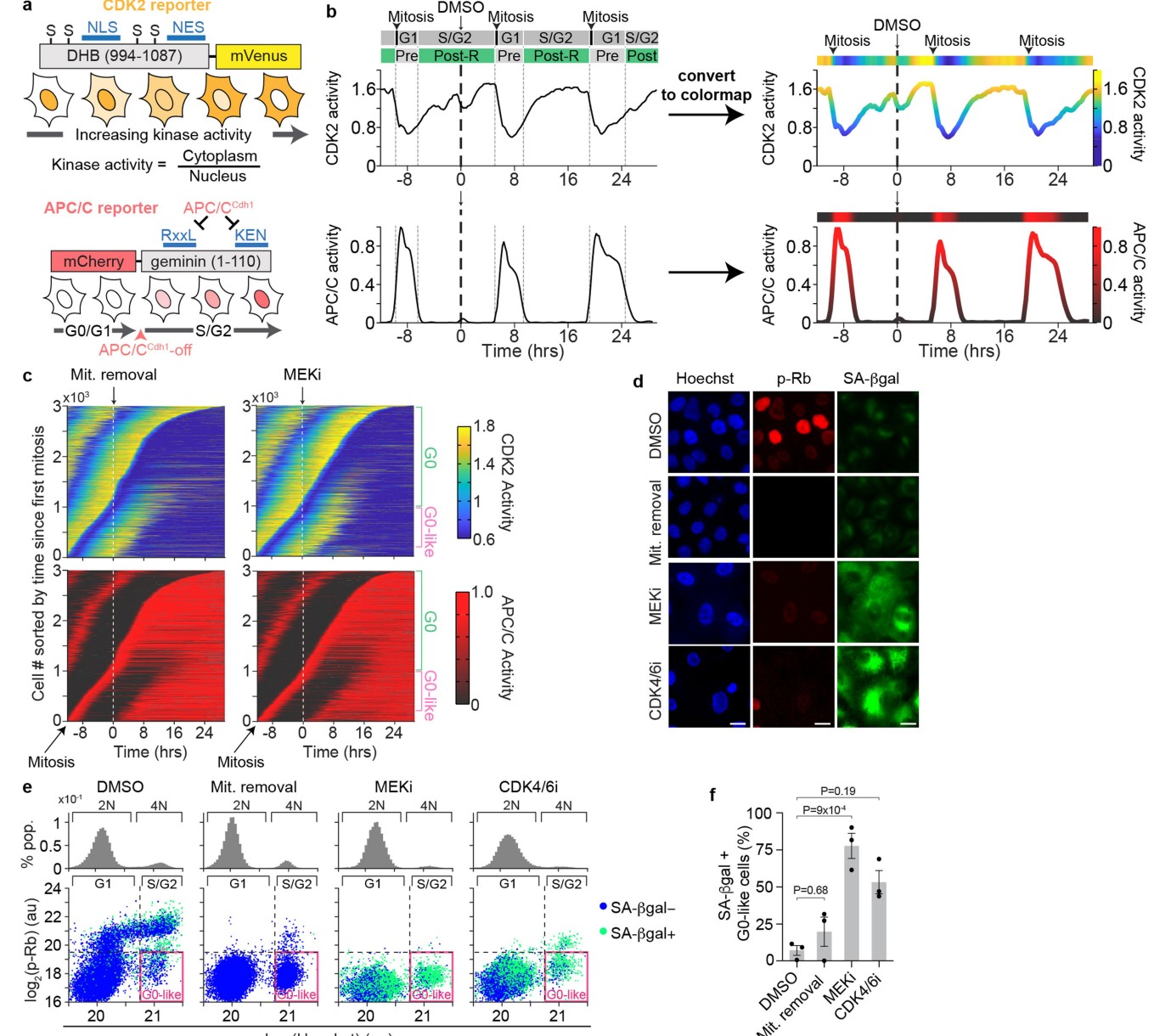

**Extended Data Fig. 2 | CDK2 and APC/C activity measured in live cells.**
**a**, Schematic of the fluorescent biosensors used for CDK2 activity (left) and APC/C activity (right). A detailed description of how the sensors work can be found in [4,15,16]. **b**, Example single-cell trace shows CDK2 (top) and APC/C (bottom) activity before and after DMSO treatment. Vertical grey lines indicate phase transitions and green bars show the region we considered cells to be post-Restriction Point (Post-R). Raw activity traces (left) are converted to a colormap format (right) to allow for analysing thousands of time series from asynchronous cells in one graph (see Figs. 1g and 2a and Extended Data Figs. 2c and 3a). **c**, Heatmaps showing CDK2 and APC/C activity aligned to mitosis for three thousands of cells treated as indicated. Mit, Mitogens. **d**, Representative fluorescent microscopy images of cells treated as indicated for 7 days and stained for DAPI to visualize nuclei, phospho-Rb, and a fluorescent probe to detect senescence-associated beta-galactosidase activity (SA-βgal; left). Scale bar is 20 μm. Images are representative cells from n = 3 independent experiments. **e**, Histograms show DNA content for each treatment (top). Scatter plots of Rb phosphorylation vs DNA content for each treatment (bottom). Each dot represents a single-cell and is coloured based on the status of SA-βgal staining. Pink boxes mark the G0-like state (hypho-phosphorylated Rb and 4N DNA content). Each plot contains N > 22,000 cells. **f**, Quantification of percent of cells that have high SA-βgal staining, 4N DNA content, and hypo-phosphorylated Rb from (e). Error bars represent SEM from n = 3 experiments. P-values were calculated using a one-way ANOVA. P-values from top to bottom: 0.19, 9 × 10⁻⁴, 0.68.

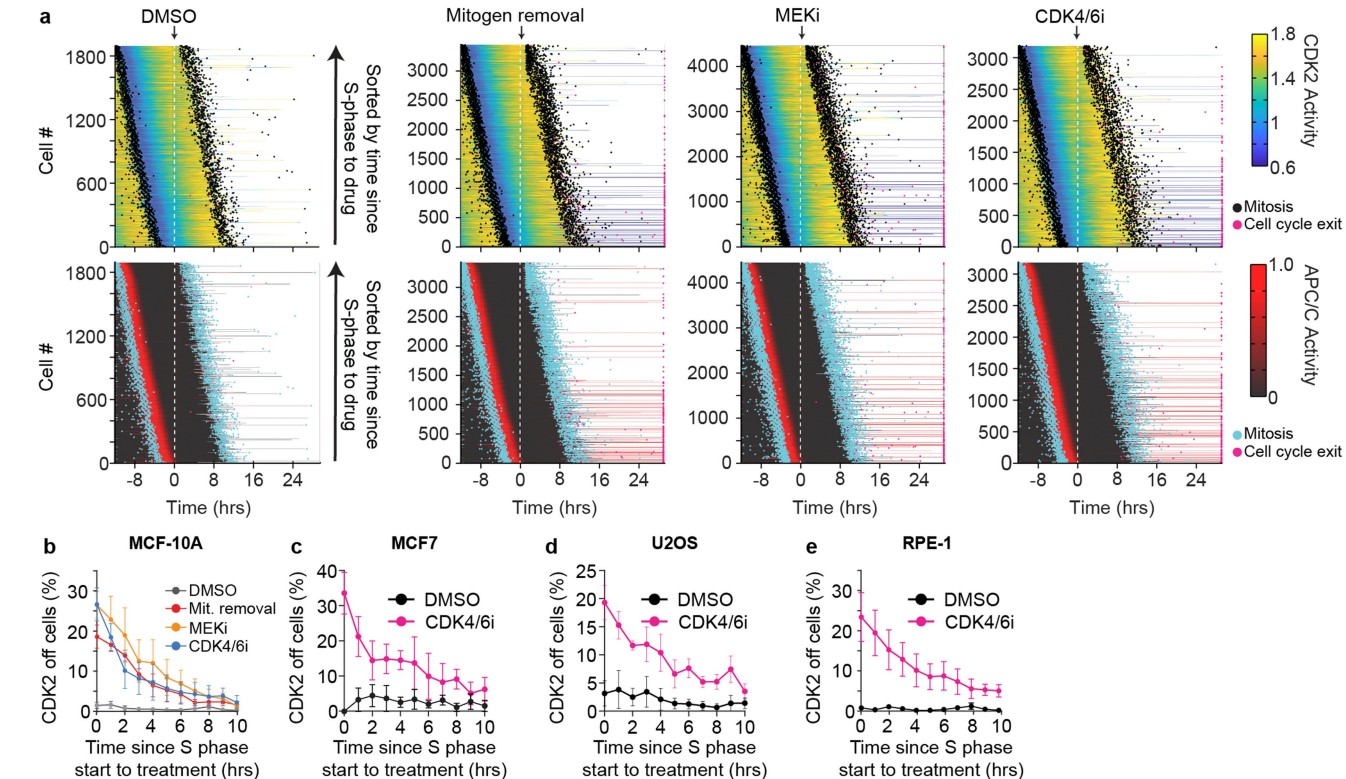

**Extended Data Fig. 3 | Proximity to S-phase start determines cell fate.**
**a**, Heatmaps of CDK2 activity (top) and APC/C activity (bottom) for post-R cells sorted by time since S-phase entry at treatment. When treated with DMSO all post-R cells divide within 24 hrs indicated by black/cyan dots. Upon mitogen removal or MEKi treatment some cells divide (black/cyan dots) and others lose CDK2 activity and exit the cell cycle (pink dots). **b**, Probability of cell cycle exit as a function of the time since the start of S phase at treatment for each treatment as indicated. Error bars represent SEM from n = 3 experiments. **c–e**, Probability of cell cycle exit as a function of the time since the start of S phase at treatment as indicated in MCF7 cells (c), U2OS cells (d), or RPE-1 cells (e). Error bars represent SEM from n = 3 experiments (c,d) or n = 5 experiments (e).

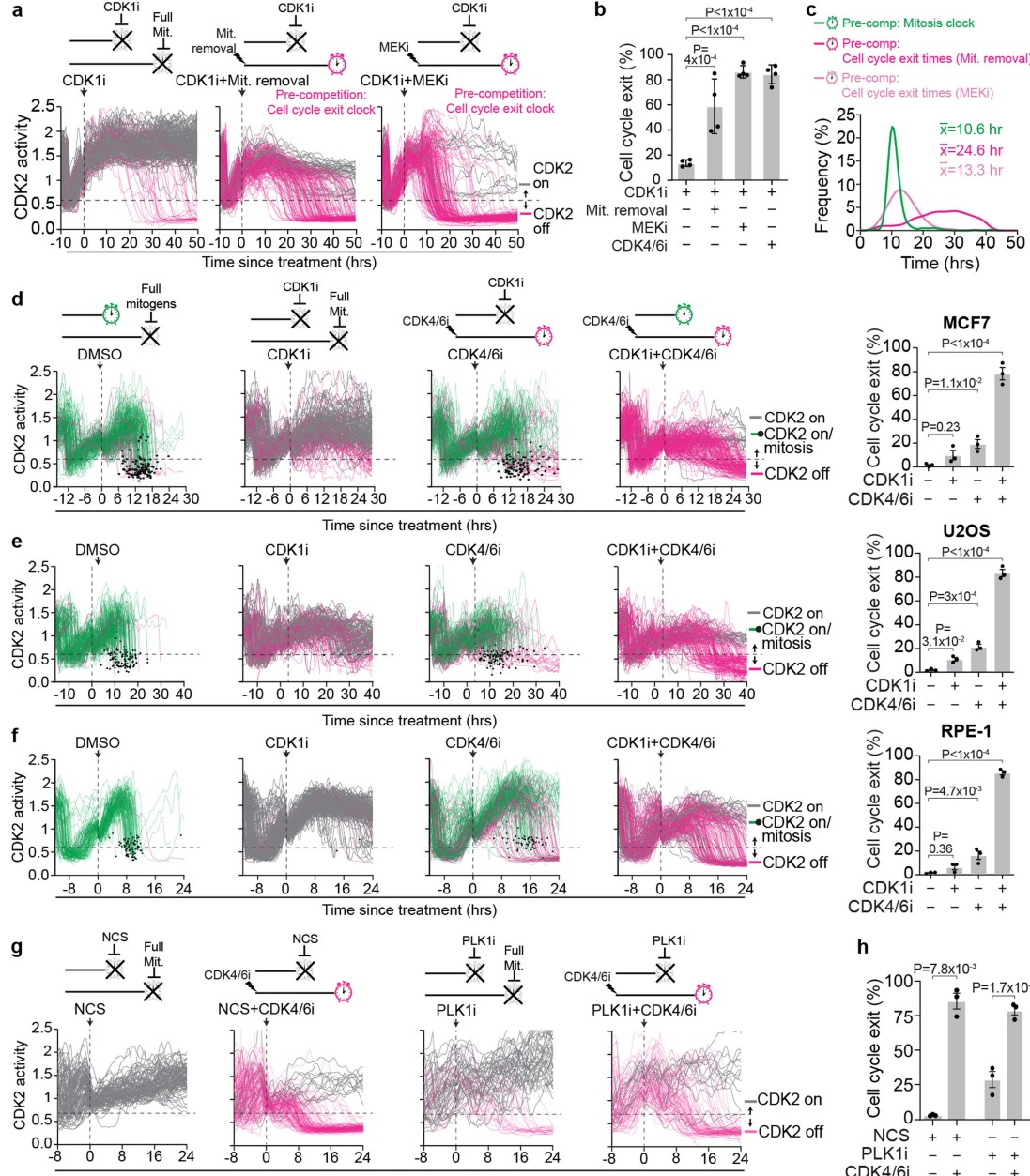

**Extended Data Fig. 4 | Cells require mitogen signalling if mitosis is blocked.**
**a**, Single cell traces of CDK2 activity aligned with respect to time of treatment for cells treated as indicated. Grey and pink traces represent cells with CDK2 > 0.6 and CDK2 < 0.6 at the end of the observation period, respectively. N = 200 cells per condition. **b**, Percentage of cells that exit the cell cycle after blocking the mitosis clock with a CDK1i and combining that with mitogen removal, MEKi, or CDK4/6i treatment. Error bars represent SEM n = 4 experiments. P-values were calculated using a one-way ANOVA. P-values from top to bottom: $<1 \times 10^{-4}$, $<1 \times 10^{-4}$, $4 \times 10^{-4}$. **c**, Histograms showing the pre-competition distribution of cell cycle exit times, measured from (a) and the pre-competition distribution of mitosis times, measured from Fig. 2e. **d**–**f**, Right, single cell traces of CDK2 activity aligned with respect to time of treatment for cells treated as indicated in MCF7 cells (d), U2OS cells (e), or RPE-1 cells (f). Green

traces depict cells that remained committed to the cell cycle and entered mitosis (indicated by black dot). Grey and pink traces represent cells with CDK2 > 0.6 and CDK2 < 0.6 at the end of the observation period, respectively. N = 200 cells per plot, with the exception of the DMSO condition in (f), which contains N = 117 cells. Left, quantification of the percent of cells exiting the cell cycle. Error bars represent SEM from n = 3 experiments. P-values were calculated using a one-way ANOVA. P-values from top to bottom; MCF7: $<1 \times 10^{-4}$, $1.1 \times 10^{-2}$, 0.23, U2OS: $<1 \times 10^{-4}$, $3 \times 10^{-4}$, $3.1 \times 10^{-2}$, RPE-1: $<1 \times 10^{-4}$, $4.7 \times 10^{-3}$, 0.36. **g**, CDK2 activity traces aligned to time of treatment for the indicated conditions in MCF-10A p21$^{-/-}$ cells. N = 99, 148, 67, and 85 cells respectively. **h**, Quantification of percent of cells that exit from (g). Error bars represent SEM from n = 3 experiments. P-values were calculated using a one-way ANOVA. P-values from top to bottom: $7.8 \times 10^{-3}$, $1.7 \times 10^{-3}$.

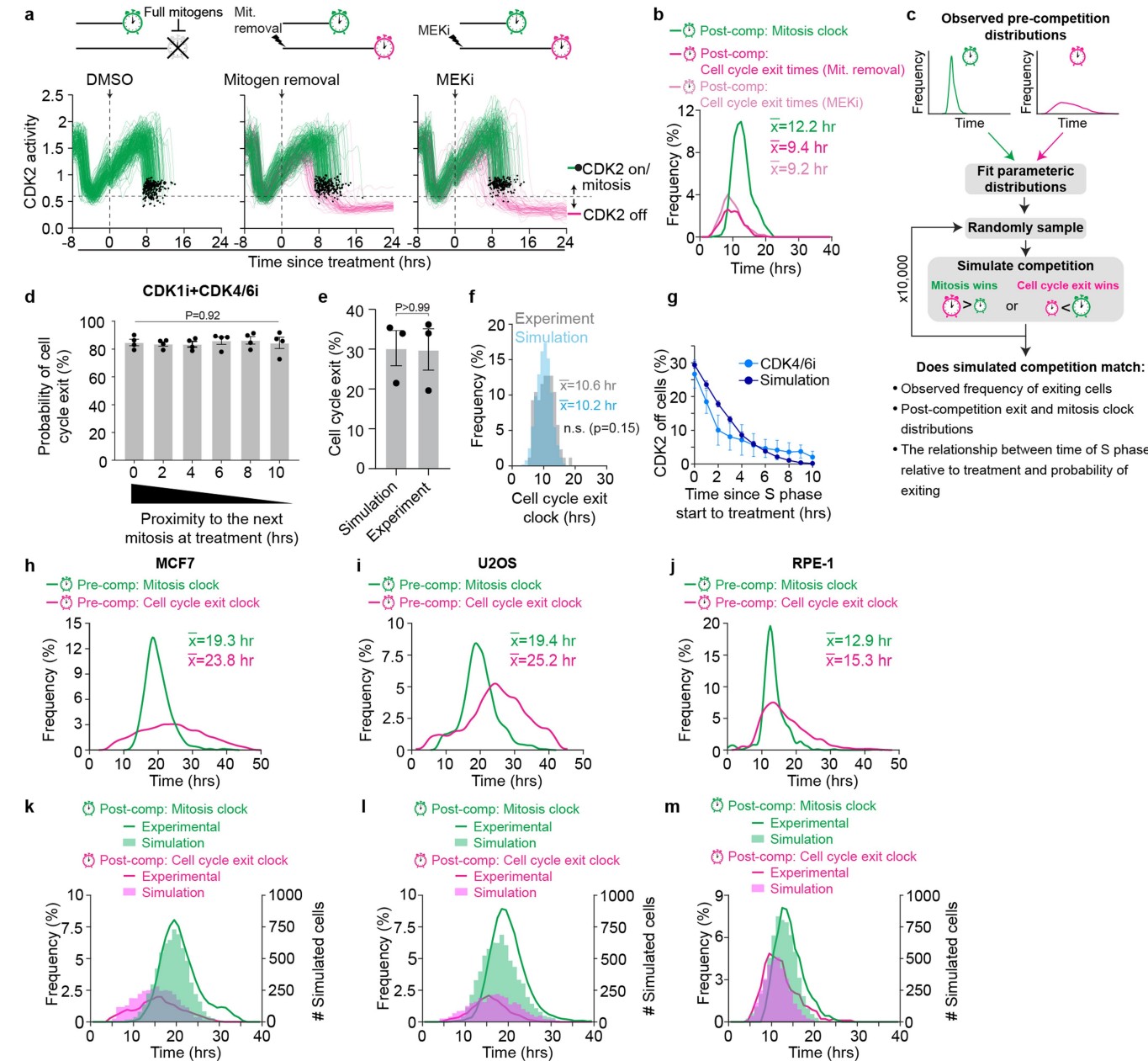

**Extended Data Fig. 5 | Simulated competition matches experimental data.**
**a**, Single cell traces of CDK2 activity for DMSO, mitogen removal, and MEKi treatment aligned to time of treatment. Green traces depict cells that remained committed to the cell cycle and entered mitosis (indicated by black dot). Pink traces depict cells that lost CDK2 activity (CDK2 < 0.6) and exited the cell cycle. N = 232, 282, and 252 cells respectively. **b**, Histograms showing the post-competition distribution of S/G2 length (mitosis clock) and the post-competition distribution of times to lose CDK2 activity (cell cycle exit clock) as measured from (a). Median times for each distribution are shown. **c**, Schematic outlining the Monte Carlo algorithm used to simulate temporal competition between the mitosis and cell cycle exit clocks. **d**, Probability of cell cycle exit as a function of time since start of S phase at treatment. Error bars represent SEM from n = 4 experiments. One-way ANOVA show no significant effect of time since S phase at treatment on the probability of exiting the cell cycle. P = 0.92. **e**, Comparison of observed frequency of cell cycle exit for cells which received the CDK4/6i within 1 hr of entering S phase with results from the

Monte Carlo simulation. Error bars represent SEM from n = 3 experiments. P-values were calculated using a two-tailed Mann-Whitney U test. P > 0.99. **f**, Comparison of observed distribution of cell cycle exit times for cells which received the CDK4/6i within 1 h of entering S phase with results from the Monte Carlo simulation. Median times for each distribution are indicated. A Wilcoxon Rank Sum test was used to test for statistical significance. Not significant (n.s.). **g**, Comparison of observed frequency of cell cycle exit as a function of time since the start of S phase at treatment with the results from the Monte Carlo simulation. Error bars represent SEM from n = 3 experiments. **h–j**, Histograms of the pre-competition times for cell cycle exit and mitosis from MCF7 (h), U2OS (i), and RPE-1 (j) cells. Data measured from single-cell data form Extended Data Fig. 4de,f. **k–m**, Histograms of the post-competition times for cell cycle exit and mitosis from MCF7 (k), U2OS (l), or RPE-1 (m) cells. Lines represent experimentally measured distributions and solid bars represent the simulated distributions from the Monte Carlo simulations.

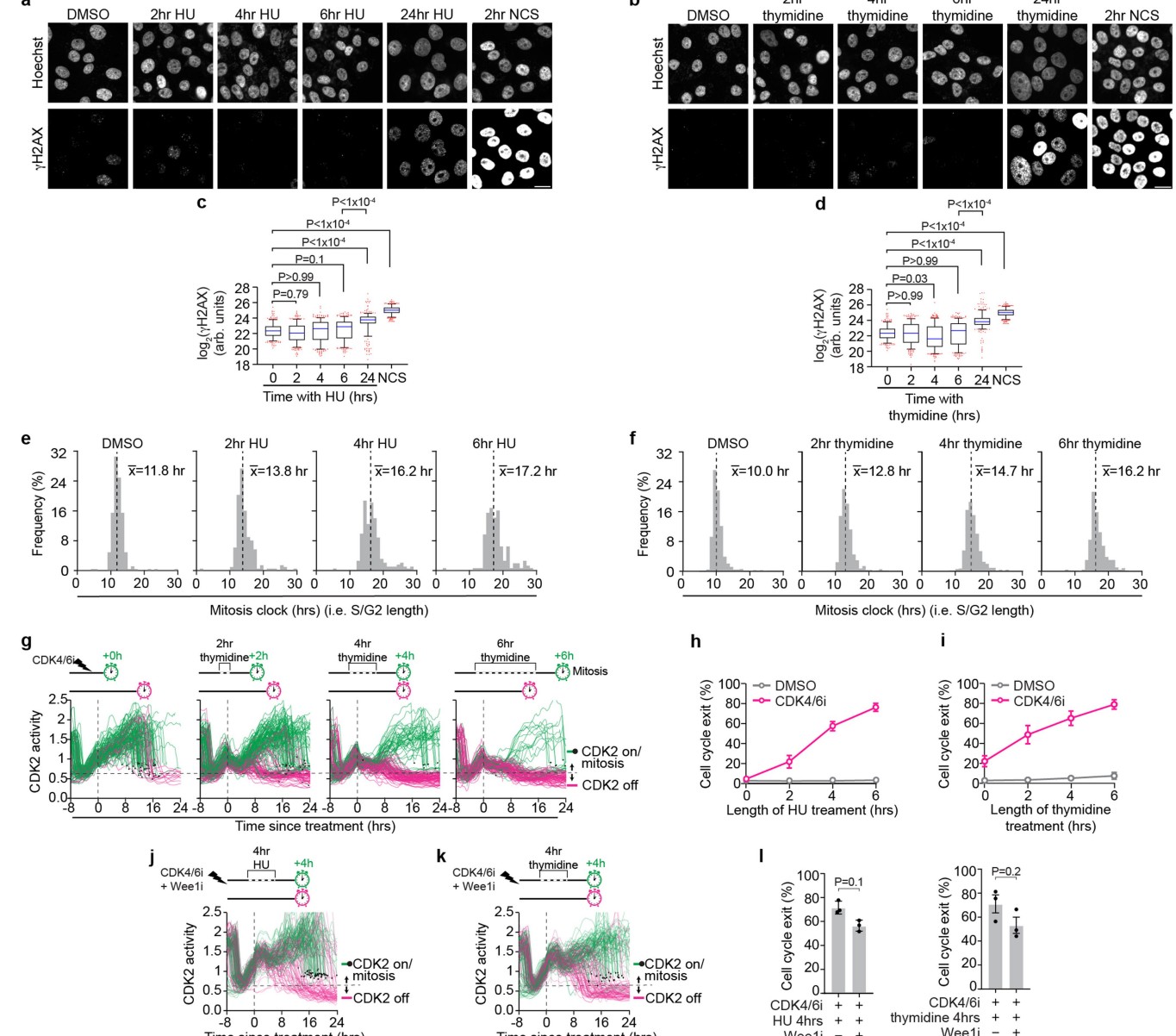

**Extended Data Fig. 6 | CDK4/6 inhibition induces exit when S/G2 is stretched. a,b,** γH2AX staining in MCF-10A p21[−/−] cells treated with either DMSO, hydroxyurea (HU), thymidine, or neocarzinostatin (NCS) for the indicated time. NCS treatment was used as a positive control. 20 μm scale bar. **c,d,** Box and whisker plot showing quantification of γH2AX nuclear intensity from (a,b). Blue line represents population mean. Whiskers represent 5–95[th] percentiles. Red dots indicate cells outside the 5–95[th] percentiles. One-way ANOVA and Kruskal-Wallace was used to test for statistical significance. P-values from top to bottom for c: <1×10[−4], <1×10[−4], <1×10[−4], 0.1, >0.99, 0.79. P-values from top to bottom for d: <1×10[−4], <1×10[−4], <1×10[−4], >0.99, 0.03, >0.99. N = 500 cells Per condition. **e,f,** Histograms showing S/G2 length of MCF-10A p21[−/−] cells after transient hydroxyurea (HU) or thymidine treatment. **g,** CDK2 activity traces from MCF-10A p21[−/−] cells aligned to time of treatment with thymidine and CDK4/6i. The length of increase in the mitosis clock is indicated above each plot. Green traces indicate cells which entered mitosis (black dots) and pink traces indicate cells which exited the cell cycle (CDK2 < 0.6). N = 129, 200, 157, or 119 cells in each condition. N = 130, 200, 158, and 120

cells respectively. **h,** Quantification of the percent of cells exiting the cell cycle after HU treatment from traces in Fig. 2j. Error bars represent SEM from n = 3 independent experiments. **i,** The percentage of MCF-10A p21[−/−] cells that exit the cell cycle after treatment with thymidine for various durations as a function of the increase in S/G2 length. Quantification from traces in (g). Error bars represent SEM from n = 3 independent experiments. **j,k,** CDK2 activity traces aligned to time of treatment with CDK4/6i and Wee1i plus a 4 h pulse of hydroxyurea (HU) or thymidine in MCF-10A p21[−/−] cells. Green traces indicate cells which entered mitosis (black dots) and pink traces indicate cells which exited the cell cycle (CDK2 < 0.6). Note that the Wee1i rescues the drop in CDK2 activity for both treatments (see Fig. 2j) but does not interfere with the ability of cells to exit the cell cycle. N = 200 cells. **l,** Quantification of the percent of cells that exited to the G0-like state after the indicated treatment. A two-tailed Mann-Whitney U test from n = 3 independent experiments shows no significant effect of Wee1 inhibitor on the ability of cells to exit to the G0-like state in response to a four hour pulse of either HU (left) or thymidine (right). P-values from left to right: 0.1, 0.2. Error bars represent SEM.

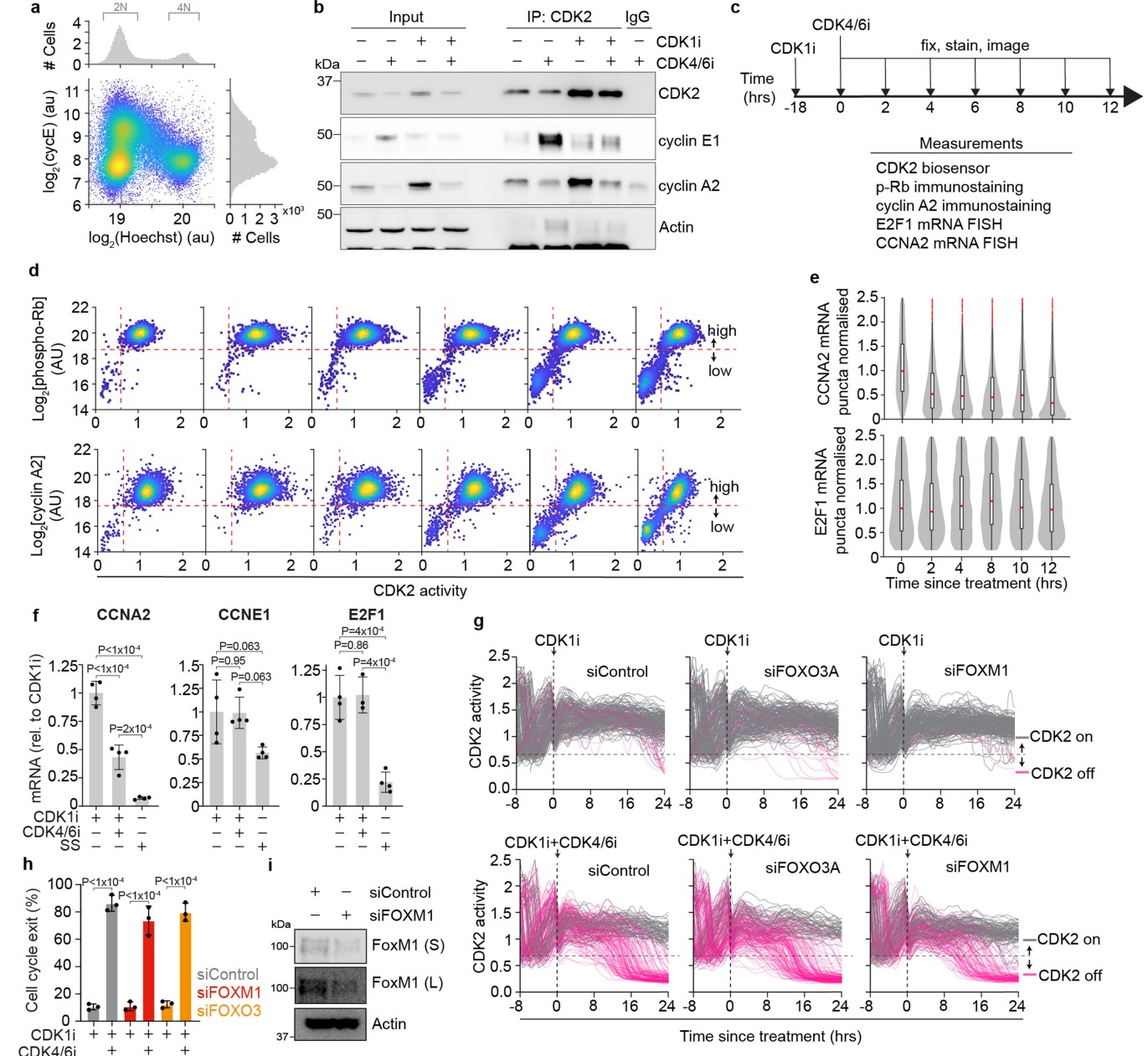

**Extended Data Fig. 7 | CDK4/6 activity maintains CCNA2 transcription.**
**a**, Asynchronous HeLa cells were fixed and then stained with a cyclin E1 antibody. Histograms show DNA content (top) and cyclin E1 levels (right). Density scatter plot of cyclin E1 levels vs DNA content (bottom). N = 63,031 cells are plotted. **b**, Coimmunoprecipitation of CDK2 in either asynchronous cells or cells treated with a CDK1i overnight and then treated with and without a CDK4/6i for 4 hrs. Representative of n = 2 experiments. **c**, Schematic outlining the experimental design used for Fig. 3b and Extended Data Fig. 7d,e. Cells were first pre-treated with a CDK1i to enrich for post-R cells, and then treated with a CDK4/6i before being fixed, stained, and imaged at the indicated timepoints. **d**, Density scatter plots of cyclin A2 levels (top row) and phospho-Rb levels (bottom row) plotted against CDK2 activity measured in single-cells. The vertical dashed red line indicates the threshold of CDK2 activity that is required to maintain Rb phosphorylation (CDK2 = 0.6). The horizontal dashed red lines separate cells with hypo- vs hyper-phosphorylated Rb (bottom row) and cells with high vs low cyclin A2 protein levels (top row). N > 1,700 Cells per plot. **e**, Violin and box plots showing single-cell levels of CCNA2 mRNA puncta (top row) and E2F1 mRNA puncta as measured by mRNA FISH. mRNA puncta levels were normalized to the

0 hr treatment. Top: N = 3850, 0 hr; 1579, 2 hr; 2252, 4 hr; 2212, 8 hr; 2316, 10 hr; 2405, 12 hr cells. Bottom: N = 2908, 0hr; 1814, 2 hr; 2306, 4 hr; 1843, 8 hr; 2030, 10 hr; 2098, 12 hr cells. The box plots show the median (centre line), interquartile range (box limits), and minimum and maximum values (whiskers). **f**, qRT-PCR data showing mRNA levels normalized to CDK1i treated after treatment with a CDK4/6i for 2 hrs, or 24 hrs of mitogen removal as a positive control. Error bars represent SD from N = 3 replicates. Representative of n = 3 experiments. P-values were calculated using a one-way ANOVA. P-values from top to bottom; CCNA2: $<1\times10^{-4}$, $<1\times10^{-4}$, $2\times10^{-4}$, CCNE1: 0.063, 0.95, 0.063, E2F1: $4\times10^{-4}$, 0.86, $4\times10^{-4}$. SS, serum starvation. **g**, Single-cell traces of CDK2 activity aligned to time of treatment for cells treated as indicated. Grey and pink traces represent cells with CDK2 > 0.6 and CDK2 < 0.6 at the end of the observation period, respectively. N = 200 Cells per plot. **h**, Quantification of the percent of cells that exit the cell cycle as in (g). Error bars represent SEM from n = 3 experiments. P-values were calculated using a one-way ANOVA. P-values from left to right: $<1\times10^{-4}$, $<1\times10^{-4}$, $<1\times10^{-4}$. **i**, Western blot validation of FoxM1 knockdown using siRNA. S, short exposure; L, long exposure. Representative of n = 2 experiments.

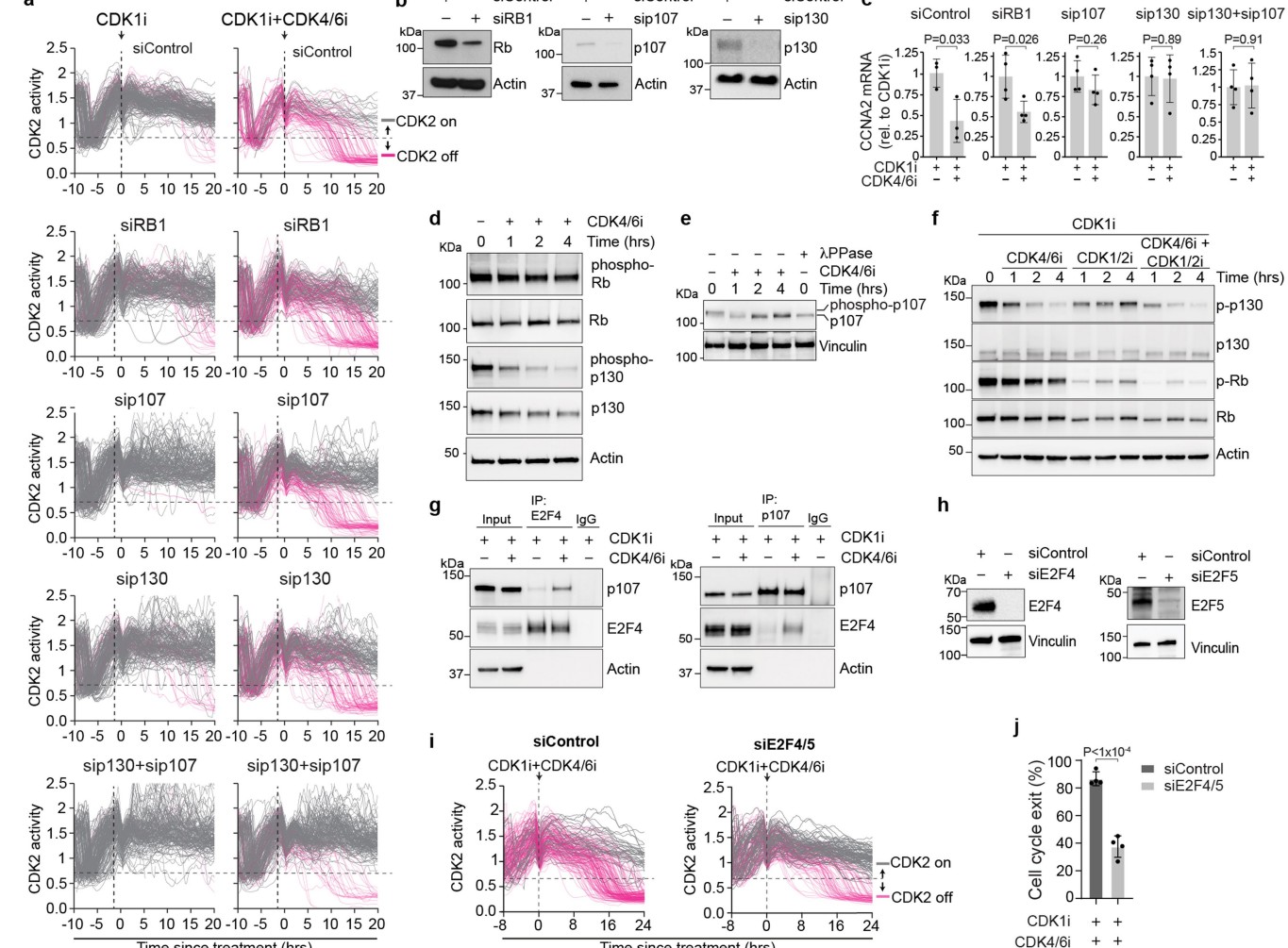

**Extended Data Fig. 8 | p107/p130 inhibit CDK2 activity after mitogen loss.**
**a**, Single-cell traces of CDK2 activity aligned to time of treatment for the indicated conditions. Grey and pink traces represent cells with CDK2 > 0.6 and CDK2 < 0.6 at the end of the observation period, respectively. N > 117 Cells per plot. **b**, Western blot validation of Rb, p107, and p130 knockdown using siRNA. Representative image of n = 2 experiments. **c**, qRT-PCR data showing mRNA levels normalized to CDK1i treatment alone for cells treated with a CDK4/6i for 2 hrs. Error bars represent SD from N = 4 replicates (N = 3 for siControl). Representative of n = 3 experiments. P-values were calculated using a one-way ANOVA. P-values from left to right: 0.033, 0.026, 0.26, 0.89, 0.91. **d**, Western blot time-course of phospho-p130 (S672) and phospho-Rb (S807/811) after CDK4/6i treatment. Cells were pre-treated with a CDK1i for 24 hrs to arrest cells in a post-R state and then treated with either DMSO or a CDK4/6i for the indicated times. Representative image of n = 5 experiments. **e**, Western blot time-course showing a loss of p107 phosphorylation after CDK4/6i treatment.

Arrows indicate the phosphorylated and unphosphorylated forms of p107. Representative image of n = 2 experiments. **f**, Western blot time-course in cells treated with a CDK1i and then either CDK4/6i alone, CDK1/2i Alone, or both. **g**, Coimmunoprecipitation of p107 and E2F4 in cells treated with a CDK1i overnight and then treated with and without a CDK4/6i for 4 hrs. Top: IP with E2F4 antibody. Bottom: IP with p107 antibody. Representative image of n = 2 experiments. **h**, Western blot validation of E2F4 and E2F5 knockdown using siRNA. Representative of n = 2 experiments. **i**, Single cell traces of CDK2 activity aligned to time of treatment for cells treated as indicated. Grey and pink traces represent cells with CDK2 > 0.6 and CDK2 < 0.6 at the end of the observation period, respectively. N = 197 and 167 cells respectively. **j**, Quantification of the percent of cells that exit the cell cycle as in (h). Error bars represent SEM from n = 4 experiments. P-values were calculated using a two-tailed Students T-test. $P < 1 \times 10^{-4}$.

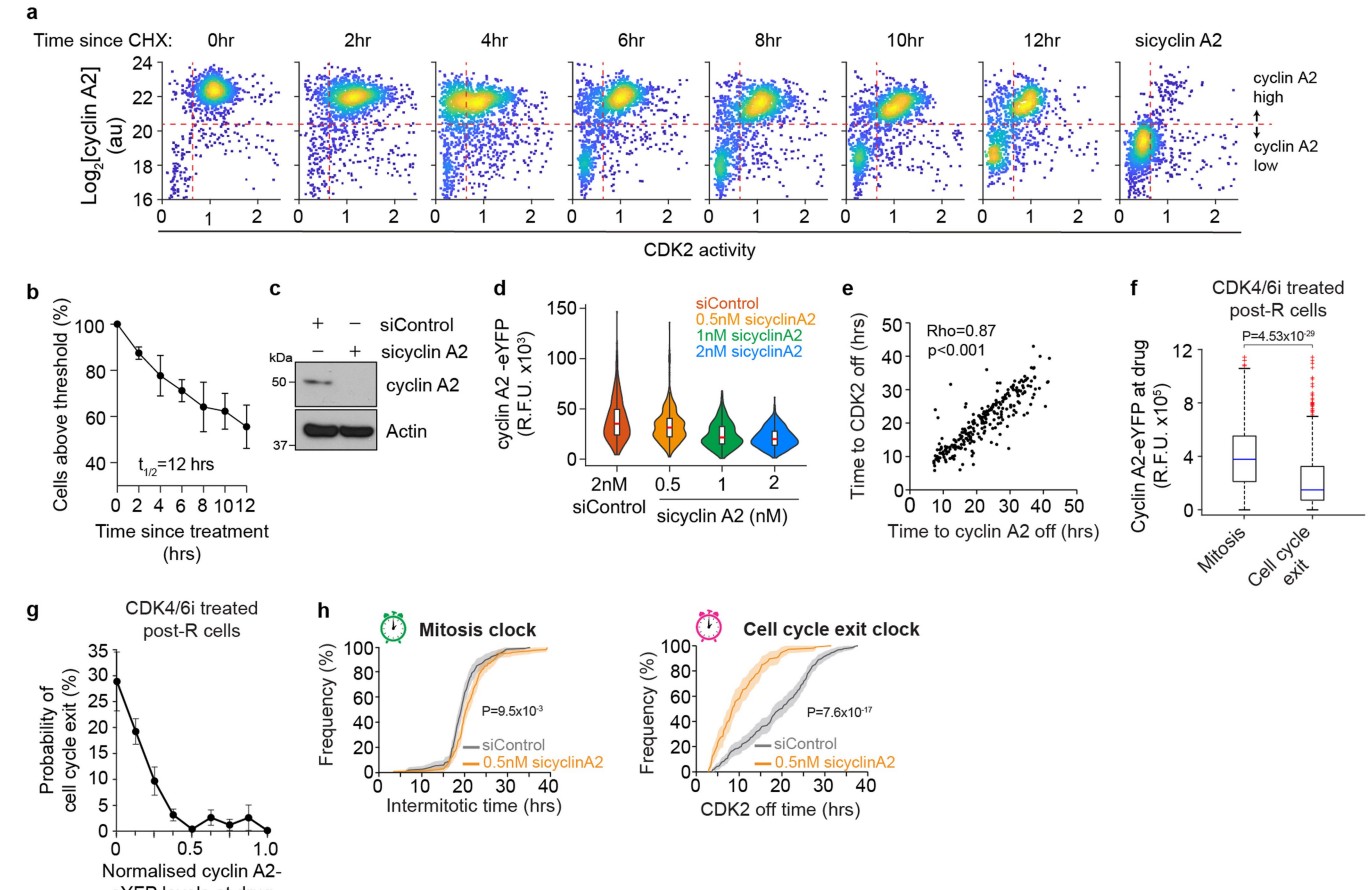

**Extended Data Fig. 9 | Cyclin A2 stability determines the cell cycle exit clock. a**, Density scatter plots of cyclin A2 levels against CDK2 activity after treatment with cycloheximide at indicated timepoints. The vertical dashed red line indicates the threshold of CDK2 activity that is required to maintain Rb phosphorylation (CDK2 = 0.6). The horizontal dashed red lines separate cells with high vs low cyclin A2 protein levels (top row). The last panel shows cyclin A2 levels for cells treated with cyclin A2 siRNA to establish the threshold for high vs low cyclin A2 levels. N > 1,300 Cells per plot. **b**, Plot of percent of cyclin A2 low cells over time after treatment with cycloheximide. Error bars represent SEM from n = 3 experiments. $t_{1/2}$ indicates the measured half-life. **c**, Western blot validation of cyclin A2 knockdown using siRNA. **d**, Violin and box plots showing single-cell cyclin A2-eYFP levels after treatment with various concentrations of siRNA targeting cyclin A2. Red line represents population mean. From left to right, N = 566, 479, 487, and 394 cells. The box plots show the median (centre line), interquartile range (box limits), and minimum and maximum values (whiskers). **e**, Scatter plot showing the correlation between the time to lose CDK2 activity (CDK2 < = 0.6) and cyclin A2 levels (cyclin A2 < = G1 levels of cyclin A2). Rho, pearson's correlation coefficient. N > 396 cells per condition. **f**, Data are a box and whisker plot of the cyclin A2-eYFP levels at the time post-R U2OS cells were treated with CDK4/6i in cells that either reached mitosis or exited the cell cycle. Blue line represents the population mean. Whiskers represent 5–95th percentiles. Red dots indicate single-cell outliers beyond the 5–95th percentiles. A Mann Whitney U test was used to test for statistical significance. P = 4.53 × 10$^{-29}$. N = 915 (mitosis) and 333 (cell cycle exit) cells. **g**, Probability of cell cycle exit as a function of the cyclin A2-eYFP levels at the time the CDK4/6i was added. Error bars represent SEM from n = 4 experiments. **h**, Empirical cumulative distribution of intermitotic times (left panel) and cell cycle exit times (right panel) for cells treated with control siRNA and 0.5 nM cyclin A2 siRNA. P-values were calculated using a two-tailed Kolmogorov Smirnov test. Mitosis clock (P = 9.5 × 10$^{-3}$) and cell cycle exit (P = 7.6 × 10$^{-17}$).

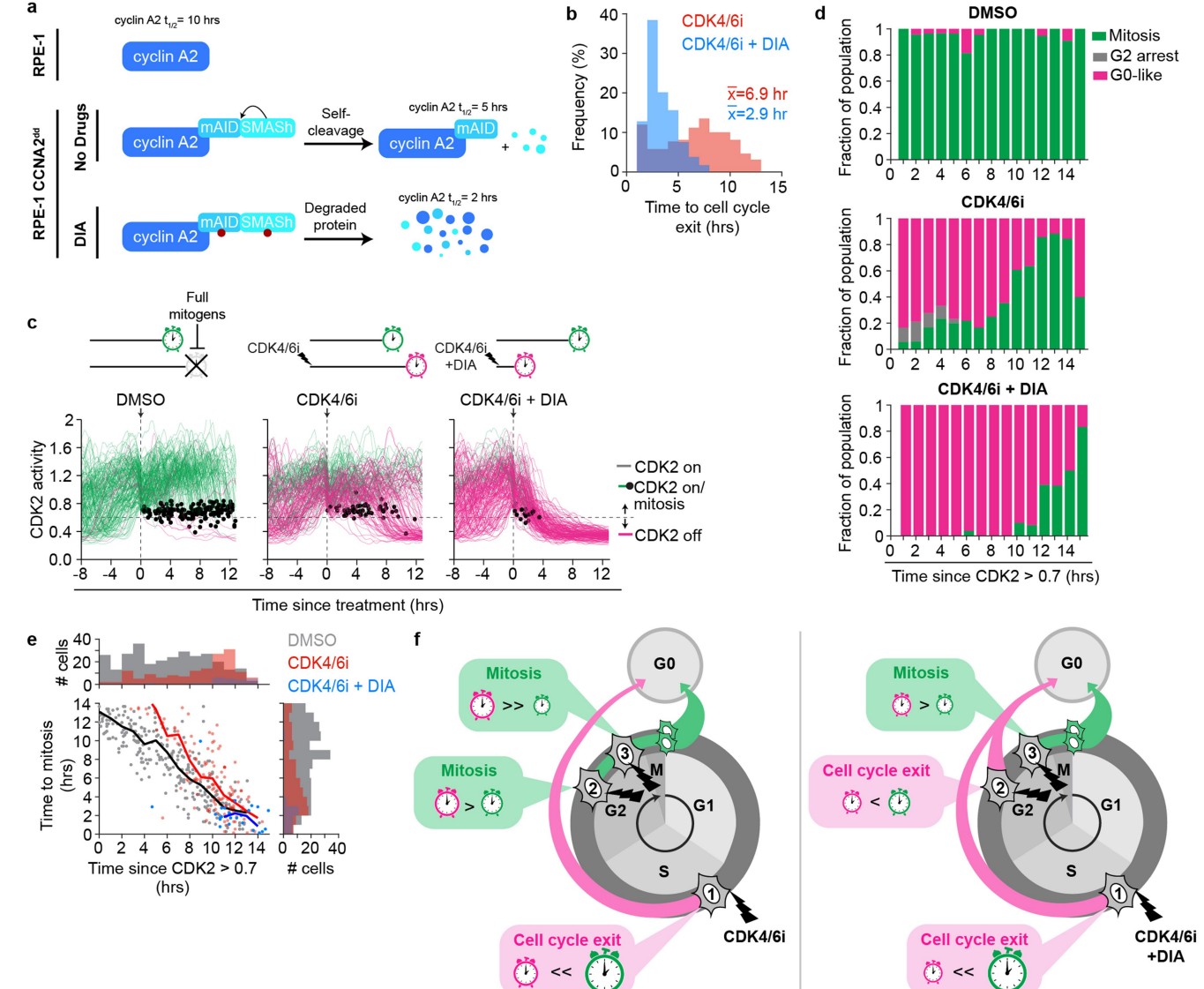

**Extended Data Fig. 10 | Destabilizing cyclin A2 shortens the cell cycle exit clock. a**, Schematic of the inducible degron system used to de-stabilize cyclin A2. For details about the inducible-degron system, see Methods section. **b**, Histogram of the cell cycle exit times for RPE-1 CCNA2dd cells treated with a CDK4/6i or CDK4/6i plus DIA. **c**, Single cell traces of CDK2 activity aligned with respect to time of treatment for cells treated as indicated in RPE-1 CCNA2dd cells. Green traces depict cells that remained committed to the cell cycle and entered mitosis (indicated by black dot). Grey and pink traces represent cells with CDK2 > 0.6 and CDK2 < 0.6 at the end of the observation period, respectively. For each condition, N = 200 cells are shown. **d**, Cell fate of RPE-1 CCNA2dd cells from (c) binned by the time since CDK2 activity rose above 0.7 when the drug was added. Representative data from n = 2 independent

experiments. **e**, Scatter plot of single cells comparing the time since CDK2 activity rose above 0.7 when the drug was added versus the time after treatment when cells reached mitosis. Single-component histograms shown above and to the right. Solid lines represent the average for each condition as indicated. DMSO, N = 280; CDK4/6i, N = 126; CDK4/6i + DIA, N = 26 cells. **f**, Left: Upon CDK4/6 inhibitor treatment, cells closer to the start of S phase, and therefore far away from mitosis (Cell 1), are more likely to exit the cell cycle, whereas cells close to mitosis (Cell 2 and 3) are more likely to reach mitosis. Right: Destabilizing cyclin A2 by treating cells with DIA shortens the cell cycle exit clock, which means that, relative to cells without DIA, cells closer to mitosis (Cell 2) when the CDK4/6 inhibitor is added are more likely to exit the cell cycle.

# Reporting Summary

## Statistics

For all statistical analyses, confirm that the following items are present in the figure legend, table legend, main text, or Methods section.

| n/a | Confirmed | |
|---|---|---|
| ☐ | ☒ | The exact sample size (*n*) for each experimental group/condition, given as a discrete number and unit of measurement |
| ☐ | ☒ | A statement on whether measurements were taken from distinct samples or whether the same sample was measured repeatedly |
| ☐ | ☒ | The statistical test(s) used AND whether they are one- or two-sided<br>*Only common tests should be described solely by name; describe more complex techniques in the Methods section.* |
| ☒ | ☐ | A description of all covariates tested |
| ☒ | ☐ | A description of any assumptions or corrections, such as tests of normality and adjustment for multiple comparisons |
| ☐ | ☒ | A full description of the statistical parameters including central tendency (e.g. means) or other basic estimates (e.g. regression coefficient) AND variation (e.g. standard deviation) or associated estimates of uncertainty (e.g. confidence intervals) |
| ☐ | ☒ | For null hypothesis testing, the test statistic (e.g. *F*, *t*, *r*) with confidence intervals, effect sizes, degrees of freedom and *P* value noted<br>*Give P values as exact values whenever suitable.* |
| ☒ | ☐ | For Bayesian analysis, information on the choice of priors and Markov chain Monte Carlo settings |
| ☒ | ☐ | For hierarchical and complex designs, identification of the appropriate level for tests and full reporting of outcomes |
| ☒ | ☐ | Estimates of effect sizes (e.g. Cohen's *d*, Pearson's *r*), indicating how they were calculated |

*Our web collection on statistics for biologists contains articles on many of the points above.*

## Software and code

Policy information about availability of computer code

| Data collection | NIS Elements (v5.11.00) |
|---|---|
| Data analysis | MATLAB (vR2020b). Automated image analysis was performed using custom MATLAB scripts as described in Cappell, S.D. et al Cell 166, 167-180 (2016) (https://github.com/scappell/Cell_tracking). Graphpad Prism 9 (v9.2.0) was used for statistical analysis. RStudio (v1.3.1093) was used for mathematical modeling along with the deSolve package. |

For manuscripts utilizing custom algorithms or software that are central to the research but not yet described in published literature, software must be made available to editors and reviewers. We strongly encourage code deposition in a community repository (e.g. GitHub). See the Nature Portfolio guidelines for submitting code & software for further information.

## Data

Policy information about availability of data

All manuscripts must include a data availability statement. This statement should provide the following information, where applicable:
- Accession codes, unique identifiers, or web links for publicly available datasets
- A description of any restrictions on data availability
- For clinical datasets or third party data, please ensure that the statement adheres to our policy

All data is available in the Source Data file. The datasets generated during and/or analyzed during the current study are also available from the corresponding author on reasonable request. All data supporting the findings of this study are available from the corresponding author on reasonable request.

## Human research participants

Policy information about studies involving human research participants and Sex and Gender in Research.

| | |
|---|---|
| Reporting on sex and gender | N/A |
| Population characteristics | N/A |
| Recruitment | N/A |
| Ethics oversight | N/A |

Note that full information on the approval of the study protocol must also be provided in the manuscript.

# Field-specific reporting

Please select the one below that is the best fit for your research. If you are not sure, read the appropriate sections before making your selection.

☒ Life sciences  ☐ Behavioural & social sciences  ☐ Ecological, evolutionary & environmental sciences

For a reference copy of the document with all sections, see nature.com/documents/nr-reporting-summary-flat.pdf

# Life sciences study design

All studies must disclose on these points even when the disclosure is negative.

| | |
|---|---|
| Sample size | Sample size was determined based on general standards for biological studies and requirements for statistical analysis, attempting to have a minimum of n=3 biological replicates with sufficient reproducibility. |
| Data exclusions | No data was excluded form the experiments. |
| Replication | All experiments in which p-values are present have been carried out with at least 3 replicates. All experiments were independently reproduced at least twice. |
| Randomization | Samples were allocated randomly for imaging and analysis. Representative single-cell traces where chosen at random from the population for visualization. |
| Blinding | Blinding was not relevant to this study. Image acquisition and analysis was conducted using automated scripts which are not subject to experimental bias. For western blots, blinding is not possible because samples need to be loaded in a particular order. |

# Reporting for specific materials, systems and methods

We require information from authors about some types of materials, experimental systems and methods used in many studies. Here, indicate whether each material, system or method listed is relevant to your study. If you are not sure if a list item applies to your research, read the appropriate section before selecting a response.

### Materials & experimental systems

| n/a | Involved in the study |
|---|---|
| ☐ | ☒ Antibodies |
| ☐ | ☒ Eukaryotic cell lines |
| ☒ | ☐ Palaeontology and archaeology |
| ☒ | ☐ Animals and other organisms |
| ☒ | ☐ Clinical data |
| ☒ | ☐ Dual use research of concern |

### Methods

| n/a | Involved in the study |
|---|---|
| ☒ | ☐ ChIP-seq |
| ☒ | ☐ Flow cytometry |
| ☒ | ☐ MRI-based neuroimaging |

## Antibodies

| | |
|---|---|
| Antibodies used | β-Actin (Abcam, ab6276, 1:10000)<br>CDK2 (CST, #18048, IP=1:100, IB=1:1000)<br>Cyclin A2 (Santa Cruz, Sc-271682, IF=1:500, IB=1:100)<br>cyclin E1 (IF; Santa Cruz, sc-247, IF=1:500, IB=1:1000) |

E2F4 (CST, 40291, IP=1:1000)
E2F4 (Thermo Scientific, MA5 11276, IB=1:1000)
E2F5 (Invitrogen, PA5-85578, 1:500)
FOXM1 (CST, 5436, 1:500)
p27 (CST, 3686, 1:500)
RB1 (CST, 9309, 1:500)
RBL1 (CST, 89798, IP: 1:1000; IB: 1:500)
RBL2 (CST, 13610, 1:1000)
Vincullin (Sigma, V9131, 1:1000)
phospho-Histone H2A.X (gammaH2AX, CST, #9718, 1:1000)
phospho-RB (Ser908/811) (CST, 9308, IB=1:1000)
phospho-RBL2 (phospho S672) (Abcam, Ab76255, 1:1000)
phospho-Rb (807/811) (Alexa Fluor 647 conjugate) (Cell Signalling Technology, #8974, IF=1:2000)
rabbit Igg (CST, 2729, 1:2000)
Goat anti-rabbit-HRP conjugated secondary (CST, 7074, 1:10000)
Horse anti-mouse-HRP conjugated secondary (CST, 7076, 1:10000)
Goat anti-mouse secondary antibody, Alexa Fluor 647 (Invitrogen, A-21241, 1:1000)
Goat anti-rabbit secondary antibody, Alexa Fluor 647 (Invitrogen, A-21245, 1:1000)

| Validation | All the antibodies used in this study are commercially available and extensively validated by the company, us, or others. Validation data is available in each of these company's website. In addition, we have confirmed the specificity of the following antibodies using siRNA-mediated knockdown and western blotting: p27 (ED Fig. 1c), FOXM1 (ED Fig. 7i), RB1 (ED Fig. 8b), p107 (RGBL1, ED Fig. 8b), p130 (RBL2, ED Fig. 8b), E2F4 (ED Fig. 8h), E2F5 (ED Fig. 8h), and cyclin A2 (ED Fig. 9c). All other antibodies were not directly validated by us but were validated by the manufacturer for the same species and application as they were used in this study. |

# Eukaryotic cell lines

Policy information about cell lines and Sex and Gender in Research

| Cell line source(s) | MCF10A (ATCC: CRL-10317)<br>RPE-1 (ATCC: CRL-4000)<br>U2OS (ATCC: HTB-96)<br>HeLa (ATCC: CRM-CCl-2)<br>HLF (ATCC: PCS-201-013)<br>HEK293T (gift from Dr. Tobias Meyer's Laboratory at Weil Cornell Medical School, ATCC: CRL3216)<br>MCF7 (gift from Dr. Jing Huang's Laboratory at the National Cancer Institute, ATCC: HTB-22)<br>U2OS CCNA2-eYFP (gift form Dr Arne Lindqvist's Laboratory at the Karolinska institute)<br>RPE-1 CCNA2-eYFP (gift form Dr Arne Lindqvist's Laboratory at the Karolinska institute)<br>RPE-1 CCNA2dd (gift from Dr. Helfrid Hochegger's Laboratory at Sussex University)<br>MCF10A p21-/- (gift from Tobias Meyer's Laboratory at Weil Cornell Medicine) |

| Authentication | Cell lines purchased from ATCC were not further authenticated. MCF7 cells were authenticated by short terminal repeat (STR) analysis peformed by the Huang Lab. HEK293T, USO2 CCNA2-eYFP, RPE-1 CCNA2-eYFP, RPE-1 CCNA2dd, and MCF10A p21-/- cell lines were not authenticated. |

| Mycoplasma contamination | Cells used in all experiments were routinely tested for mycoplasma contimination and only mycoplasma-negative cells were used in experiments |

| Commonly misidentified lines<br>(See ICLAC register) | No commonly misidentified cell lines were used in the study |

