## [Peer Review File · Nature]

Manuscript Title: Loss of CDK4/6 activity in S/G2 phase leads to cell cycle reversal

Reviewer Comments & Author Rebuttals

Reviewer Reports on the Initial Version:

Referees' comments:

Referee #1 (Remarks to the Author):

In this paper, Cornwell et al. proposed a revised Restriction Point model. In their model, rather than irreversibly committing to complete the cell cycle after R point, the cells will either complete mitosis or go to a G1-like state directly from the G2-phase, depending on whether they have enough CyclinA-CDK2 activity to drive them through G2 into mitosis. The authors showed that CDK4/6 activity is essential to maintain high Cyclin A levels in S/G2 phase by phosphorylating p107 and p130 pocket proteins, which pose an inhibitory effect on Cyclin A transcription when dephosphorylated. If the mitogen is removed or CDK4/6 activity is inhibited, Cyclin A transcription is suppressed by p107/p130 and Cyclin A protein levels start to drop. Depending at which point Cyclin A is inhibited, a cell may or may not have enough CDK2-CyclinA activity to sustain it through G2. If the levels of CDK2-CyclinA drop below a critical level then the cell ends up in a G1/0 state with 4N DNA content. This revised R point model explains the probabilistic behavior of cells following mitogen removal and also proposes an underlying molecular mechanism. It is therefore an important finding for the mammalian cell cycle field and suitable for a high-profile journal. However, given that the work is proposing to revise some long-held beliefs, and may be a landmark in the field, there are some experiments that require further corroboration or investigation. Below are my specific comments that should be addressed:

1. The positive feedback loop in the original R point model involves both Cyclins E and A. However, in this study, the authors only looked at Cyclin A. As a major part of the feedback loop, it is necessary to also investigate the dynamics of Cyclin E expression and Cdk2-CycE activity during the cell cycle and after mitogen removal. Especially given that that Cyclin E seems unaffected by CDK4/6i (Extended 5d), it is unclear why CDK2-CyclinE cannot maintain the cell cycle exit clock? Is it because Cyclin E is degraded in S?
2. From Figure 2e and I, it is clear that the CDK2 activity also dropped after CDK1i treatment and HU treatment. Since the CDK1i and HU were used to extend the S/G2 phase but not affecting CDK2, its effect on CDK2 activity complicates the interpretation. Can the authors use a more specific approach to extend S/G2 without profoundly affecting CDK2 activity? Possibly the CDK2as cells from Fisher could be used. In general, I think the CDK2 sensor is most likely reporting a mix of CDK2 and CDK1 activities (eg Schwarz et al 2018).
3. The authors showed that CDK4/6i affects p107/p130 phosphorylation in S/G2, which is not the case for RB, so that Cyclin A transcription is inhibited. For this mechanism to work, p107/p130 might need to be specifically phosphorylated by CDK4/6 but not CDK2 or CDK1. But the authors didn't provide enough evidence to support that, especially for the CDK2 phosphorylation on p107/p130. In addition, in Extended 6d, they showed a time course of phospho-p130 after CDK4/6 inhibition. However, even after 4 hours, there's still residual p-p130 that can be seen, and this time scale is far longer than a common phosphorylation turn-over. This actually suggests that CDK4/6 is not directly phosphorylating p130.
4. The authors propose a linear pathway model instead of a feedback model, however the observed phenomena do not necessarily contradict the existence of a feedback loop. The feedback loop can still exist on RB, but with an extra layer of regulation through CDK4/6-p107/p130. The reversible R point model can still have feedback loops. You have cell cycle transitions driven by

positive feedback that are reversible, even though they are bistable. The examination of mitosis in xenopus extracts are a good example of this (eg Sha et al PNAS 2003; Pomerening et al NCB 2003). The authors haven't disproven the existence of positive feedback as implied in their schematic.

5. In the revised R point model, two clocks are competing, and the cell cycle exit clock depends on Cyclin A protein level. The authors propose that once Cyclin A transcription is inhibited (via CDK4/6i or mitogen removal), the cell cycle exit timer is set by the Cyclin A protein half-life. Therefore, it will be a very strong support to the model if the authors could manipulate Cyclin A protein stability and show that the cell cycle exit clock length is sensitive to the half-life of Cyclin A. This is a difficult experiment, but would be very helpful in supporting the major point of the manuscript.

Referee #2 (Remarks to the Author):

This very interesting paper reexamines the mechanism underpinning the restriction point, traditionally defined as the time during G1 phase at which a cell irreversibly commits to carrying out a round of DNA replication and mitosis. The conventional view has been that double-negative feedback between Rb and CDK2 (via E2F) and then between CDK2 and Rb (via direct multisite phosphorylation) generates an irreversible bistable control circuit that accounts for the irreversible, all or none character of restriction point traversal.

One problem with this idea, and a key observation for this paper, is that although many S-phase cells will complete DNA replication and mitosis after washing out mitogens or applying MEK or CDK4/6 inhibitors, not all of them will, and this fact is true in all of the cell lines examined here, although the exact proportion of outlier cells depends somewhat on the treatment (washing out mitogens vs. adding an inhibitor) and the cell line chosen.

To account for the outliers, the authors have proposed an alternative hypothesis, which they call the competing clock model. If their hypothetical mitotic clock is maximally stimulated (with full growth medium) and presumably running maximally quickly, few of the cells fail to complete S- and M-phases because mitosis occurs before the cell cycle exit clock goes off, and the two fates (mitosis and exit) are mutually exclusive. On the other hand, if the mitotic clock is slowed by treatment with a CDK1 or PLK inhibitor or by triggering the DNA damage checkpoint (three ways of slowing or blocking CDK1 activation and mitotic entry), the cell cycle exit clock goes off in most of the cells and so a high fraction of the cells exit the cell cycle. Moreover, these cells lose their CDK2 activity before exiting the cell cycle, as reported by a human helicase B-based fluorescent biosensor, showing that under these circumstances, CDK2 activity is not infinitely self-sustaining.

Next they show through some nice quantitative experiments that if one assumes that normally the cell cycle exit clock is a little slower and a lot more variable from cell-to-cell than the mitotic clock is, and further assume that the clocks can be regulated independently, one can account for the proportion of cells that exit the cell cycle when S-phase has been extended by various durations of HU treatment. This is a satisfying test of the competing clock model.

Finally, they present evidence that continued CDK4/6-mediated inhibition of p107 and p130, rather than CDK2-mediated inhibition of Rb, is needed to keep S phase cells from exiting the cell cycle, and they show that an ODE model without the CDK2-Rb double negative feedback can account for the proportion of "outliers" present under various circumstances.

I think that this is a very strong paper, with the data in Figs 1 and 2 and the competing clocks concept being the principal strengths. I have some minor criticisms, but on the whole this is a fascinating new look at an important old problem.

Minor criticisms:

1. Line 17-18. Perhaps it would be more accurate to say "cyclin A transcription depends upon CDK4/6 activity throughout the cell cycle."
2. Line 11 and elsewhere: The protein name "cyclin" should not be capitalized (likewise for any protein name that is not an abbreviation or acronym—think actin, huntingtin, insulin, trypsin...)
3. In general the authors' whole "probabilistic vs. deterministic" theme is the one aspect of the paper that I do not think is well supported by the data, and which I think is not clearly thought through or explained. Really it is the fact that "outlier cells require mitogen signaling to maintain Rb phosphorylation [line 40]" at times way past when R is traditionally thought to have occurred, rather than the probabilistic character of R at the level of a population of cells (which could have arisen simply from cell-to-cell variability) that doesn't fit well with the classical view of R. In other words, these outlier cells are not mitogen-independent the way post-R cells are supposed to be. The authors need to do some re-writing to clean this up.
4. Fig 1a, c. The authors say that before R, you have CDK2-Rb off, and after R you have CDK2-Rb on. I know what they mean, but CDK2 and Rb are in a double-negative feedback relationship. Thus, before R you have Rb on (i.e. active as an E2F inhibitor) and CDK2 off, and after R you have Rb off and CDK2 on.
5. Lines 54-56. How about "we serum-starved MCF-10A cells or treated them with a MEK inhibitor or a CDK4/6 inhibitor...." Without the "or" that I added, this sentence could be read as meaning that the authors serum-starved the cells and treated them with one of the inhibitors.
6. The example cell time courses shown in insets in Fig 1e are too small to read.
7. Fig S1. It would be helpful to point out in the figure legend which of these cells are primary cells (I think it's just the HLFs), which of them are immortal but non-transformed, and which (if any) are transformed.
8. Line 94. For clarity, please change "closer to S-phase" to "closer to the start of S-phase" (assuming that's what you mean).
9. Line 112. This should be "blocking CDK4/6 signalling" rather than "blocking mitogen signalling".
10. Line 120. Likewise—"CDK4/6 signalling", not "mitogen signalling".
11. Fig 4b. The authors should point out that their "linear" model actually has a lot of non-linearity designed in; it just doesn't include the double-negative feedback loop between Rb and CDK2, and therefore does not yield bistability. Because of the non-linearity, the output of the model is still highly switchlike, but it is not hysteretic or irreversible.
12. Line 268. If the two cell fates—cell cycle exit vs. mitosis—are mutually exclusive, you could say there must be some sort of double negative feedback here, where cell cycle exit makes it hard or impossible for mitosis to occur, and mitotic entry makes it hard or impossible for cell cycle exit to occur. If so then there may well still be bistability involved in the process, albeit at the level of the mutually exclusive mitotic vs. cell cycle exit states rather than mutually exclusive CDK2-on Rb-off vs. CDK2-off Rb-on states. This is probably worth pointing out in the discussion section.
13. Also worth pointing out in the Discussion: nothing presented here proves that the CDK2-Rb circuit is not bistable. The authors do show that CDK2 activation is not irreversible, but a bistable circuit might yield hysteretic but not irreversible CDK2 activation (just as the mitotic control circuit yields hysteretic but not irreversible CDK1 activation). The important point to me, though, is that

one does not need to assume that there is bistability to account for any of the experimental results presented here—the authors' competing clocks hypothesis, and the linear or no-feedback mathematical model, suffice (as long as high degrees of ultrasensitivity are included).

Referee #3 (Remarks to the Author):

This work aims to determine how cells commit to proliferation, starting from an observation that, when mitogen signalling is disrupted, there is often a small fraction of cells (~5-15%) in a cell population that exit the cell cycle in G2 rather than completing a whole round of division by completing mitosis. This observation calls into question the irreversible nature of the Restriction Point (R) i.e. that once cells have passed R, they will complete mitosis, even in the absence of ongoing mitogenic signalling. The authors then combine single-cell imaging, perturbation, molecular biology and modelling approaches to show that there are two competing fates for cells that have passed R when mitogens are withdrawn and that the choice between these two fates relies on the stability of Cyclin A protein which sets a timer for mitosis. If Cyclin A protein is maintained then cells complete mitosis, if not, cells exit the cell cycle in G2. This property seems to be unique to Cyclin A and the authors show how Cyclin A transcription is regulated by CDK4/6 signalling to p107/p130-repressive E2F complexes. This is a novel and unexpected result that Cyclin A transcription is regulated by release of repressive transcription factors, rather than a positive feedback loop between CycA/CDK2 and E2F1, as has been widely believed. Finally, they show combining their findings could lead to a new combination of CDK4/6i and doxorubicin to drive cell cycle exit from G2 and senescence in cancer cells.

This paper was a pleasure to read and, while I am unsure about the analogy they have used with the two competing clocks (see comments below), the data themselves are novel and exciting. The discovery that CCNA2 transcription is not controlled by positive feedback by E2F1 and the proposal of a linear model is unexpected and important. This finding also calls into question other aspects of our current understanding of the system. The idea that passing R does not represent irreversible commitment to cell division is not necessarily a new one but the idea of a timer determining cell fate (mitosis versus cell cycle exit) is convincing. The inclusion of cartoons to explain some quite tricky concepts was also very welcome. I have a few comments and question below that I hope will improve the manuscript. I am unqualified to comment on the modelling aspects and other reviewers will be better placed to assess that part of the work. I am not convinced about the logic behind the combined use of doxorubicin and CDK4/6i in the clinic, for reasons outlined below (but this shouldn't distract from the main findings of the paper).

Major comments:

1. Testing the mitotic timing model using hydroxyurea. The authors use hydroxyurea to delay the mitotic clock (Figure 2, Extended Data Figure 4). However, hydroxyurea will cause replication stress during S-phase (which is why S-phase gets longer). This will activate p53 and so G2 gets longer because of p21 accumulation. p21 will then suppress CDK2 activity in G2 and lead to cell cycle exit. The longer the HU treatment, the more damage, more p21 accumulation and more cells undergo cell cycle exit. Therefore, I don't think this data helps to make their point about the two clocks and the role of Cyclin A as this is inducing a whole different mechanism to drive cell cycle exit. If they could show the same result in p21KO cells then this would be more convincing, but I suspect they will not see the same phenotype. Are there other ways to extend cell cycle length such as reducing the concentration of growth factors?
2. Logic of combining doxorubicin and CDK4/6i in cancer treatment. I can see why they have proposed this based on their data, and the results shown are convincing. However, my big problem here is that CDK4/6i seem to have a unique sensitivity towards transformed cells (for reasons people are only just starting to understand). However, the treatment regime proposed here would lead to the loss of that cancer cell specificity since it would also cause non-transformed cells to exit the cell cycle in G2 and become senescent, which could lead to adverse side-effects. Although they

don't show this here, I predict that if they performed the experiments shown in Extended Figure 8 in non-transformed cells (e.g. MCF10A) they would see the same results as with U2OS. I would recommend taking this part of the work out of the paper. The rest of the work stands well on its own.

3. The analogy of having two competing clocks. I don't find the analogy of having two competing clocks helpful (and I also worry that it will annoy some in the cell cycle field). In reality, is this not a timer? If cells get to mitosis before the timer (Cyclin A protein) runs out, then cells divide and complete the cycle. If not, they exit the cycle from G2. The authors may point to Fig 3I to argue that there are two distinct clocks but I am not sure it does show that? Do all cells in that experiment have a reduction in Cyclin A protein (i.e. homogenous depletion). Or are cells that have slightly less CycA already (the ones that would be most likely to exit in G2 in normal conditions) likely to be affected more so CDK2 activity is lost faster?

4. Cyclin A protein being key to the timer. One experiment that I think is missing to show clearly that the level of Cyclin A protein determines the fate between mitosis and cell cycle exit would be to use the CycA2-mVenus line they have and treat with CDK4/6i and see how the level of CycA2 is related to cell fate. According to the model, cells with less Cyclin A protein at the end of S-phase should have a higher probability of exiting the cell cycle than those with a higher level of CycA protein.

Minor comments:

1. It is known that CycA/CDK2 also shuts down activator E2F transcription (this is also mentioned in the text). However, this interaction is missing from Figure 3A. Have the authors considered including this in their ODE model to see how it would change CDK2 activity?

2. In Extended Figure 2c, how long after treatment was SA B-gal assayed?

3. Extended Figure 6D, can the authors quantify the reduction in phospho-p130 relative to the total amount of p130? This is important as the total amount of p130 protein appears to decrease after CDK4/6i treatment, which has been described for pRb (Dang et al., Nat Comms 2021).

4. Figure 4f – it wasn't clear how these experiments were performed and the timing determined? A better description would be appreciated.

5. In the IP experiments – how long was CDK4/6i added to cells before the IP?

6. Why is SEM used in some figures and SD in others?

7. Typos: Extended Figure 2 – should be a and not c in the legend. Extended Figure 3e – typo on graph – should be removal or removed.

Referee #4 (Remarks to the Author):

The study by Cornwell and colleagues, on the temporal competition between opposing fates underlying cell cycle fate choices, addresses the commitment between cell cycle exit or proliferation. It has been extensively described, and more or less taken for granted, from work performed in many different model systems that the Restriction (R) point bifurcates the dependency on proliferation signals so that pre-R cells read the presence/absence of mitogens to commit to proliferation or cell cycle exit. In the absence of mitogens, pre-R cells exit the cell cycle, while once the R point has been bypassed, cell cycle exit can no longer occur until the next G1 and so cells will finish the cell cycle with cell division.

In this study, the authors decided to interrogate if this long standing assumption holds always true. They cultured MCF-10 cells in the absence of serum or inhibited CDK4/6 or MEK and found that cells can exit the cell cycle even before mitosis. They propose that cells somehow read the activity of CDK2-CyclinA, informing on the time elapsed between S-phase and mitosis. In other words, they propose that there is no irreversible commitment to proliferation, but rather a higher probability of enter mitosis as cells approach M phase.

Understanding cell cycle regulation and the underlying principles that govern transitions from cell cycle phases in order to complete it or exit the cell cycle are of the uttermost importance in

biology. They influence developmental decisions and have the potential of helping designing therapies related with cell cycle dysfunction. This work is interesting and raises novel questions in the cell cycle field. However, even if quite relevant for the cell cycle field, I am not sure that it delivers a straightforward message that explains their findings. Moreover, the relevance of their findings, is also difficult to assess. They use many different drugs that may have a certain effect beyond the one being described here. Further, certain findings may be interpreted otherwise, and so in the end is difficult to assess the novelty and relevance of their findings. The main findings of this study can be summarized by the explanation that the R-point has been described due to the half-life of cyclin A2 allowing sufficient CDK2 activity even in the absence of proliferation signals. Even if interesting and appealing, I am not sure that this represents such a breakthrough in the field to be published in this journal.

Major points:

1) The authors very frequently mentioned that their work is against the text book work and reveals novel findings related with cell cycle exit before mitosis. In Fig 1d, they show that Mitogen removal, Meki, CDK4/6i, result in the appearance of a novel cell population -G0-like with hypo-phospho-Rb. But the frequency of these cells ranges from 8.7% to 14.6%. Which in my opinion is very minor, even if worth studying. But the main point here is that, the large majority of cells behaves as expected, no? So, while these results are interesting, I think they show that the text book view is corrected, no? At least for MCF-10A cells.

2) Still related with this point, I do not understand the choice of cells and certain approaches. The choice of MCF-10A cells and 48hrs of treatment. How long is a cell cycle in MCF-10A cells? The authors also use other cell lines (see below) shown in Supp Fig 1a. Take for example RPE1 cells, which have a cell cycle of around 22hrs. 48hrs of treatment appears quite long, as it will expose cells to a long time in the presence of an inhibitor or without serum. Could this lead to certain artifacts and so to the observations described here? It is even surprising that cells can survive this long without serum, no? Maybe an alternative approach would be to film these cells expressing the FUCCI system, so that they can follow individual cell behaviour from a known starting point. And like this, expose less time the cells to these drugs. This approach also has the advantage of providing information on how long will it take to get to the G0-like status.

3) The authors show in Sup Fig 1 that many other cells also respond in this way- like MCF-7, U2OS, RPE-1, HLF. The frequencies of G0-like is always low- most of the times below 10%. Now these cells are quite different in terms of their origin – cancer cells (U2OS), diploid and stable (RPE-1). How to integrate the fact that they basically respond in the same way with the G0 like population at levels of 10% or below?

4) But more importantly, they behave very differently according to their FACS profiles in other cell cycle phases. If one looks at Supp Fig 1a, almost each cell line, behaves differently after each treatment in terms of the distribution of cell populations. For example, RPE1 cells have almost no cells in G1 or S/G2 in MeKi or CDK4/6i, while this is not the case for MCF-7 or U2OS. Does this suggest that the R point is already different for all of them? Would this mean that even the way these cells transit from one phase to the other is quite unique and particular for a given cell line and so the principles described for one (MCF10), may not apply for another cell line? This would have consequences on the generalisation of the mechanism put forward related with Cyc A expression and the balance between closer to S or M -phases. This should be tested in all the cells to be certain that even cells with different R points may follow this rule.

5) The authors propose that cell cycle exit vs mitotic entry maybe be decided over a kind of competition between two fates. In other words, that every cell after S-phase has the capacity to exit or enter G2 and M-phase. What contributes to the cell fate choice then is proposed to be

related with how long cells have to complete the cell cycle. ` Thus, the authors extend "the time from S-phase to mitosis" by treating cells with HU. HU induces DNA damage and so I am not sure this is the right tool. Treatment with HU will in principle block DNA synthesis and so what the authors are measuring is the time required to finalize DNA replication and the strength of the checkpoints. If their model is correct, a cell that exists S-phase and has a lengthier G2 phase, should be less committed to enter mitosis. Maybe just prolonging G2 could be a way of testing this in \pm mitogen conditions? But ideally, this should be done independently of DNA damage.

6) The data related with cyclin A or Phospho Rb high/low levels presented in Supp Fig 5b. The authors state that the levels remained high for "6-8 hours before declining". I can see. A population of low levels emerging at this time, but there are still many cells in the high level category. So, how can the authors distinguish between these two?

7) The effect of cyclin A depletion on CDK activity is expected. But I am confused with its effect on mitotic entry. The mitotic timer is not affected. Can cells enter mitosis without CycA? Also difficult to understand is the effect of the level of depletion according to different siRNA concentrations. Moreover, the cell cycle exit is first detected at the same time- around 5hrs. It is true that more cells exit the cell cycle in CycA depletion, but maybe these cells have actually not even completed S-phase? I find it difficult to access how CycA depletion for a certain number of hours is not impacting the cell cycle per se. Maybe a inducible degradation system coupled to cell cycle markers would be a more useful tool.

8) The possibility that these results may contribute to improve chemotherapy remains highly speculative. in vivo models should be employed and I think testing this possibility in U2OS cells is really not relevant for this type of question.

Minor points:

1) Graphs like the ones shown in Fig1e or Supp Fig 2b are very difficult to interpret. The 2 cell cycle trajectories described on the graph on the right in Fig 1e, cannot be identified in the graphs mentioned previously.

2) The characterization of senescence Supp Fig 2C. There are small and large cells bGal positive.

3) Supp Fig 2C lacks a scale bar.

Author Rebuttals to Initial Comments:

General response to all reviewers:

Thank you for providing such thoughtful feedback to our manuscript. In this revised manuscript, we have included **41 new or revised figure panels** as well as **5 new supplemental videos**. By incorporating your suggestions, we believe the manuscript is now much strengthened. Below you will find a point-by-point response to each of the suggestions that were raised. Your comments are shown in *italics* and our responses are shown in **blue**.

Referee #1 (Remarks to the Author):

In this paper, Cornwell et al. proposed a revised Restriction Point model. In their model, rather than irreversibly committing to complete the cell cycle after R point, the cells will either complete mitosis or go to a G1-like state directly from the G2-phase, depending on whether they have enough CyclinA-CDK2 activity to drive them through G2 into mitosis. The authors showed that CDK4/6 activity is essential to maintain high Cyclin A levels in S/G2 phase by phosphorylating p107 and p130 pocket proteins, which pose an inhibitory effect on Cyclin A transcription when dephosphorylated. If the mitogen is removed or CDK4/6 activity is inhibited, Cyclin A transcription is suppressed by p107/p130 and Cyclin A protein levels start to drop. Depending at which point Cyclin A is inhibited, a cell may or may not have enough CDK2-CyclinA activity to sustain it through G2. If the levels of CDK2-CyclinA drop below a critical level then the cell ends up in a G1/0 state with 4N DNA content. This revised R point model explains the probabilistic behavior of cells following mitogen removal and also proposes an underlying molecular mechanism. It is therefore an important finding for the mammalian cell cycle field and suitable for a high-profile journal. However, given that the work is proposing to revise some long-held beliefs, and may be a landmark in the field, there are some experiments that require further corroboration or investigation. Below are my specific comments that should be addressed:

1. The positive feedback loop in the original R point model involves both Cyclins E and A. However, in this study, the authors only looked at Cyclin A. As a major part of the feedback loop, it is necessary to also investigate the dynamics of Cyclin E expression and Cdk2-CycE activity during the cell cycle and after mitogen removal. Especially given that Cyclin E seems unaffected by CDK4/6i (Extended 5d), it is unclear why CDK2-CyclinE cannot maintain the cell cycle exit clock? Is it because Cyclin E is degraded in S?

We thank the reviewer for raising this important point. Cyclin E-CDK2 activity does indeed play an important role in cell cycle progression. Cyclin E is transcriptionally induced by E2F1-3 early in G1 phase and is the first cyclin to activate CDK2 during the cell cycle¹. In G1 phase, cyclin E-CDK2 activity is responsible for hyper-phosphorylating Rb and further activating E2F1-3. However, once cells enter S phase, the APC/C is inactivated and cyclin A2 protein levels accumulate. Once in S phase, CDK2, bound to either cyclin E or cyclin A2, phosphorylates cyclin E targeting it for poly-ubiquitination by FBW7 and subsequent proteasomal degradation^{1,2}. Indeed, the half-life of cyclin E has been experimentally measured and is reported to be only 30 minutes, much shorter than the half-life of cyclin A2³. Therefore, as the reviewer suggested, Cyclin E is degraded during S/G2 phase making cyclin A2 the major cyclin activating CDK2 in S/G2 phase.

We have also conducted some new experiments to help address this important point. We

stained for cyclin E1 levels using immunofluorescence and plotted DNA content vs. cyclin E1 levels in asynchronous cultures of HeLa cells. Consistent with previous reports, we find that cyclin E1 levels are highest in G1 phase and then fall during S/G2 phase (new Extended Data Fig. 8a). We attempted similar cyclin E1 staining in MCF-10A, but the antibody for cyclin E1 was not sensitive enough for immunofluorescence. Therefore, in MCF-10A cells we performed an immunoprecipitation experiment with CDK2 to see which cyclins were bound to CDK2 in S/G2 phase (new Extended Data Fig. 8b). We found that, in cultures of MCF-10A cells, levels of cyclin E1 bound to CDK2 were highest in G1-arrested cells (CDK4/6 inhibitor treated) followed by a decrease of total, and CDK2-bound, cyclin E1 in G2-arrested cells (CDK1i treatment). In line with this, the amount of cyclin A2 bound to CDK2 is low in G1- and high in G2-arrested cells. After adding a CDK4/6 inhibitor to the CDK1 inhibited cells, the levels of cyclin A2, total and bound to CDK2, decreased dramatically. This corresponds to the loss of CDK2 activity as observed with our CDK2-activity sensor. While there is some cyclin E1 bound to CDK2 even in G2 cells, it is unchanged after adding a CDK4/6 inhibitor, implying that it is not responsible for the CDK2-sensor phosphorylation. The observed binding between CDK2-cyclin E1 in these conditions could be due to cells that were not arrested by the CDK1 inhibitor and were at a different stage of the cell cycle. Considering the published literature as well as these additional experiments, cyclin E is largely irrelevant to the cell cycle exit clock.

2. From Figure 2e and i, it is clear that the CDK2 activity also dropped after CDK1i treatment and HU treatment. Since the CDK1i and HU were used to extend the S/G2 phase but not affecting CDK2, its effect on CDK2 activity complicates the interpretation. Can the authors use a more specific approach to extend S/G2 without profoundly affecting CDK2 activity? Possibly the CDK2as cells from Fisher could be used. In general, I think the CDK2 sensor is most likely reporting a mix of CDK2 and CDK1 activities (eg Schwarz et al 2018).

The CDK2 activity does dip a bit after CDK1i treatment and bit more after HU treatment. As has been demonstrated by Schwarz et al, CDK1 when bound to cyclin A2 is capable of phosphorylating the CDK2 biosensor⁴. So likely this dip after CDK1i treatment is due to the small pool of cyclin A2-CDK1 in the cell. Nevertheless, this small drop is not sufficient to induce cell-cycle exit on its own. In order to more specifically inhibit CDK1 activity and leave CDK2 activity unperturbed we would ideally need CDK1as cells rather than CDK2as cells. However, it's likely that a CDK1as protein would still be bound and activated by cyclin A2 at some small level and therefore the same small dip in the CDK2 activity sensor would still be expected to occur.

The drop in CDK2 activity after HU treatment is due to Wee1 activity. HU inhibits ribonucleotide reductase and thus results in the depletion of deoxyribonucleotides. This has the effect of stalling active replication forks because the pool of dNTPs is not sufficient to support DNA replication. In the short term, stalled replication forks are stabilized and therefore do not result in DNA damage (new Extended Data Fig. 6a-d). As part of the fork stabilization process, ATR-CHK1 are activated which in turn inactivate the CDC25 phosphatases⁵. These phosphatases remove inhibitory phosphorylations from CDK2. When CDC25 is inhibited by CHK1, these inhibitory phosphorylations, like the ones mediated by Wee1, remain on CDK2 and therefore reduce CDK2 activity slowing down new origin firing⁶. In order to still use HU to extend S/G2 without profoundly affecting CDK2 activity we simultaneously treated cells with HU and a Wee1 inhibitor (MK-1775), which prevents the inhibitory phosphorylations on CDK2.

As shown in new Extended Data Fig. 7d,f, the Wee1 inhibitor rescued the dip in CDK2 activity following HU treatment, yet we still observe cells exiting the cell cycle into the G0-like state when treated with the CDK4/6 inhibitor. As an alternative approach to extending S/G2 phase we also repeated this experiment using thymidine as a replacement for HU. There is also a drop in CDK2 activity after thymidine treatment and this drop can also be reversed by co-treatment with a Wee1 inhibitor (new Extended Data Fig. 7e,f). Again, we observed cells exiting the cell cycle into the G0-like state when treated with the CDK4/6 inhibitor and as little as a 4 hour pulse of thymidine, with or without the Wee1 inhibitor. With these new experiments, we extended S/G2 length without profoundly interfering with CDK2 activity, and we still observe cells exiting into the G0-like state after CDK4/6 inhibition.

Finally, as suggested by Reviewers 3 and 4, we also attempted to extend S/G2 length by reducing the concentration of EGF. However, this approach only decreased the number of cells entering the cell cycle and did not have any effect on S/G2 length (Rebuttal Fig. 1a-c). Thus, this approach was not useful for testing our hypothesis.

Rebuttal Fig. 1: Reducing growth factor levels does not stretch S/G2 length.

MCF-10A p21^{-/-} cells expressing the CDK2 and APC/C activity sensors were incubated with minimal growth media supplemented with the indicated concentration of EGF and subjected to live-cell imaging for 48 hours. Note that full growth media contains 20,000 pg/mL EGF. **a**, Cumulative distribution function (CDF) plot of the intermitotic time (IMT) for cells incubated with the indicated concentration of EGF. **b**, CDF of the S/G2 length (defined as the time from APC/C inactivation to the subsequent anaphase) for cells incubated with the indicated concentration of EGF. **c**, Bar graph of the relative proportion of cells that either immediately re-entered the cell cycle (CDK2 inc), went to a temporary quiescent state before re-entering the cell cycle (CDK2 trans), or entered long-term quiescence for the duration of the imaging (CDK2 low).

3. The authors showed that CDK4/6i affects p107/p130 phosphorylation in S/G2, which is not the case for RB, so that Cyclin A transcription is inhibited. For this mechanism to work, p107/p130 might need to be specifically phosphorylated by CDK4/6 but not CDK2 or CDK1. But the authors didn't provide enough evidence to support that, especially for the CDK2 phosphorylation on p107/p130. In addition, in Extended 6d, they showed a time course of phospho-p130 after CDK4/6 inhibition. However, even after 4 hours, there's still residual p-p130 that can be seen, and this time scale is far longer than a common phosphorylation turnover. This actually suggests that CDK4/6 is not directly phosphorylating p130.

We thank the reviewer for raising this point. The Reviewer correctly suggests that for our proposed mechanism to work that p107/p130 need to be specifically phosphorylated by CDK4/6 but not CDK2 or CDK1 in the S and G2 phases of the cell cycle. Previous studies have shown that p107 is exclusively phosphorylated by CDK4/6 and not by CDK2, and that p107-induced

cell cycle arrest can be released by CDK4/6 but not CDK2⁷. This is in agreement with our analysis of p107 phosphorylation after CDK4/6 inhibitor treatment (Extended Data Fig. 9e), which shows that p107 is completely dephosphorylated within 1 hour of CDK4/6 inhibitor treatment while at the same time CDK2 activity is still high and Rb is still hyper-phosphorylated (Fig. 3b). If CDK2 was capable of phosphorylating p107 we would not expect to see such rapid loss of p107 phosphorylation after CDK4/6 inhibitor treatment.

Regarding the contribution of CDK2 activity to p130 phosphorylation. Previous studies have shown that some phospho-sites on p130 are exclusive to CDK4/6 and others are potentially phosphorylated by CDK2 and CDK4/6⁸. The phosphorylation site on p130 we evaluated in this study (S672) is specifically phosphorylated by CDK4/6 and has been shown previously to be dephosphorylated upon cell cycle entry after serum starvation and linked to regulation of cell cycle genes through disruption of the DREAM complex⁹. We attempted to test a number of additional p130 phosphorylation sites using commercially available phospho-specific antibodies, but all of the other antibodies we tested failed our validation attempts and appear not to specifically recognize p130.

To get around the problem of a lack of quality phospho-specific antibodies that are commercially available, we experimentally tested whether CDK2 activity may contribute to p130 phosphorylation. As the reviewer points out in Extended Data Fig. 9d there is some residual p130 that has not been dephosphorylated even after 4hrs treatment with a CDK4/6 inhibitor alone. To test whether this residual phosphorylated p130 was due to CDK2 activity we co-treated cells with a CDK2 inhibitor as well as a CDK4/6 inhibitor (new Extended Data Fig. 9f). We find that the addition of a CDK2 inhibitor, which effectively abolished Rb phosphorylation, did not lead to further loss of p130 phosphorylation. In addition, treatment with the CDK2 inhibitor alone effectively abolished Rb, but not p130, phosphorylation. With respect to a potential contribution from CDK1, given that in these experiments all cells were treated in the presence of a CDK1 inhibitor, we conclude that CDK1 has no contribution to the phosphorylation of either p107 or p130.

Thus, we believe the new data provided in this revision provides stronger evidence for specific regulation of p107/p130 by CDK4/6.

4. The authors propose a linear pathway model instead of a feedback model, however the observed phenomena do not necessarily contradict the existence of a feedback loop. The feedback loop can still exist on RB, but with an extra layer of regulation through CDK4/6-p107/p130. The reversible R point model can still have feedback loops. You have cell cycle transitions driven by positive feedback that are reversible, even though they are bistable. The examination of mitosis in xenopus extracts are a good example of this (eg Sha et al PNAS 2003; Pomerening et al NCB 2003). The authors haven't disproven the existence of positive feedback as implied in their schematic.

The reviewer makes a fair point that a feedback loop may exist but that the extra layer of regulation through CDK4/6-p107/p130 is dominant. We addressed this point in the Discussion section with the following new text:

“We cannot exclude the existence of positive feedback loops being involved in this signaling pathway since similar signalling systems, such as the one controlling the G2/M transition, have also been shown to contain positive feedback loops yet are still reversible^{10,11}. However, our data demonstrates that the dominant signalling architecture is largely linear in nature, resulting in a lack of hysteresis.”

5. In the revised R point model, two clocks are competing, and the cell cycle exit clock depends on Cyclin A protein level. The authors propose that once Cyclin A transcription is inhibited (via CDK4/6i or mitogen removal), the cell cycle exit timer is set by the Cyclin A protein half-life. Therefore, it will be a very strong support to the model if the authors could manipulate Cyclin A protein stability and show that the cell cycle exit clock length is sensitive to the half-life of Cyclin A. This is a difficult experiment, but would be very helpful in supporting the major point of the manuscript.

We thank the reviewer for this suggestion. Due to the difficulty in performing this experiment as the reviewer points out, we approached it a few different ways. First, we measured the cyclin A2 half-life in additional cell lines including U2OS and RPE1 cells. Interestingly, we found that the cyclin A2 half-life in U2OS cells was longer than in MCF-10A cells, suggesting that cyclin A2 is more stable in U2OS cells (new Fig. 31). Strikingly, the measured cell cycle exit clock and mitosis clock in U2OS cells were also longer than in MCF-10A cells (new Fig. 31). Likewise, we found that the half-life of cyclin A in RPE1 cells was shorter than in MCF-10A cells, and again we measured a shorter cell cycle exit clock (new Fig. 31). Thus, across different cell lines, the cyclin A2 half-life is correlated with the cell cycle exit clock (new Fig. 31). Notably, when we looked at the best-fit line for this correlation, we noticed the y-intercept was less than 2 hrs, which is remarkably similar to the time it takes for CCNA2 mRNA to be lost after CDK4/6 inhibition. Thus, if the cyclin A2 were to be completely unstable, then the exit clock would essentially be the time it takes for the cyclin A2 mRNA to be degraded. In addition, this new data raises the intriguing possibility that cancer cells, which often have a longer inter-mitotic time due to chromosomal abnormalities, may concomitantly increase the stability of cyclin A2 in order to maintain cell cycle progression. The temporal coupling of both the cell cycle exit clock and the mitosis clock across cell lines indicates there is a strong evolutionary pressure to ensure the mitosis clock is slightly shorter than the exit clock.

Next, we directly manipulated cyclin A2 protein stability as suggested by the reviewer. We obtained an RPE1 cell line where the endogenous cyclin A2 was edited to contain two inducible degron sequences: an auxin-inducible degron (AID) and a SMASH tag (RPE CCNA2^{dd}) (new Extended Data Fig. 11a)^{12,13}. The addition of the inducible degron already reduced the stability of cyclin A2 from 10 hours in the parental line to 5 hours in the engineered line. The cyclin A2 stability could be further reduced to 2 hours by addition of a DIA “cocktail” (new Fig. 31). To demonstrate how this reduction in the half-life of cyclin A2 impacted the cell cycle exit clock we treated cells with a CDK4/6 inhibitor alone or in combination with DIA and measured the mitosis and cell cycle exit clock in single cells (new Extended Data Fig. 11b). As expected, all cells treated with DMSO made it to mitosis and did not exit the cell cycle. CDK4/6 inhibitor treatment caused cells that were closer to S phase when treated to exit the cell cycle instead of making it to mitosis, with only cells between 10-14 hrs after S phase entry dividing

(new Extended Data Fig. 11c-e). Co-treatment with a CDK4/6 inhibitor and DIA had a more pronounced effect on cell cycle exit with all cells exiting the cell cycle up to 12 hrs after S phase entry. Thus, by reducing the half-life of cyclin A2 and decreasing the timing of the cell cycle exit clock, cells became more sensitive to the loss of CDK4/6 signalling and were less likely to divide and more likely to exit the cell cycle.

In summary, we manipulated cyclin A2 protein stability both by taking advantage of the natural variability in cyclin A2 stability across cell lines as well as using an inducible degron to specifically shorten the cyclin A2 half-life. In both cases, we measured a strong correlation in the cyclin A2 half-life and the length of the cell cycle exit clock.

Referee #2 (Remarks to the Author):

This very interesting paper reexamines the mechanism underpinning the restriction point, traditionally defined as the time during G1 phase at which a cell irreversibly commits to carrying out a round of DNA replication and mitosis. The conventional view has been that double-negative feedback between Rb and CDK2 (via E2F) and then between CDK2 and Rb (via direct multisite phosphorylation) generates an irreversible bistable control circuit that accounts for the irreversible, all or none character of restriction point traversal.

One problem with this idea, and a key observation for this paper, is that although many S phase cells will complete DNA replication and mitosis after washing out mitogens or applying MEK or CDK4/6 inhibitors, not all of them will, and this fact is true in all of the cell lines examined here, although the exact proportion of outlier cells depends somewhat on the treatment (washing out mitogens vs. adding an inhibitor) and the cell line chosen.

To account for the outliers, the authors have proposed an alternative hypothesis, which they call the competing clock model. If their hypothetical mitotic clock is maximally stimulated (with full growth medium) and presumably running maximally quickly, few of the cells fail to complete S- and M-phases because mitosis occurs before the cell cycle exit clock goes off, and the two fates (mitosis and exit) are mutually exclusive. On the other hand, if the mitotic clock is slowed by treatment with a CDK1 or PLK inhibitor or by triggering the DNA damage checkpoint (three ways of slowing or blocking CDK1 activation and mitotic entry), the cell cycle exit clock goes off in most of the cells and so a high fraction of the cells exit the cell cycle. Moreover, these cells lose their CDK2 activity before exiting the cell cycle, as reported by a human helicase B-based fluorescent biosensor, showing that under these circumstances, CDK2 activity is not infinitely self-sustaining.

Next they show through some nice quantitative experiments that if one assumes that normally the cell cycle exit clock is a little slower and a lot more variable from cell-to-cell than the mitotic clock is, and further assume that the clocks can be regulated independently, one can account for the proportion of cells that exit the cell cycle when S-phase has been extended by various durations of HU treatment. This is a satisfying test of the competing clock model. Finally, they present evidence that continued CDK4/6-mediated inhibition of p107 and p130, rather than CDK2-mediated inhibition of Rb, is needed to keep S phase cells from exiting the cell cycle, and they show that an ODE model without the CDK2-Rb double negative feedback can account for the proportion of “outliers” present under various circumstances.

I think that this is a very strong paper, with the data in Figs 1 and 2 and the competing clocks concept being the principal strengths. I have some minor criticisms, but on the whole this is a fascinating new look at an important old problem.

Minor criticisms:

1. Line 17-18. Perhaps it would be more accurate to say “cyclin A transcription depends upon CDK4/6 activity throughout the cell cycle.”

We agree that this is a more precise description and have modified the text accordingly.

2. Line 11 and elsewhere: The protein name “cyclin” should not be capitalized (likewise for any protein name that is not an abbreviation or acronym—think actin, huntingtin, insulin, trypsin...)

We apologize for this mistake. The text has been modified accordingly.

3. In general the authors’ whole “probabilistic vs. deterministic” theme is the one aspect of the paper that I do not think is well supported by the data, and which I think is not clearly thought through or explained. Really it is the fact that “outlier cells require mitogen signaling to maintain Rb phosphorylation [line 40]” at times way past when R is traditionally thought to have occurred, rather than the probabilistic character of R at the level of a population of cells (which could have arisen simply from cell-to-cell variability) that doesn’t fit well with the classical view of R. In other words, these outlier cells are not mitogen-independent the way post-R cells are supposed to be. The authors need to do some re-writing to clean this up.

The Reviewer makes a very good point. Our original thought process behind the “probabilistic vs deterministic” discussion was our observation that the probability of cells exiting to the G0-like state decreased as cells progressed towards mitosis (shown in Fig. 2a,b). However, as the Reviewer rightly points out, later in the manuscript we demonstrate that the decision to either exit or go through mitosis is determined by the relative timing of the cell cycle exit clock and mitosis clock. Thus, we have removed the discussion of probabilistic cell cycle commitment, we have revised our model in Fig. 4g, and revised the accompanying text in the results section:

“This revised model of cell cycle commitment implies that there is no single point when cells irreversibly commit to proliferation that can be defined by a single molecular event but is rather determined by the cells proximity to mitosis as well as the levels of cyclin A2 protein when mitogen signaling is lost (Fig. 4g).”

4. Fig 1a, c. The authors say that before R, you have CDK2-Rb off, and after R you have CDK2-Rb on. I know what they mean, but CDK2 and Rb are in a double-negative feedback relationship. Thus, before R you have Rb on (i.e. active as an E2F inhibitor) and CDK2 off, and after R you have Rb off and CDK2 on.

We thank the Reviewer for pointing out this point of confusion. We have addressed this by modifying Fig. 1a,c and the text as suggested by the Reviewer.

5. Lines 54-56. How about “we serum-starved MCF-10A cells or treated them with a MEK inhibitor or a CDK4/6 inhibitor....” Without the “or” that I added, this sentence could be read as meaning that the authors serum-starved the cells and treated them with one of the inhibitors.

Yes, this is a great point. We have edited the text accordingly.

6. The example cell time courses shown in insets in Fig 1e are too small to read.

We have now modified Fig. 1e to address this comment as well as a similar comment by Reviewer 4 (Minor Point #1). We have now elevated the example single-cell times courses to their own figure panel (new Fig. 1e,f), rather than an inset with the heatmaps (new Fig. 1g). We hope the increase in the size of the example cell time courses will make them easier to read. We also included a schematic in new Extended Data Fig. 2b to explain how the CDK2 and APC/C activity time courses are converted to a colormap, linearized, and stacked to generate the heatmaps shown in Fig. 1g, Fig. 2a, Extended Data Fig. 2c, and Extended Data Fig. 3a. Finally, we have now included supplementary videos (new Supplementary Video 1-5) for these same example cells. We hope these changes will improve the clarity of the data as well as its legibility.

7. Fig S1. It would be helpful to point out in the figure legend which of these cells are primary cells (I think it's just the HLFs), which of them are immortal but non-transformed, and which (if any) are transformed.

This information has now been added to the figure legend of Extended Data Fig. 1a.

“a, Effect of loss of mitogen signalling on Rb phosphorylation and DNA content in diverse cell types: MCF7, transformed breast epithelial cells; U2OS, transformed osteosarcoma cells; RPE-1, non-transformed hTERT-immortalized retina pigmented epithelial cells; HLF, primary human lung fibroblasts; MCF-10A, non-transformed breast epithelial cells.”

8. Line 94. For clarity, please change “closer to S-phase” to “closer to the start of S-phase” (assuming that's what you mean).

Yes, the reviewer is correct, and this suggestion certainly helps add clarity to the manuscript. We have made this correction to the text.

9. Line 112. This should be “blocking CDK4/6 signalling” rather than “blocking mitogen signalling”.

We have made this correction to the text.

10. Line 120. Likewise—“CDK4/6 signalling”, not “mitogen signalling”.

We have made this correction to the text.

11. Fig 4b. The authors should point out that their “linear” model actually has a lot of non-linearity designed in; it just doesn’t include the double-negative feedback loop between Rb and CDK2, and therefore does not yield bistability. Because of the non-linearity, the output of the model is still highly switchlike, but it is not hysteretic or irreversible.

We thank the reviewer for this suggestion. We have added the following text to the Discussion section to make this point clearer:

“This linear pathway contains ultrasensitive signaling nodes, which makes the system bistable and switch-like, but the lack of a dominant positive-feedback loop means the system is still reversible.”

12. Line 268. If the two cell fates—cell cycle exit vs. mitosis—are mutually exclusive, you could say there must be some sort of double negative feedback here, where cell cycle exit makes it hard or impossible for mitosis to occur, and mitotic entry makes it hard or impossible for cell cycle exit to occur. If so then there may well still be bistability involved in the process, albeit at the level of the mutually exclusive mitotic vs. cell cycle exit states rather than mutually exclusive CDK2-on Rb-off vs. CDK2-off Rb-on states. This is probably worth pointing out in the discussion section.

This is a good suggestion and we have edited the Discussion section to point this out:

“Given that the two competing fates (mitosis vs cell cycle exit) are mutually exclusive there is likely a double-negative feedback loop at the level of these cell fate decisions that ensures once a cell exits, it makes it impossible to undergo mitosis and vice versa.”

13. Also worth pointing out in the Discussion: nothing presented here proves that the CDK2-Rb circuit is not bistable. The authors do show that CDK2 activation is not irreversible, but a bistable circuit might yield hysteretic but not irreversible CDK2 activation (just as the mitotic control circuit yields hysteretic but not irreversible CDK1 activation). The important point to me, though, is that one does not need to assume that there is bistability to account for any of the experimental results presented here—the authors’ competing clocks hypothesis, and the linear or no-feedback mathematical model, suffice (as long as high degrees of ultrasensitivity are included).

We agree with this point and have added the following text to the Discussion section:

“This linear pathway contains ultrasensitive signaling nodes, which makes the system bistable and switch-like, but the lack of a dominant positive-feedback loop means the system is still reversible.”

Referee #3 (Remarks to the Author):

This work aims to determine how cells commit to proliferation, starting from an observation that, when mitogen signalling is disrupted, there is often a small fraction of cells (~5-15%) in a cell population that exit the cell cycle in G2 rather than completing a whole round of division by completing mitosis. This observation calls into question the irreversible nature of the Restriction Point (R) i.e. that once cells have passed R, they will complete mitosis, even in the absence of ongoing mitogenic signalling. The authors then combine single-cell imaging, perturbation, molecular biology and modelling approaches to show that there are two competing fates for cells that have passed R when mitogens are withdrawn and that the choice between these two fates relies on the stability of Cyclin A protein which sets a timer for mitosis. If Cyclin A protein is maintained then cells complete mitosis, if not, cells exit the cell cycle in G2. This property seems to be unique to Cyclin A and the authors show how Cyclin A transcription is regulated by CDK4/6 signalling to p107/p130-repressive E2F complexes. This is a novel and unexpected result that Cyclin A transcription is regulated by release of repressive transcription factors, rather than a positive feedback loop between CycA/CDK2 and E2F1, as has been widely believed. Finally, they show combining their findings could lead to a new combination of CDK4/6i and doxorubicin to drive cell cycle exit from G2 and senescence in cancer cells.

This paper was a pleasure to read and, while I am unsure about the analogy they have used with the two competing clocks (see comments below), the data themselves are novel and exciting. The discovery that CCNA2 transcription is not controlled by positive feedback by E2F1 and the proposal of a linear model is unexpected and important. This finding also calls into question other aspects of our current understanding of the system. The idea that passing R does not represent irreversible commitment to cell division is not necessarily a new one but the idea of a timer determining cell fate (mitosis versus cell cycle exit) is convincing. The inclusion of cartoons to explain some quite tricky concepts was also very welcome. I have a few comments and question below that I hope will improve the manuscript. I am unqualified to comment on the modelling aspects and other reviewers will be better placed to assess that part of the work. I am not convinced about the logic behind the combined use of doxorubicin and CDK4/6i in the clinic, for reasons outlined below (but this shouldn't distract from the main findings of the paper).

Major comments:

1. Testing the mitotic timing model using hydroxyurea. The authors use hydroxyurea to delay the mitotic clock (Figure 2, Extended Data Figure 4). However, hydroxyurea will cause replication stress during S-phase (which is why S-phase gets longer). This will activate p53 and so G2 gets longer because of p21 accumulation. p21 will then suppress CDK2 activity in G2 and lead to cell cycle exit. The longer the HU treatment, the more damage, more p21 accumulation and more cells undergo cell cycle exit. Therefore, I don't think this data helps to make their point about the two clocks and the role of Cyclin A as this is inducing a whole different mechanism to drive cell cycle exit. If they could show the same result in p21KO cells then this would be more convincing, but I suspect they will not see the same phenotype. Are there other ways to extend cell cycle length such as reducing the concentration of growth factors?

We thank the reviewer raising this important point. First, we should point out that we did use p21 KO cells for this particular experiment (MCF-10A p21^{-/-}). Similar to the reviewer, we were initially concerned that the p53-p21 pathway may interfere with the interpretation of this experiment. Therefore, we had originally decided to use p21^{-/-} cells to eliminate that potential interference. We apologize for the confusion and have now specified when we used MCF-10A

p21^{-/-} cells in the relevant figure legends. More importantly however, is we still see cell cycle exit in these p21^{-/-} cells when we stretched S/G2 phase by as little as 2 hours and treat the cells with a CDK4/6 inhibitor (Fig. 2k). This demonstrates that the loss of CDK2 activity in this experiment is not due to CIPs like p21, but as we show later in the manuscript, by the loss of cyclin A2 mRNA and protein.

Second, HU inhibits ribonucleotide reductase and thus results in the depletion of deoxyribonucleotides. This has the effect of stalling active replication forks because the pool of dNTPs is not sufficient to support DNA replication. In the short term, stalled replication forks are stabilized and therefore do not result in DNA damage^{14,15}. However, long-term HU treatment does result in DNA damage. At the time scales we used HU (less than or equal to 6 hrs) there should be no DNA damage occurring according to previously published reports^{16,17}. To demonstrate this point, we treated cells with 2, 4, 6, or 24 hrs with HU and stained for the DNA damage marker γ H2AX (new Extended Data Fig. 6a,b). We find no increase in γ H2AX if treated with HU for 6 hours or less. In fact, if anything, there is a reduction in DNA damage because DNA replication is stalled and there is therefore less replication-associated endogenous damage occurring. We do see an increase in γ H2AX staining after 24 hrs of HU treatment, consistent with previous studies that show long-term HU treatment causes DNA damage^{14,16,17}. In addition, the lack of DNA damage by these short-term treatments explains why we see these phenotypes even in p21^{-/-} cells.

Third, we will point out that the short-term treatment with HU only delays the S/G2 length by the precise time the cells were incubated with the treatment (eg. 2 hours of HU treatment delays S/G2 length by ~2 hours; see Extended Data Fig. 7a,b) and after washing out the drugs the cells are still able to divide normally. This is consistent with no DNA damage being induced, since the damage would be likely to delay mitosis by greater than the time of treatment or prevent mitosis entirely.

Fourth, we repeated all these experiments with thymidine to extend cell cycle length and found similar results to when we used HU (new Extended Data Fig. 6c,d and 7b,c,e,f,h).

Given these four points, we do not believe that the HU treatments we used, nor the thymidine treatments that we now include in the revised manuscript, are inducing DNA damage that could be interfering with the interpretation of our results. However, we did explore the possibility of stretching S/G2 phase using other methods such as reducing the concentration of growth factors as the Reviewer as well as Reviewer 4 suggested. However, the main effect of growing cells with reduced concentrations of growth factors is not to stretch the inter-mitotic time or the S/G2 length (Rebuttal Fig. 1a,b), but to rather reduce the population of cells that enters the cell cycle in the first place (Rebuttal Fig. 1c). Thus, this approach was not useful for testing our hypothesis.

Rebuttal Fig. 1: Reducing growth factor levels does not stretch S/G2 length.

MCF-10A cells expressing the CDK2 and APC/C activity sensors were incubated with minimal growth media supplemented with the indicated concentration of EGF and subjected to live-cell imaging for 48 hours. Note that full growth media contains 20,000 pg/mL EGF. **a**, Cumulative distribution function (CDF) plot of the intermitotic time (IMT) for cells incubated with the indicated concentration of EGF. **b**, CDF of the S/G2 length (defined as the time from APC/C inactivation to the subsequent anaphase) for cells incubated with the indicated concentration of EGF. **c**, Bar graph of the relative proportion of cells that either immediately re-entered the cell cycle (CDK2 inc), went to a temporary quiescent state before re-entering the cell cycle (CDK2 trans), or entered long-term quiescence for the duration of the imaging (CDK2 low).

2. *Logic of combining doxorubicin and CDK4/6i in cancer treatment. I can see why they have proposed this based on their data, and the results shown are convincing. However, my big problem here is that CDK4/6i seem to have a unique sensitivity towards transformed cells (for reasons people are only just starting to understand). However, the treatment regime proposed here would lead to the loss of that cancer cell specificity since it would also cause non-transformed cells to exit the cell cycle in G2 and become senescent, which could lead to adverse side-effects. Although they don't show this here, I predict that if they performed the experiments shown in Extended Figure 8 in non-transformed cells (e.g. MCF-10A) they would see the same results as with U2OS. I would recommend taking this part of the work out of the paper. The rest of the work stands well on its own.*

We thank the reviewer for this suggestion. Given this comment as well as the comment from Reviewer 4, we think it is best to remove Extended Data Fig. 8.

3. *The analogy of having two competing clocks. I don't find the analogy of having two competing clocks helpful (and I also worry that it will annoy some in the cell cycle field). In reality, is this not a timer? If cells get to mitosis before the timer (Cyclin A protein) runs out, then cells divide and complete the cycle. If not, they exit the cycle from G2. The authors may point to Fig 3I to argue that there are two distinct clocks but I am not sure it does show that? Do all cells in that experiment have a reduction in Cyclin A protein (i.e. homogenous depletion). Or are cells that have slightly less CycA already (the ones that would be most likely to exit in G2 in normal conditions) likely to be affected more so CDK2 activity is lost faster?*

The purpose of the clock analogy was to conceptualise the phenomenological observations we made before we could uncover the molecular mechanism explaining these observations. The reviewer is correct that “timer” is also a useful analogy but we think using “clock” vs “timer” is largely a semantic argument. The mitosis clock we describe in the manuscript refers to the time it takes for a cell to progress from S phase to mitosis and the cell cycle exit clock, or timer, refers to the time it takes for cells to lose CDK2 activity and exit the

cell cycle after the loss of mitogen signalling. Given that mitosis and cell cycle exit are two distinct and mutually exclusive fate outcomes we developed the conceptual framework of two distinct clocks that represent the time it takes to realise each of these two mutually exclusive fates. Therefore, our nomenclature of two distinct clocks merely describes our observations of two distinct fates for the cell and the time it takes for each fate to be realised.

Furthermore, in our Monte Carlo simulations we assumed that these clocks are independent. To demonstrate experimentally that these two clocks are independent we sought to manipulate the timing of the cell cycle exit clock by partially knocking down cyclin A2 and assessing how that affected the timing of mitosis (mitosis clock) and the time to lose CDK2 activity (the cell cycle exit clock). The results for this experiment, as shown in Extended Data Fig. 10i, support our assumption of independent competing clocks since we were able to drastically shorten the cell cycle exit clock by 50% with only a <5% change in the mitosis clock. Thus, our observation of two distinct fates that have different timings supports our idea of their being two distinct clocks present within the cell, while Extended Data Fig. 10i demonstrates that the timing of these two fates (clocks) are controlled independently of one another.

We have also included new data showing the level of depletion of cyclin A2 protein in these cells (new Extended Data Fig. 10e). We observed a unimodal rather than bimodal knockdown effect, suggesting that the siRNA treatment led to fairly homogenous knockdown of cyclin A2 in all cells. It is difficult for us to assess whether cells with slightly less cyclin A2 were more sensitive to the siRNA treatment. However, we considered that if that was the case then we should be able to identify a subpopulation of cells that lost CDK2 activity faster than that observed with non-targeting siRNA. However, when we analysed the time to exit the cell cycle for cells treated with cyclin A2 compared to non-targeting siRNA we found that the minimum time to exit the cell cycle was similar (Extended Data Fig. 10i). These data suggest that 1) the knockdown of cyclin A2 was homogenous across all cells analysed and 2) that this led to a reduction in the exit time in all cells rather than a subpopulation of cells that had lower starting levels of cyclin A2.

4. Cyclin A protein being key to the timer. One experiment that I think is missing to show clearly that the level of Cyclin A protein determines the fate between mitosis and cell cycle exit would be to use the CycA2-mVenus line they have and treat with CDK4/6i and see how the level of CycA2 is related to cell fate. According to the model, cells with less Cyclin A protein at the end of S-phase should have a higher probability of exiting the cell cycle than those with a higher level of CycA protein.

This is a great suggestion from the Reviewer. We re-analyzed our data from previous experiments and now show that the level of cyclin A2 protein at the time the CDK4/6 inhibitor was added is predictive of the cell fate. Cells with low cyclin A2 protein at the time of CDK4/6 inhibitor treatment are more likely to exit than cells with higher levels of cyclin A2 (new Extended Data Fig. 10h). In addition, we analysed the cyclin A2 levels in cells which exited the cell cycle after CDK4/6 inhibitor treatment and compared it to the level of cyclin A2 in cells which made it to mitosis and divided (new Extended Data Fig. 10g). We find that the cells which exited the cell cycle had a median cyclin A2 level less than 50% of those which made it to

mitosis. We believe this new analysis suggested by the reviewer has strengthened our conclusions that cyclin A2 protein levels are key to the cell cycle exit clock, or timer.

Minor comments:

1. It is known that CycA/CDK2 also shuts down activator E2F transcription (this is also mentioned in the text). However, this interaction is missing from Figure 3A. Have the authors considered including this in their ODE model to see how it would change CDK2 activity?

Yes, we did consider this interaction in the ODE model but it did not have an effect on CDK2 activity. The main effect of adding this interaction to the model is to make E2F levels more closely track with CDK2 activity, but because the model does not include feedback from E2F back onto CDK2, there was no effect on CDK2 activity. Furthermore, because of the lack of the feedback between E2F and cyclin A2, including this interaction in our model does not change the lack of hysteresis that we observed in Fig. 4f. For these reasons, we chose not to include the interaction in the ODE model.

2. In Extended Figure 2c, how long after treatment was SA B-gal assayed?

We apologize for omitting this piece of information and we appreciate the reviewer bringing this to our attention. The cells were treated for 7 days before SA β -gal was assayed. We have added this information to the Methods section and the relevant figure legend.

3. Extended Figure 6D, can the authors quantify the reduction in phospho-p130 relative to the total amount of p130? This is important as the total amount of p130 protein appears to decrease after CDK4/6i treatment, which has been described for pRb (Dang et al., Nat Comms 2021).

We have now included a quantification of the phospho-p130 levels normalized to the total levels of p130 as suggested by the reviewers. This data is now included as new Fig. 3i.

4. Figure 4f – it wasn't clear how these experiments were performed and the timing determined? A better description would be appreciated.

We apologize if our original description of this experiment was confusing and we thank the Reviewer for pointing this issue out. We can certainly provide a better description in the methods section. The added space in the Methods section compared to the figure legend we think will allow us to go into more detail about how this critical experiment was performed. We include the new description here as well for the Reviewer's convenience.

“Our ODE model results and experimental data suggest a lack of hysteresis in post-R cells with respect to mitogen signalling. To test for hysteresis in pre- and post-R cells we treated them with different doses of a MEK inhibitor and measured the fraction of cells that exited the cell cycle (CDK2 activity <0.6) as measured at different times after treatment (4, 10, 15, and 24 hrs) by live-cell imaging. All cells were pre-imaged to establish the history of each cell, and then cells were treated with a MEK inhibitor (Trametinib, S2673, Selleckchem, USA) at doses ranging from 0.01 nM to 100 nM. After treatment, cells were

continuously imaged for more than 24 hours. To determine cell cycle status, CDK2 activity was measured using the CDK2 reporter as described above.

To identify the effect of MEK inhibition on pre-R cells we treated pre-R cells with various concentrations of a MEK inhibitor. In MCF-10A cells, MEK inhibition does not lead to cell cycle arrest when added in G1 phase¹⁸, indicating these cells are born post-R. Therefore, in our analysis we selected daughter cells whose mother had received the MEK inhibitor between 0 and 6 hours before mitosis. These daughter cells were considered as pre-R cells and the fraction of cells at each dose of MEK inhibitor that had exited the cell cycle (low CDK2 activity) 4 hours after mitosis was plotted. Additional time points after MEK inhibitor treatment were not plotted for pre-R cells since 4hrs after mitosis was sufficient for nearly all cells to exit the cell cycle and generate a clear dose-response curve.

To identify the effect of MEK inhibition on post-R cells we treated post-R cells with a combination of MEK inhibitor and a CDK1 inhibitor (to prevent the cells entering mitosis). We then measured the fraction of these cells that had exited the cell cycle (low CDK2 activity) at 4, 10, 15, and 24 hours after treatment. The same proportion of post-R cells will exit the cell cycle at the same dose of MEK inhibitor as pre-R cells, if given sufficient time (comparing blue line vs grey line). This analysis provides evidence for a lack of hysteresis in CDK2 activity with respect to mitogen signalling since both pre-R and post-R cells will lose CDK2 activity at the same dose of MEK inhibitor if given enough time to reach steady state.”

5. In the IP experiments – how long was CDK4/6i added to cells before the IP?

Again, we apologize for omitting this piece of information and we appreciate the reviewer bringing this omission to our attention. The cells were treated for 4 hours before doing the IP. We have added this information to the relevant figure legend.

6. Why is SEM used in some figures and SD in others?

Thank you for pointing out this potential inconsistency. We have gone back through the manuscript and double checked to make sure that the description for all error bars is correct. We use SEM when representing error between biological replicates and use SD when representing population spread from either single-cell data or technical replicates, in which case data is a representative experiment from $n \geq 3$ biological replicates. The exact n is listed in the figure legend for each panel.

7. Typos: Extended Figure 2 – should be a and not c in the legend. Extended Figure 3e – typo on graph – should be removal or removed.

Thank you for pointing out this error. It has been corrected.

Referee #4 (Remarks to the Author):

The study by Cornwell and colleagues, on the temporal competition between opposing fates underlying cell cycle fate choices, addresses the commitment between cell cycle exit or proliferation. It has been extensively described, and more or less taken for granted, from work performed in many different model systems that the Restriction (R) point bifurcates the dependency on proliferation signals so that pre-R cells read the presence/absence of mitogens to commit to proliferation or cell cycle exit. In the absence of mitogens, pre-R cells exit the cell cycle, while once the R point has been bypassed, cell cycle exit can no longer occur until the next G1 and so cells will finish the cell cycle with cell division.

In this study, the authors decided to interrogate if this long standing assumption holds always true. They cultured MCF-10 cells in the absence of serum or inhibited CDK4/6 or MEK and found that cells can exit the cell cycle even before mitosis. They propose that cells somehow read the activity of CDK2-CyclinA, informing on the time elapsed between S-phase and mitosis. In other words, they propose that there is no irreversible commitment to proliferation, but rather a higher probability of enter mitosis as cells approach M phase.

Understanding cell cycle regulation and the underlying principles that govern transitions from cell cycle phases in order to complete it or exit the cell cycle are of the uttermost importance in biology. They influence developmental decisions and have the potential of helping designing therapies related with cell cycle dysfunction. This work is interesting and raises novel questions in the cell cycle field. However, even if quite relevant for the cell cycle field, I am not sure that it delivers a straightforward message that explains their findings. Moreover, the relevance of their findings, is also difficult to assess. They use many different drugs that may have a certain effect beyond the one being described here. Further, certain findings may be interpreted otherwise, and so in the end is difficult to assess the novelty and relevance of their findings. The main findings of this study can be summarized by the explanation that the R-point has been described due to the half-life of cyclin A2 allowing sufficient CDK2 activity even in the absence of proliferation signals. Even if interesting and appealing, I am not sure that this represents such a breakthrough in the field to be published in this journal.

Major points:

1) The authors very frequently mentioned that their work is against the text book work and reveals novel findings related with cell cycle exit before mitosis. In Fig 1d, they show that Mitogen removal, Meki, CDK4/6i, result in the appearance of a novel cell population -G0-like with hypo-phospho-Rb. But the frequency of these cells ranges from 8.7% to 14.6%. Which in my opinion is very minor, even if worth studying. But the main point here is that, the large majority of cells behaves as expected, no? So, while these results are interesting, I think they show that the text book view is corrected, no? At least for MCF-10A cells.

Throughout the manuscript we compare and contrast our data to that which has been extensively published in scientific literature as well as basic cell biology textbooks. Below we highlight our rationale for making this comparison and for our statements that our work is ‘against the textbook model’:

*Page 5, Line 91: “Therefore, in contrast to the **textbook model** of the R-point (Fig. 1c) we find that not all post-R cells are irreversibly committed to proliferation since some cells exit the cell cycle to a G0-like state after loss of mitogen signalling (Fig. 1j).”*

We make this statement in the manuscript because it is widely believed that post-R cells will complete mitosis and divide into two daughter cells when mitogens are removed or mitogen signalling is blocked¹⁸⁻²³. Therefore, our observation of a population of 4N cells which have lost Rb phosphorylation is against this widely held and widely published prediction of the restriction point model.

Page 6, Line 127: “This means that post-R cells are unable to sustain CDK2 activity in the absence of CDK4/6 signalling, in stark contrast to the **textbook model** in which CDK2 activity is self-sustaining via a positive feedback loop. “

And

Page 11, Line 247: “This lack of true irreversibility in CDK2 activation challenges the **textbook model** which states that hysteresis in CDK2 activity with respect to mitogen signalling underlies irreversible cell cycle commitment at the R-point.”

And

Page 11, Line 232: “Our data demonstrating that loss of mitogen signaling cyclin A2 transcription is repressed by p107/p130, several hours before any detectable changes in CDK2 activity or Rb phosphorylation, is at odds with the **textbook model** which predicts that cyclin A2-CDK2 activity is self-sustaining due to a positive feedback loop”

It is widely published that the irreversible decision for post-R cells to divide after loss of mitogen can be attributed to a signalling architecture involving CDK2 and Rb that contains a positive feedback loop, and thus from a signalling perspective CDK2 activity in post-R cells should not be affected at all by any changes to upstream mitogen signalling^{4,21,24-26}. However, in our manuscript we show the opposite is true, i.e. that loss of upstream mitogen signalling can completely reverse Rb phosphorylation and CDK2 activity in post-R cells. This has not been previously described in the literature and therefore our data disproves this concept that CDK2 and Rb phosphorylation are reversible and thus also disproves the textbook model.

The reason we originally studied these outlier cells is because their existence is not predicted by the current Restriction Point model. If the Restriction Point model were correct, then it should be able to account for these cells. By trying to understand why these relatively small percentage of cells behaved counter to the Restriction Point model we made the surprising observation that the Restriction Point model is wrong for all cells, not just this small population of cells. In other words, we did not find that these “outlier” cells are fundamentally different from the rest of the cells in the population, but rather we found that all cells are capable of exiting the cell cycle in response to loss of mitogen signaling.

In this study, we described the signaling network underlying cell cycle progression that exists in 100% of cells, and not just a small percent of cells. In support of this conclusion, in Fig. 2 we demonstrate how a temporal competition between mitosis and cell cycle exit predicts why we observed only ~10% of cells exiting into the G0-like state. We test this prediction by preventing mitosis and now see the population of cells exiting into the G0-like state goes from 10% to nearly 100% of cells. This demonstrates that all cells fail to irreversibly commit to the cell cycle. Furthermore, we show that simply stretching the S/G2 length by a few hours (Fig. 2k and Extended Data Fig. 7d) is also sufficient to result in nearly 100% of cells exiting into the

G0-like state. Thus, the main point here is that the large majority of cells actually behave counter to the Restriction Point model, its just that entry into mitosis precluded this observation in all but a relatively small percentage of outlier cells. This is the justification for saying the textbook view is not correct.

2) Still related with this point, I do not understand the choice of cells and certain approaches. The choice of MCF-10A cells and 48hrs of treatment. How long is a cell cycle in MCF-10A cells? The authors also use other cell lines (see below) shown in Supp Fig 1a. Take for example RPE1 cells, which have a cell cycle of around 22hrs. 48hrs of treatment appears quite long, as it will expose cells to a long time in the presence of an inhibitor or without serum. Could this lead to certain artifacts and so to the observations described here? It is even surprising that cells can survive this long without serum, no? Maybe an alternative approach would be to film these cells expressing the FUCCI system, so that they can follow individual cell behaviour from a known starting point. And like this, expose less time the cells to these drugs. This approach also has the advantage of providing information on how long will it take to get to the G0-like status.

Regarding our choice of cells. We selected MCF-10A cells as the main cell line to be used in the study since they have been extensively utilised as a normal (immortalized, non-transformed, breast-epithelial) cell line for studying the cell cycle. We also selected RPE-1 cells as they are also used routinely to study the cell cycle. In addition to these non-transformed cell lines we also chose to include the transformed cell lines U2OS and MCF7 cells. Since by showing that even cancer cells can respond similarly, i.e. exit the cell cycle from S/G2 phase due to loss of mitogen signalling, it increases the generality of our claims and more strongly supports our claim of a new universal model of cell cycle commitment. The Reviewer correctly points out that each of these cell lines has a different intermitotic time with MCF-10A and RPE-1 cells having relatively short intermitotic times compared to MCF7 and U2OS cells. While we did not select the cells based on this particular characteristic, it turned out to be informative since we find the cell cycle exit clock appeared to be coupled with the intermitotic clock across different cell lines (new Fig. 3l and new Extended Data Fig. 5h-m).

Regarding the length of treatment being too long and cells not being able to survive in the absence of serum. We should note that MCF-10A cells are typically grown in a media containing 5% horse serum, EGF, insulin, and several other supplements. During serum starvation, the cells are grown in their typical media but without the 5% horse serum, EGF, and insulin but supplemented with BSA. The precise media recipe is included in the Methods section. Furthermore, it is well established that cells in culture can survive for many days without serum, and it is standard practice in the field to serum-starve cells for anywhere from 2 to 7 days to induce quiescence or to generate cells that progress through the cell cycle in a synchronized fashion^{22,27-30}. During the time cells spend in the absence of serum negligible cell death occurs and cells are able to re-enter the cell cycle and divide as normal after being re-stimulated with mitogen-containing serum.

We utilized live-cell imaging exactly as described by the Reviewer extensively in this study as shown in Fig. 1e-g, Extended Data Fig. 1c, and for most of the experiments in Fig. 2 and 3. The methods used for these live-cell imaging experiments are outlined in detail in the Methods section. Briefly, we did live-cell imaging with cells expressing the geminin-mCherry FUCCI reporter (geminin levels were converted to APC/C activity³¹) as well as a live-cell CDK2

reporter²². We imaged the cells before adding the drugs to first establish each cell's location in the cell cycle. Then we spiked in either CDK4/6i or MEKi or removed mitogens and continued imaging the cells for up to 48 additional hours. This allowed us to follow single cells as they exited into the G0-like state instead of undergoing mitosis. We now provide several Supplementary Videos from these live-cell imaging experiments to demonstrate how example cells exit into the G0-like state (new Supplementary Videos 1-5).

In Fig. 2e-g, Extended Data Fig. 4a-f, and Extended Data Fig. 5a,b we indeed measure how long it will take to get to the G0-like status for each single cell. It was these precise measurements and comparing the time it takes for cells to reach the G0-like status or mitosis that led us to develop the temporal competition model to explain cell cycle commitment.

3) The authors show in Sup Fig 1 that many other cells also respond in this way- like MCF-7, U2OS, RPE-1, HLF. The frequencies of G0-like is always low- most of the times below 10%. Now these cells are quite different in terms of their origin – cancer cells (U2OS), diploid and stable (RPE-1). How to integrate the fact that they basically respond in the same way with the G0 like population at levels of 10% or below?

By investigating different cell lines, we made the striking observation that the exit clock and the mitosis clock times appear to be coupled across cell lines. In other words, if one is longer in a particular cell line the other is also longer (new Extended Data Fig. 5h-j). When we preformed our Monte Carlo simulations for MCF7, U2OS, and RPE1 cells, in addition to the MCF-10A in the original manuscript, we found that our simple competition model accounts for the frequencies of G0-like cells for each of these cell lines (new Extended Data Fig. 5k-m). Thus, the reason these cell lines respond the same way is that the differences between the mitotic clock and the cell cycle exit clock are similar between cell lines and that our model of temporal competition can be applied to diverse cell types.

4) But more importantly, they behave very differently according to their FACS profiles in other cell cycle phases. If one looks at Supp Fig 1a, almost each cell line, behaves differently after each treatment in terms of the distribution of cell populations. For example, RPE1 cells have almost no cells in G1 or S/G2 in MeKi or CDK4/6i, while this is not the case for MCF-7 or U2OS. Does this suggest that the R point is already different for all of them? Would this mean that even the way these cells transit from one phase to the other is quite unique and particular for a given cell line and so the principles described for one (MCF10), may not apply for another cell line? This would have consequences on the generalisation of the mechanism put forward related with Cyc A expression and the balance between closer to S or M -phases. This should be tested in all the cells to be certain that even cells with different R points may follow this rule.

We would not necessarily say that the R point is already different for all of the cell lines, but it has been shown previously that whether or not a cell is born pre-R or post-R does vary between different cell lines and is also reflected in the different proportions of cells within the G0 population in untreated conditions. So yes, in some sense there is a difference in the way the different cell lines transition from mitosis to G1, some cells first enter a spontaneous quiescent state, while others are born with hyperphosphorylated Rb. This has previously been shown partly due to a competition between p21 and Cyclin D levels in the mother cell which is inherited by

the daughter cell³²⁻³⁴. However, our focus has not been on these G0 cells but rather on cells which have already entered S phase.

The reason that RPE1, HLF, and MCF-10A cells show a depleted G1 and S/G2 population in response to the various treatment conditions reflects the fact that these treatments are able to prevent cells from transitioning from G0->G1->S phase without mitogen signalling. This is in contrast to MCF7 and U2OS cells, which are transformed cells, some of which appear to be refractory to the loss of mitogen signalling and are able to complete more than one round of the cell cycle even in the absence of mitogens or in the presence of drugs that block the mitogen signalling pathway. So indeed there are also differences in the way each cell line responds to the drugs. For example, some U2OS and MCF7 cells which received the drug treatment in G2 phase are able to enter mitosis, divide, progress through the subsequent G1 and S phase and then exit to G0-like. These cells were not included in our subsequent live-cell imaging experiments since we only focused on cells that received the drug in S/G2 and then either entered mitosis and divided or exited from S/G2 to the G0-like state.

Despite these differences in the frequency of cells in different cell cycle phases at any given time, and the differences in the way cells respond to the perturbations, we present a unified model that can account for a certain percentage of cells exiting the cell cycle from S/G2 phase. As the reviewer suggests we have now included additional data to support the generality of this model for different cell lines presented in Extended Data Fig. 1.

First, we show that for U2OS, MCF7, and RPE1 cells that the probability of exiting to G0-like is highest for cells closer to S phase and further from M phase (new Extended Data Fig. 3c-e). This suggests our model of temporal competition between cell cycle exit and mitosis may also explain whether these other cell lines will exit the cell cycle or not in response to loss of mitogen signalling.

Second, we show for U2OS, MCF7, and RPE1 cells that combining a CDK1 inhibitor and a CDK4/6 inhibitor causes nearly all cells to lose CDK2 activity and exit into the G0-like state (new Extended Data Fig. 4d-f). This suggests that in the four cell lines tested, CDK2 is not self-sustaining in post-R cells.

Third, we performed Monte Carlo simulations on the experimentally measured pre-competition clocks (new Extended Data Fig. 5h-j) for cell cycle exit and mitosis clocks in U2OS, MCF7, and RPE1 cells. We find that our simple model of a competition between these two clocks can account for the frequencies of cell cycle exit that we experimentally measured (new Extended Data Fig. 5k-m).

Fourth, we measured the cyclin A2 half-life in U2OS, RPE1, and RPE1 cells edited to express an inducibly degraded cyclin A2, and found a strong correlation between the cyclin A2 half-life and the length of the cell cycle exit clock in response to CDK4/6 inhibition (new Fig. 3l).

In summary, by testing different cell lines as suggested by the reviewer, we find all of the key components of our model are supported across both transformed and non-transformed cell

lines. We believe this additional data included in our revised manuscript demonstrates that the competing clock model is applicable to diverse cell lines.

5) The authors propose that cell cycle exit vs mitotic entry maybe be decided over a kind of competition between two fates. In other words, that every cell after S-phase has the capacity to exit or enter G2 and M-phase. What contributes to the cell fate choice then is proposed to be related with how long cells have to complete the cell cycle. Thus, the authors extend “the time from S-phase to mitosis” by treating cells with HU. HU induces DNA damage and so I am not sure this is the right tool. Treatment with HU will in principle block DNA synthesis and so what the authors are measuring is the time required to finalize DNA replication and the strength of the checkpoints. If their model is correct, a cell that exists S-phase and has a lengthier G2 phase, should be less committed to enter mitosis. Maybe just prolonging G2 could be a way of testing this in \pm mitogen conditions? But ideally, this should be done independently of DNA damage.

We thank the reviewer raising this important point. First, we show that the short-term HU treatments used in this experiment do not induce DNA damage. HU inhibits ribonucleotide reductase and thus results in the depletion of deoxyribonucleotides. This has the effect of stalling active replication forks because the pool of dNTPs is not sufficient to support DNA replication. In the short term, stalled replication forks are stabilized and therefore do not result in DNA damage^{14,15}. However, long-term HU treatment does result in DNA damage. At the time scales we used HU (less than or equal to 6 hrs) there should be no DNA damage occurring according to previously published reports^{16,17}. To demonstrate this point, we treated cells with 2, 4, 6, or 24 hrs with HU and stained for the DNA damage marker γ H2AX (new Extended Data Fig. 6a,b). We find no increase in γ H2AX if treated with HU for 6 hours or less. In fact, if anything, there is a reduction in DNA damage because DNA replication is stalled and there is therefore less replication-associated endogenous damage occurring. We do see an increase in γ H2AX staining after 24 hrs of HU treatment, consistent with previous studies that show long-term HU treatment causes DNA damage^{14,16,17}.

Second, we will point out that the short-term treatment with HU only delays the S/G2 length by the precise time the cells were incubated with the treatment (eg. 2 hours of HU treatment delays S/G2 length by 2 hours; see Extended Data Fig. 7a) and the treated cells are still able to undergo mitosis. This is consistent with no DNA damage being induced, since the damage would be likely to delay mitosis by greater than the time of treatment or prevent mitosis entirely.

Third, we repeated all these experiments with thymidine to extend cell cycle length and found similar results to when we used HU (new Extended Data Fig. 6c,d and 7b,c,e,f,h).

Given these three points, we do not believe that the HU treatments we used, nor the thymidine treatments that we now include in the revised manuscript are inducing DNA damage that could be interfering with the interpretation of our results. However, we did explore the possibility of stretching S/G2 phase using other methods such as reducing the concentration of growth factors as the Reviewer as well as Reviewer 3 suggested. However, the main effect of growing cells with reduced concentrations of growth factors is not to stretch the inter-mitotic time or the S/G2 length (Rebuttal Fig. 1a,b), but to rather reduce the population of cells that enters the cell cycle in the first place (Rebuttal Fig. 1c). Thus, this approach was not useful for

testing our hypothesis.

Rebuttal Fig. 1: Reducing growth factor levels does not stretch S/G2 length.

MCF-10A cells expressing the CDK2 and APC/C activity sensors were incubated with minimal growth media supplemented with the indicated concentration of EGF and subjected to live-cell imaging for 48 hours. Note that full growth media contains 20,000 pg/mL EGF. **a**, Cumulative distribution function (CDF) plot of the intermitotic time (IMT) for cells incubated with the indicated concentration of EGF. **b**, CDF of the S/G2 length (defined as the time from APC/C inactivation to the subsequent anaphase) for cells incubated with the indicated concentration of EGF. **c**, Bar graph of the relative proportion of cells that either immediately re-entered the cell cycle (CDK2 inc), went to a temporary quiescent state before re-entering the cell cycle (CDK2 trans), or entered long-term quiescence for the duration of the imaging (CDK2 low).

6) The data related with cyclin A or Phospho Rb high/low levels presented in Supp Fig 5b. The authors state that the levels remained high for “6-8 hours before declining”. I can see a population of low levels emerging at this time, but there are still many cells in the high level category. So, how can the authors distinguish between these two?

We thank the Reviewer for raising this point and we apologise for not clearly describing the data in our original manuscript. To distinguish between phospho-pRb high and low cells we used a threshold that separates cells with low CDK2 activity ($CDK2 < 0.6$). This relationship between CDK2 activity and Rb phosphorylation has been described previously²². Since cells were pre-treated with a CDK1i then nearly all cells at the 0 hr timepoint contain hyper-phosphorylated Rb and thus can be used to identify a threshold that separates hyper- vs hypo-phosphorylated Rb (the dashed red horizontal line).

Similarly, cells with high vs low levels of cyclin A2 were distinguished by drawing a horizontal line for the 0 hr condition, where all cells contain high levels of cyclin A2. This horizontal line also separates cells which have been depleted of cyclin A2 by siRNA treatment (see Extended Data Fig. 10a), indicating that it provides a meaningful readout of cells that lack expression of cyclin A2 protein. We have now included these methodological details to the revised manuscript.

7) The effect of cyclin A depletion on CDK activity is expected. But I am confused with its effect on mitotic entry. The mitotic timer is not affected. Can cells enter mitosis without CycA? Also difficult to understand is the effect of the level of depletion according to different siRNA

concentrations. Moreover, the cell cycle exit is first detected at the same time- around 5hrs. It is true that more cells exit the cell cycle in CycA depletion, but maybe these cells have actually not even completed S-phase? I find it difficult to access how CycA depletion for a certain number of hours is not impacting the cell cycle per se. Maybe a inducible degradation system coupled to cell cycle markers would be a more useful tool.

We thank the Reviewer for raising this point and we apologise for the confusion. We have now included violin plots showing the level of depletion of cyclin A2 at the time of treatment according to the different siRNA concentrations (new Extended Data Fig. 10e). We find that at low doses of 0.5 nM siRNA which do not completely abolish cyclin A2 protein (~15% reduction in median cyclin A2 levels), that these cells are still able to enter mitosis and complete the cell cycle in only a slightly delayed fashion (a 1 hour difference in the median intermitotic for control siRNA and 0.5nM cyclin A2 siRNA treated cells) (Extended Data Fig. 10i). At higher doses of siRNA, which resulted in stronger depletion of cyclin A2 (~40% reduction or greater), we do observe a significant impact on mitotic entry as indicated by the reviewer.

Strikingly, as we show in Extended Data Fig. 10i, even though a relatively small reduction in cyclin A2 protein had only a minor effect on the mitosis clock, it resulted in a substantial reduction in the cell cycle exit clock. This result supports our model that the mitosis and cell cycle exit clocks are independent. Furthermore, in Fig. 3j we used the increasing concentrations of cyclin A2 siRNA to show that the amount of cyclin A2 present within a cell at the time of treatment influences its time to exit the cell cycle. In this experiment, cells are co-treated with a CDK1 inhibitor (which prevents entry to mitosis) and a CDK4/6 inhibitor. Thus, given the presence of a CDK1 inhibitor which prevents entry to mitosis, the effect of cyclin A2 protein depletion on mitotic entry is irrelevant since the mitosis clock is already disabled.

However, we agree with the Reviewer (as well as Reviewer 1, comment 5) that an inducible degradation system coupled with cell cycle markers would be a more useful tool to directly manipulate cyclin A2 protein stability and assess its impact on the timing of the cell cycle exit clock. To this end, we obtained an RPE1 cell line where the endogenous cyclin A2 was edited to contain two inducible degron sequences: an auxin-inducible degron (AID) and a SMASH tag (RPE CCNA2^{dd}) (new Extended Data Fig. 11a)^{12,13}. The addition of the inducible degron already reduced the stability of cyclin A2 from 10 hours in the parental line to 5 hours in the engineered line. The cyclin A2 stability could be further reduced to 2 hours by addition of a DIA “cocktail” (new Fig. 3l). To demonstrate how this reduction in the half-life of cyclin A2 impacted the cell cycle exit clock we treated cells with a CDK4/6 inhibitor alone or in combination with DIA and measured the mitosis and cell cycle exit clock for single cells (new Extended Data Fig. 11b). As expected, all cells treated with DMSO made it to mitosis and did not exit the cell cycle. CDK4/6 inhibitor treatment caused cells that were closer to S phase when treated to exit the cell cycle instead of making it to mitosis, with only cells between 10-14 hrs after S phase entry dividing (Extended Data Fig. 11c-f). Co-treatment with a CDK4/6 inhibitor and DIA had a more pronounced effect on cell cycle exit with all cells exiting the cell cycle up to 12 hrs after S phase entry. Thus, by reducing the half-life of cyclin A2 and thus decreasing the timing of the cell cycle exit clock cells became more sensitive to the loss of CDK4/6 signalling and were less likely to divide and more likely to exit the cell cycle.

To determine if cyclin A2 depleted cells are unable to complete S phase, and that this is why they are exiting the cell cycle, we treated U2OS cells with various concentrations of cyclin A2 siRNA as well as a CDK1 inhibitor and CDK4/6 inhibitor. After 24 hours, we fixed and immunostained the cells with a phospho-Rb antibody. We observed cells with 4N DNA content and hypo-phosphorylated Rb in all conditions, indicating that cell can exit the cell cycle into the G0-like state at any point during S/G2 phase. Thus, despite cyclin A2 depletion, these cells are likely completing S phase.

Rebuttal Fig. 2: Cells with reduced cyclin A2 levels exit with 4N DNA content

U2OS cells were transfected with the indicated concentration of cyclin A2 siRNA and then treated with a CDK1i and CDK4/6i to induce cell-cycle exit. The cells were then fixed and stained after 24 hours.

8) *The possibility that these results may contribute to improve chemotherapy remains highly speculative. in vivo models should be employed and I think testing this possibility in U2OS cells is really not relevant for this type of question.*

We thank the Reviewer for raising this point. Given this comment as well as the comment from Reviewer 3, we think it is best to remove Extended Data Fig. 8.

Minor points:

1) *Graphs like the ones shown in Fig 1e or Supp Fig 2b are very difficult to interpret. The 2 cell cycle trajectories described on the graph on the right in Fig 1e, cannot be identified in the graphs mentioned previously.*

We thank the Reviewer for raising this issue. Reviewer 2 made a similar point that these graphs are too small to read. To address this issue, we have now modified Fig. 1e. We have now elevated the example single-cell time courses to their own figure panel (new Fig. 1e,f), rather than an inset with the heatmaps (new Fig. 1g). We hope the increase in the size of the example cell time courses will make them easier to read. We also included a schematic in new Extended Data Fig. 2b to explain how the CDK2 and APC/C activity time courses are converted to a colormap, linearized, and stacked to generate the heatmaps shown in Fig. 1g, Fig. 2a, Extended Data Fig. 2c, and Extended Data Fig. 3a. Finally, we have now included supplementary videos (new Supplementary Video 1-3) for these same example cells. We hope that these changes will help make the data easier to interpret. By presenting the data 3 different ways (example single-cell traces, heatmaps, and example videos) readers with different scientific backgrounds will

hopefully be able to better understand the data. We hope these changes will improve the clarity and legibility of the data.

2) *The characterization of senescence Supp Fig 2C. There are small and large cells bGal positive.*

With the data shown in Extended Data. Fig. 2d (previously 2c; note the new numbering due to additional figure panels) it is difficult to determine cell size very precisely since there is not a plasma membrane marker. We will also note these images were taken with a wide-field microscope with a relatively large depth of field. This makes it difficult to ascertain the volume of cells.

3) *Supp Fig 2C lacks a scale bar.*

We thank the reviewer for pointing out this error. We have now included a scale bar and refer to it in the figure legend.

References

- 1 Siu, K. T., Rosner, M. R. & Minella, A. C. An integrated view of cyclin E function and regulation. **11**, 57-64, doi:10.4161/cc.11.1.18775 (2012).
- 2 Koepp, D. M. *et al.* Phosphorylation-dependent ubiquitination of cyclin E by the SCFFbw7 ubiquitin ligase. *Science* **294**, 173-177, doi:10.1126/science.1065203 (2001).
- 3 Won, K. A. & Reed, S. I. Activation of cyclin E/CDK2 is coupled to site-specific autophosphorylation and ubiquitin-dependent degradation of cyclin E. *The EMBO journal* **15**, 4182-4193 (1996).
- 4 Schwarz, C. *et al.* A Precise Cdk Activity Threshold Determines Passage through the Restriction Point. *Mol Cell* **69**, 253-264 e255, doi:10.1016/j.molcel.2017.12.017 (2018).
- 5 Saldivar, J. C., Cortez, D. & Cimprich, K. A. The essential kinase ATR: ensuring faithful duplication of a challenging genome. **18**, 622-636, doi:10.1038/nrm.2017.67 (2017).
- 6 Daigh, L. H., Liu, C., Chung, M., Cimprich, K. A. & Meyer, T. Stochastic Endogenous Replication Stress Causes ATR-Triggered Fluctuations in CDK2 Activity that Dynamically Adjust Global DNA Synthesis Rates. **7**, 17-27 e13, doi:10.1016/j.cels.2018.05.011 (2018).
- 7 Beijersbergen, R. L., Carlee, L., Kerkhoven, R. M. & Bernards, R. Regulation of the retinoblastoma protein-related p107 by G1 cyclin complexes. *Genes Dev* **9**, 1340-1353, doi:10.1101/gad.9.11.1340 (1995).
- 8 Farkas, T., Hansen, K., Holm, K., Lukas, J. & Bartek, J. Distinct phosphorylation events regulate p130- and p107-mediated repression of E2F-4. *J Biol Chem* **277**, 26741-26752, doi:10.1074/jbc.M200381200 (2002).
- 9 Schade, A. E., Oser, M. G., Nicholson, H. E. & DeCaprio, J. A. Cyclin D-CDK4 relieves cooperative repression of proliferation and cell cycle gene expression by DREAM and RB. *Oncogene* **38**, 4962-4976, doi:10.1038/s41388-019-0767-9 (2019).

- 10 Pomerening, J. R., Sontag, E. D. & Ferrell, J. E., Jr. Building a cell cycle oscillator: hysteresis and bistability in the activation of Cdc2. *Nat Cell Biol* **5**, 346-351, doi:10.1038/ncb954 (2003).
- 11 Sha, W. *et al.* Hysteresis drives cell-cycle transitions in *Xenopus laevis* egg extracts. **100**, 975-980, doi:10.1073/pnas.0235349100 (2003).
- 12 Crncec, A. & Hochegger, H. Degron Tagging Using mAID and SMASh Tags in RPE-1 Cells. *Methods in molecular biology* **2415**, 183-197, doi:10.1007/978-1-0716-1904-9_14 (2022).
- 13 Hegarat, N. *et al.* Cyclin A triggers Mitosis either via the Greatwall kinase pathway or Cyclin B. *The EMBO journal* **39**, e104419, doi:10.15252/emboj.2020104419 (2020).
- 14 Musialek, M. W. & Rybaczek, D. Hydroxyurea-The Good, the Bad and the Ugly. *Genes (Basel)* **12**, doi:10.3390/genes12071096 (2021).
- 15 Singh, A. & Xu, Y. J. The Cell Killing Mechanisms of Hydroxyurea. *Genes (Basel)* **7**, doi:10.3390/genes7110099 (2016).
- 16 Ercilla, A. *et al.* New origin firing is inhibited by APC/CCdh1 activation in S-phase after severe replication stress. *Nucleic Acids Res* **44**, 4745-4762, doi:10.1093/nar/gkw132 (2016).
- 17 Ercilla, A. *et al.* Acute hydroxyurea-induced replication blockade results in replisome components disengagement from nascent DNA without causing fork collapse. **77**, 735-749, doi:10.1007/s00018-019-03206-1 (2020).
- 18 Min, M., Rong, Y., Tian, C. & Spencer, S. L. Temporal integration of mitogen history in mother cells controls proliferation of daughter cells. *Science* **368**, 1261-1265, doi:10.1126/science.aay8241 (2020).
- 19 Zetterberg, A. & Larsson, O. Kinetic analysis of regulatory events in G1 leading to proliferation or quiescence of Swiss 3T3 cells. **82**, 5365-5369 (1985).
- 20 Zetterberg, A., Larsson, O. & Wiman, K. G. What is the restriction point? *Current opinion in cell biology* **7**, 835-842 (1995).
- 21 Yao, G., Lee, T. J., Mori, S., Nevins, J. R. & You, L. A bistable Rb-E2F switch underlies the restriction point. *Nat Cell Biol* **10**, 476-482, doi:10.1038/ncb1711 (2008).
- 22 Spencer, S. L. *et al.* The proliferation-quiescence decision is controlled by a bifurcation in CDK2 activity at mitotic exit. *Cell* **155**, 369-383, doi:10.1016/j.cell.2013.08.062 (2013).
- 23 Planas-Silva, M. D. & Weinberg, R. A. The restriction point and control of cell proliferation. **9**, 768-772 (1997).
- 24 Weinberg, R. A. The retinoblastoma protein and cell cycle control. *Cell* **81**, 323-330 (1995).
- 25 Chung, M. *et al.* Transient Hysteresis in CDK4/6 Activity Underlies Passage of the Restriction Point in G1. *Mol Cell* **76**, 562-573 e564, doi:10.1016/j.molcel.2019.08.020 (2019).
- 26 Sherr, C. J. Cancer cell cycles. *Science* **274**, 1672-1677 (1996).
- 27 Chen, M. *et al.* Serum starvation induced cell cycle synchronization facilitates human somatic cells reprogramming. **7**, e28203, doi:10.1371/journal.pone.0028203 (2012).
- 28 Wang, X. *et al.* Exit from quiescence displays a memory of cell growth and division. *Nat Commun* **8**, 321, doi:10.1038/s41467-017-00367-0 (2017).
- 29 Marescal, O. & Cheeseman, I. M. Cellular Mechanisms and Regulation of Quiescence. *Dev Cell* **55**, 259-271, doi:10.1016/j.devcel.2020.09.029 (2020).

- 30 Matson, J. P. *et al.* Intrinsic checkpoint deficiency during cell cycle re-entry from quiescence. **218**, 2169-2184, doi:10.1083/jcb.201902143 (2019).
- 31 Cappell, S. D., Chung, M., Jaimovich, A., Spencer, S. L. & Meyer, T. Irreversible APC(Cdh1) Inactivation Underlies the Point of No Return for Cell-Cycle Entry. *Cell* **166**, 167-180, doi:10.1016/j.cell.2016.05.077 (2016).
- 32 Arora, M., Moser, J., Phadke, H., Basha, A. A. & Spencer, S. L. Endogenous Replication Stress in Mother Cells Leads to Quiescence of Daughter Cells. **19**, 1351-1364, doi:10.1016/j.celrep.2017.04.055 (2017).
- 33 Barr, A. R. *et al.* DNA damage during S-phase mediates the proliferation-quiescence decision in the subsequent G1 via p21 expression. *Nat Commun* **8**, 14728, doi:10.1038/ncomms14728 (2017).
- 34 Yang, H. W., Chung, M., Kudo, T. & Meyer, T. Competing memories of mitogen and p53 signalling control cell-cycle entry. *Nature* **549**, 404-408, doi:10.1038/nature23880 (2017).

Reviewer Reports on the First Revision:

Referees' comments:

Referee #1 (Remarks to the Author):

The authors have done an excellent job revising their manuscript in response to all the reviewer comments.

The only remaining issue I have pertains to my previous point #4:

"4. The authors propose a linear pathway model instead of a feedback model, however the observed phenomena do not necessarily contradict the existence of a feedback loop. The feedback loop can still exist on RB, but with an extra layer of regulation through CDK4/6-p107/p130. The reversible R point model can still have feedback loops. You have cell cycle transitions driven by positive feedback that are reversible, even though they are bistable. The examination of mitosis in xenopus extracts are a good example of this (eg Sha et al PNAS 2003; Pomerening et al NCB 2003). The authors haven't disproven the existence of positive feedback as implied in their schematic."

The authors have not addressed this point and have not proven any linear model. To do so, they would have to show that cyclin A-Cdk2 activity has no impact on cyclin A synthesis - which they have definitely not done. But, that fact doesn't really impact the main finding of the paper that degradation of cyclin A leads to a reverse out of the high CDK activity state. This is true regardless of the presence or absence of a positive feedback loop centered on cyclin A since such feedback loops can be reversible as per the analysis by Jim Ferrell and colleagues of mitotic regulation.

The address this point the authors should stick more clearly to their findings. Claiming to establish a 'linear model', which they haven't established, would only add unnecessary confusion as to what their precise contribution to the field is. They need to rewrite the text and caption surrounding figure 4 to be more in line with their findings.

Referee #2 (Remarks to the Author):

All of my suggestions have been satisfactorily accommodated. I recommend publication.

Referee #3 (Remarks to the Author):

The authors have addressed all my questions and the extra Cyclin A stability experiments are beautiful and convincing. Congratulations on a nice piece of work!

Very few minor edits:

- The legend of Figure 3b refers to the wrong extended figure. I think this should now refer to Extended Figure 8.
- The legend to Extended Figure 2b uses "Mit." as an abbreviation for Mitosis which isn't used on the figure but then in Ext. Fig 2c Mit. is used to abbreviate Mitogen. Just delete this abbreviation from the legend.
- Extended Figure 8f uses the abbreviation "SS" which I guess is serum starved but this is not referred to in the legend.

Referee #4 (Remarks to the Author):

The revised version of the article by Cornwell and colleagues, on the temporal competition between opposing fates underlying cell cycle fate choices, has improved substantially. I found that the new experiments have contributed to clarify most points, with a few exceptions, which I am pointing out below. I liked the auxin degron for CycA. I think it really contributed to show the role of CycA in this process.

1) The extension of G2 with HU and thymidine. This is such an important point of this paper- G2 duration-I think the authors should try to find an alternative to this treatment.

On the DNA damage- they argue that the levels of HU and thymidine used do not generate DNA damage in the short incubation timings of 2 hrs. Although the images chosen in Sup Fig6a, show exactly this, the quantifications, even if statistically significant only at 24hrs do not show major differences. Why were they presented as log2 graphs? And how was this calculated? Total nuclei fluorescence? I could not find this information.

But even if they do not generate DNA damage, these treatments cause fork stalling and delay S-phase completion, and so, in the end they are maybe influencing G2 timing through activation of checkpoints that may impact on the regulation of cell cycle factors as the ones being characterized.

I perfectly understand that it is hard to extend G2 duration but alternative ways should be provided. Maybe partial PLK1 inhibition (specially because this is used in earlier in the paper) or WEE1 over-expression, although I have to admit that I am not sure that WEE1 over-expression will work in this system.

2) The senescence fate of the cells in the absence of mitogens and CDK4/6 inhibition or Mek inhibition. The immunolabeling is not convincing. The signal should be in the nucleus and it is in the cytoplasm and this is the case for both small and large nuclei. Better data need to be provided. In addition, by looking at the FACS data provided in SFig. 2e, I see SA-bgal+ cells in the 2n population for Meki and CDK4/6i. In SFig. 2f, the quantifications of SA-bgal+ cells includes 2n and 4n?

3) The cell cycle exit on the population of cells treated with CDK1 and CDK4/6 inhibitors (Figure 3d-e). The assumption of cell cycle exit was taken based on the low levels of CDK2? Can the authors show this in another way, for this population of cells? Like do they enter senescence? Why before time zero are these cells already pink, meaning showing low CDK activity?

4) The paper is still not easy to follow, specially the figures. Maybe indications of the cell lines used each time and the experimental set up. This may facilitate comprehension.

Author Rebuttals to First Revision:

General response to all reviewers:

In this revised manuscript, we have incorporated the suggestions from the reviewers. Below you will find a point-by-point response to each of the suggestions that were raised. Your comments are shown in *italics* and our responses are shown in **blue**.

Referee #1 (Remarks to the Author):

The authors have done an excellent job revising their manuscript in response to all the reviewer comments.

The only remaining issue I have pertains to my previous point #4:

"4. The authors propose a linear pathway model instead of a feedback model, however the observed phenomena do not necessarily contradict the existence of a feedback loop. The feedback loop can still exist on RB, but with an extra layer of regulation through CDK4/6-p107/p130. The reversible R point model can still have feedback loops. You have cell cycle transitions driven by positive feedback that are reversible, even though they are bistable. The examination of mitosis in xenopus extracts are a good example of this (eg Sha et al PNAS 2003; Pomerening et al NCB 2003). The authors haven't disproven the existence of positive feedback as implied in their schematic."

The authors have not addressed this point and have not proven any linear model. To do so, they would have to show that cyclin A-Cdk2 activity has no impact on cyclin A synthesis - which they have definitely not done. But, that fact doesn't really impact the main finding of the paper that degradation of cyclin A leads to a reverse out of the high CDK activity state. This is true regardless of the presence or absence of a positive feedback loop centered on cyclin A since such feedback loops can be reversible as per the analysis by Jim Ferrell and colleagues of mitotic regulation.

The address this point the authors should stick more clearly to their findings. Claiming to establish a 'linear model', which they haven't established, would only add unnecessary confusion as to what their precise contribution to the field is. They need to rewrite the text and caption surrounding figure 4 to be more in line with their findings.

We thank the reviewer for their feedback on this point. We agree that our data do not disprove the existence of positive feedback as we had implied in our original schematic shown in Fig. 4. Accordingly, we have now removed all references to a 'linear pathway' in Fig. 4 and the accompanying text. Instead, based on the Reviewer's suggestion, we have highlighted that our results demonstrate the presence of a 'feedforward loop' that regulates CDK2 activity in post-R cells. We have modified the text and Fig. 4. to show that this feedforward loop through CDK4/6-p107/p130-E2F4/5 provides an additional layer of regulation in addition to, but not substituting, a positive feedback loop between CDK2 and Rb. Accordingly, we have also revised our ODE model so that it includes the presence of a positive feedback loop between CDK2 and Rb, as well as the feedforward loop through CDK4/6-p107/p130-E2F4/5. Importantly, this updated ODE model still accounts for our experimental observations that loss of mitogen signalling in post-R

cells leads to reversibility in CDK2 activity in a non-hysteretic fashion. We have also updated the Discussion to more clearly state that the reversibility we find in CDK2 activity after loss of mitogen signalling occurs due to the dominant effect of the feedforward loop through CDK4/6-p107/p30-Cyclin A2 after loss of mitogen signalling, rather than stating it occurs due to a lack of positive feedback between CDK2-Rb. We agree with the Reviewer that their suggested changes do indeed provide a more appropriate description of our experimental data and better highlight the precise contribution of our work to the field.

Referee #2 (Remarks to the Author):

All of my suggestions have been satisfactorily accommodated. I recommend publication.

We thank the reviewer for their feedback.

Referee #3 (Remarks to the Author):

The authors have addressed all my questions and the extra Cyclin A stability experiments are beautiful and convincing. Congratulations on a nice piece of work!

Very few minor edits:

- The legend of Figure 3b refers to the wrong extended figure. I think this should now refer to Extended Figure 8.*
- The legend to Extended Figure 2b uses "Mit." as an abbreviation for Mitosis which isn't used on the figure but then in Ext. Fig 2c Mit. is used to abbreviate Mitogen. Just delete this abbreviation from the legend.*
- Extended Figure 8f uses the abbreviation "SS" which I guess is serum starved but this is not referred to in the legend.*

We thank the reviewer for their feedback and have made all the recommended changes to the text and figure legends.

Referee #4 (Remarks to the Author):

The revised version of the article by Cornwell and colleagues, on the temporal competition between opposing fates underlying cell cycle fate choices, has improved substantially. I found that the new experiments have contributed to clarify most points, with a few exceptions, which I am pointing out below. I liked the auxin degron for CycA. I think it really contributed to show the role of CycA in this process.

1) The extension of G2 with HU and thymidine. This is such an important point of this paper- G2 duration-I think the authors should try to find an alternative to this treatment.

On the DNA damage- they argue that the levels of HU and thymidine used do not generate DNA damage in the short incubation timings of 2 hrs. Although the images chosen in Sup Fig6a, show exactly this, the quantifications, even if statistically significant only at 24hrs do not show major differences. Why were they presented as log2 graphs? And how was this calculated? Total

nuclei fluorescence? I could not find this information.

But even if they do not generate DNA damage, these treatments cause fork stalling and delay S-phase completion, and so, in the end they are maybe influencing G2 timing through activation of checkpoints that may impact on the regulation of cell cycle factors as the ones being characterized.

I perfectly understand that it is hard to extend G2 duration but alternative ways should be provided. Maybe partial PLK1 inhibition (specially because this is used in earlier in the paper) or WEE1 over-expression, although I have to admit that I am not sure that WEE1 over-expression will work in this system.

We thank the Reviewer for their feedback and agree this is an important experiment that is technically difficult.

In the short term HU and Thymidine treatments do not cause DNA damage. As the Reviewer points out we tested whether short-term (2-6hrs) of HU/Thymidine treatment triggered a DNA damage response by staining for gammaH2AX. The data we provided in our revised manuscript shows that 2-6hrs of HU/Thymidine treatment did not lead to a significant increase in H2AX levels, while 24 hour treatment led to a significant increase in H2AX levels.

We determined the levels of H2AX by integrating the total nuclear intensity of the signal for each individual cell and were plotted on a \log_2 scale. When quantifying immunofluorescence images, it is standard in the field to present the data in \log_2 scale, reflecting the non-linearity of primary and secondary antibody binding and to aid visualization. The relative difference between each treatment condition is unchanged by plotting on a \log_2 scale and therefore this does not compromise our conclusion that there is no significant effect of 2 - 6hrs of HU/Thymidine treatment on DNA damage.

Regarding the delay in S-phase completion influencing G2 timing through activation of checkpoints. Our results show that when cells are treated with HU or Thymidine for 2 hour increments that the overall S/G2 length is increased by the time of treatment with HU/Thymidine. The fact that the extension in the length of S/G2 phase has a one-to-one relationship with the duration of treatment (i.e. 2hr treatment HU/Thymidine leads to a 2hr increase in S/G2 length) suggests that these treatments did not have any unintended effects on G2 timing through activation of checkpoints. Furthermore, to avoid any potential effect of checkpoint signalling on cell cycle factors that we are characterizing (i.e. CDK2 activity) we utilised a cell line that lacks p21 (p21^{-/-}). This rules out any potential for checkpoint activation leading to direct CDK2 inhibition by p21 as a reason why we observe cells exit the cell cycle. Furthermore, we also tested whether co-treatment of HU/Thymidine with a Wee1 inhibitor was able to prevent cell cycle exit, indicating that checkpoint activation and direct inhibition of CDK2 by Wee1 may be responsible for the cell cycle exit we observed. However, the fact we still observe a similar proportion of cells exiting the cell cycle even in the presence of a Wee1 inhibitor argues against the role of Wee1 activity in forcing cell cycle exit. Thus, we find that HU/Thymidine treatment by itself does not induce DNA damage nor trigger cell cycle checkpoints, that cells treated with HU/Thymidine are able to complete the cell cycle and divide normally, and that Wee1 and p21 do not play a role in cell cycle exit. Collectively these data

argue against a role for activation of checkpoint signalling in regulating cell cycle exit upon treatment with CDK4/6 inhibitor and HU/Thymidine.

We will also note that in all of our treatments to either block the G2/M transition or stretch S/G2 length, we observed very few cells exiting the cell cycle into the G0-like state, indicating that the treatments alone are unlikely to impact the regulation of cell cycle factors as the ones being characterized (see Rebuttal Figure 1a-e). Instead, only when we combined these treatments with a CDK4/6 inhibitor did we observe nearly all cells exiting the cell cycle into the G0-like state. This tells us the reason the cells are exiting the cell cycle is due to the CDK4/6 inhibitor, and not the various other methods we used to block or stretch S/G2 phase.

Rebuttal Figure 1.

a-e, CDK2 activity traces from single cells treated as indicated. Each of these plots appears in manuscript in the indicated figure, placed side-by-side here for easier comparison.

Regarding the use of PLK inhibition or Wee1 over-expression to extend S/G2 phase. Since both PLK inhibition and CDK1 inhibition arrests cells at the G2/M boundary we also had previously considered whether such treatments may be utilised to extend S/G2 length instead of using HU/Thymidine. However, the issue with this approach is that, for example, if we treat asynchronous cells for 2 hours with a CDK1i or PLKi this will only have an effect on cells which were about to enter mitosis, while cells that were in S or G2 phases at least 2 hours from mitosis

will not be affected by this treatment and it will not lead to an elongation of S/G2 length. Thus, PLK or CDK1 inhibition cannot effectively be used to test our hypothesis.

2) The senescence fate of the cells in the absence of mitogens and CDK4/6 inhibition or Mek inhibition. The immunolabeling is not convincing. The signal should be in the nucleus and it is in the cytoplasm and this is the case for both small and large nuclei. Better data need to be provided. In addition, by looking at the FACS data provided in SFig. 2e, I see SA-bgal+ cells in the 2n population for Meki and CDK4/6i. In SFig. 2f, the quantifications of SA-bgal+ cells includes 2n and 4n?

We thank the Reviewer for raising this point and we apologise for the confusion. Beta-galactosidase is a lysosomal enzyme that resides predominantly within dense granules in the cytoplasm¹. Senescent cells exhibit elevated beta-galactosidase activity in the cytoplasm and is often referred to as senescent-associated beta-galactosidase (SA-βGal) activity. The assay to measure SA-βGal activity involves incubating the cells with a colorimetric or fluorescent substrate of the beta-galactosidase enzyme. Whether using commercially available colorimetric or fluorescent beta-gal staining kits, the resulting staining is always entirely cytoplasmic or perinuclear (i.e. occurring around the nucleus). Thus, it is entirely normal and expected to have the staining be cytoplasmic as we show in ED Fig. 2d. We are including the website of the manufacturer that includes more details and example images showing cytoplasmic localization of the SA-β-gal activity (www.dojindo.com/products/SG04/).

We would like to point out that in Extended Data Fig. 2f, the bar graphs only represent cells exiting the cell cycle in G0-like phase and become senescent. Thus, cells with 4N DNA content, hypo phosphorylated-Rb and high SA-β-gal activity. This was also noted in the figure legend. Senescent cells with 2N DNA content would be expected based on the textbook model of cell cycle control, but senescent cells with 4N DNA content were unexpected. Therefore, we are highlighting these unexpected cells and quantifying them in ED Fig. 2f.

3) The cell cycle exit on the population of cells treated with CDK1 and CDK4/6 inhibitors (Figure 3d-e). The assumption of cell cycle exit was taken based on the low levels of CDK2? Can the authors show this in another way, for this population of cells? Like do they enter senescence? Why before time zero are these cells already pink, meaning showing low CDK activity?

We thank the Reviewer for pointing out this confusion. First, we colored the entire single-cell trace either pink or grey based on the terminal cell fate. In other words, the color does not represent the instantaneous cell state. Second, in order to address the question about whether the CDK1 and CDK4/6 inhibitor-treated cells exited the cell cycle, we have measured several key cell cycle indicators. In addition to the CDK2 activity sensor we show in the manuscript, we now show a loss of phospho-Rb staining as well as a survival assay that demonstrates the cells no longer proliferate, even after washing the drug off (Rebuttal Figure 2). Thus, the cells have permanently exited the cell cycle after being treated with the combination of CDK1 and CDK4/6 inhibitors.

4) The paper is still not easy to follow, specially the figures. Maybe indications of the cell lines used each time and the experimental set up. This may facilitate comprehension.

We thank the Reviewer for their suggestion. We have made sure to include details of each cell line used for each figure panel as well as to include as much possible detail regarding the experimental set up within the Figure legends and Methods section. We hope that our revisions will help with comprehension.

Literature cited

- 1 Tokmakov, A. A. & Sato, K. I. Activity and intracellular localization of senescence-associated beta-galactosidase in aging *Xenopus* oocytes and eggs. *Exp Gerontol* **119**, 157-167, doi:10.1016/j.exger.2019.02.002 (2019).

- 2 Moser, J., Miller, I., Carter, D. & Spencer, S. L. Control of the Restriction Point by Rb and p21. *Proc Natl Acad Sci U S A* **115**, E8219-E8227, doi:10.1073/pnas.1722446115 (2018).
- 3 Sharpless, N. E. & Sherr, C. J. Forging a signature of in vivo senescence. *Nat Rev Cancer* **15**, 397-408, doi:10.1038/nrc3960 (2015).
- 4 Mitra, M., Ho, L. D. & Collier, H. A. An In Vitro Model of Cellular Quiescence in Primary Human Dermal Fibroblasts. *Methods in molecular biology* **1686**, 27-47, doi:10.1007/978-1-4939-7371-2_2 (2018).
- 5 Min, M. & Spencer, S. L. Spontaneously slow-cycling subpopulations of human cells originate from activation of stress-response pathways. *PLoS Biol* **17**, e3000178, doi:10.1371/journal.pbio.3000178 (2019).

Reviewer Reports on the Second Revision:

Referees' comments:

Referee #4 (Remarks to the Author):

I thank the authors for addressing the points I have raised. I congratulate them on their very interesting paper.